# Bilevel Optimization over Saddle Points of Zero-Sum Markov Games

**Zihao Zheng** [1 2]  **Irwin King** [1]  **Songtao Lu** [1 2]

## Abstract

Reinforcement learning (RL) often has a hierarchical structure, where an upper-level (UL) learner selects model parameters and a lower-level (LL) decision-making process responds, naturally leading to a bilevel optimization problem. Most existing bilevel RL methods assume a single-policy LL Markov decision process (MDP), and therefore fail to capture competitive structures arising in applications such as incentive design, where multiple policies interact. We study bilevel optimization problems in which the LL problem is a regularized min–max zero-sum Markov game and the UL objective is optimized through the saddle-point equilibrium induced by the LL game. In this work, we propose penalty-augmented Nikaido–Isoda descent–ascent (PANDA), a penalty-based first-order policy-gradient method based on the Nikaido–Isoda function. By exploiting the min–max game structure, PANDA avoids computing UL hypergradients and does not require second-order information. We prove that PANDA converges to stationary points without convexity assumptions on either the UL or LL objectives. Moreover, PANDA reaches an $\epsilon$-stationary point in $\tilde{\mathcal{O}}(\epsilon^{-1})$ iterations with sample complexity $\tilde{\mathcal{O}}(\epsilon^{-3})$, matching the best-known rates for bilevel RL with single-policy LL MDPs. Experiments demonstrate the superior performance of PANDA over closely related baselines.

## 1. Introduction

Bilevel reinforcement learning (BRL) studies hierarchical decision-making settings in which an upper-level (UL) learner optimizes high-level variables that shape a lower-level (LL) reinforcement learning (RL) problem. This paradigm has recently gained momentum as a flexible tool for modeling and solving complex learning and control problems, and a growing body of work has developed principled bilevel algorithms together with convergence guarantees (Chakraborty et al., 2024; Thoma et al., 2024; Shen et al., 2025).

Despite this progress, most existing BRL methods treat the LL as a single-policy Markov Decision Process (MDP) (Chakraborty et al., 2024; Thoma et al., 2024; Gaur et al., 2025; Zeng et al., 2025). This assumption sidesteps the competitive structure that is central to many real applications, where multiple decision-makers with opposing objectives interact and the UL must reason through their equilibrium behavior. A canonical model for such multi-policy interactions is the min–max zero-sum Markov game (MMZSMG), which has been extensively studied from both theoretical and algorithmic perspectives (Bai et al., 2020; Zeng et al., 2022; Cen et al., 2024) and has found broad use in artificial intelligence (Munos et al., 2024; Kuba et al., 2025).

However, bilevel optimization over saddle points of regularized min–max zero-sum Markov games (BOSMG) at the LL remains comparatively underdeveloped. In particular, there is an urgent need for algorithms that can optimize UL variables through the LL saddle-point equilibrium while remaining computationally efficient in large-scale, sample-based settings. The main obstacle is that an MMZSMG induces strategic coupling between two opposing policies: each player's optimal response depends on the other's behavior. This competitive interaction is fundamentally different from a single-policy MDP and makes algorithm design substantially more challenging, since one must account for the simultaneous optimization of both players. As a result, many BRL methods tailored to single-policy LL MDPs do not transfer directly.

For instance, hyper policy gradient descent (HPGD) proposed in Thoma et al. (2024) and soft BRL algorithm (SoBiRL) proposed in Yang et al. (2025) use first-order oracles to construct hypergradients, but their constructions rely on structural properties specific to single-policy MDPs. In an MMZSMG, the coupled two-policy optimization breaks these derivations, rendering the corresponding hypergra-

[1]Department of Computer Science and Engineering, The Chinese University of Hong Kong [2]Shun Hing Institute of Advanced Engineering, The Chinese University of Hong Kong. Correspondence to: Songtao Lu <stlu@cse.cuhk.edu.hk>.

*Proceedings of the $43^{rd}$ International Conference on Machine Learning*, Seoul, South Korea. PMLR 306, 2026. Copyright 2026 by the author(s).

*Table 1.* Comparison of the proposed PANDA algorithm with existing methods. Deter. and Stoc. denote deterministic and stochastic settings, respectively. N/A indicates that the method does not provide theoretical guarantees on iteration complexity or sample complexity. [†] indicates methods applicable to multi-agent competitive settings. - means sample complexity is not considered in deterministic algorithms.

| Algorithms | LL Problem | Deter. or Stoc. | Iteration Complexity | Sample Complexity | Oracle |
|---|---|---|---|---|---|
| PARL (Chakraborty et al., 2024) | Max | Deter. | $\tilde{\mathcal{O}}(\epsilon^{-1})$ | - | 1st + 2nd |
| PBRL (Shen et al., 2025) | Max | Deter. | $\tilde{\mathcal{O}}(\epsilon^{-1.5})$ | - | 1st |
| HPGD (Thoma et al., 2024) | Max | Stoc. | $\tilde{\mathcal{O}}(\epsilon^{-2})$ | N/A | 1st |
| SoBiRL (Yang et al., 2025) | Max | Stoc. | $\tilde{\mathcal{O}}(\epsilon^{-1.5})$ | $\tilde{\mathcal{O}}(\epsilon^{-3.5})$ | 1st |
| First-Order BRL (Gaur et al., 2025) | Max | Stoc. | $\tilde{\mathcal{O}}(\epsilon^{-1})$ | $\tilde{\mathcal{O}}(\epsilon^{-3})$ | 1st |
| SLAC (Zeng et al., 2025) | Max | Stoc. | $\tilde{\mathcal{O}}(\epsilon^{-3})$ | $\tilde{\mathcal{O}}(\epsilon^{-3})$ | 1st |
| Meta-Gradient (Yang et al., 2022) | Min–Max [†] | Stoc. | N/A | N/A | 1st |
| DA (Wang et al., 2023) | Min–Max | Deter. | $\tilde{\mathcal{O}}(\epsilon^{-1})$ | - | 1st + 2nd |
| PBRL (Shen et al., 2025) | Min–Max | Deter. | $\tilde{\mathcal{O}}(\epsilon^{-1.5})$ | - | 1st |
| **PANDA (Ours)** | Min–Max | Stoc. | $\tilde{\mathcal{O}}(\epsilon^{-1})$ | $\tilde{\mathcal{O}}(\epsilon^{-3})$ | 1st |

dient formulas inapplicable. Likewise, penalty-based approaches in BRL, such as the first-order approach to BRL (First-Order BRL) proposed in Gaur et al. (2025) and the single-loop actor-critic algorithm (SLAC) proposed in Zeng et al. (2025), often use value-function-based reformulations that encode LL optimality as constraints for a single policy, which do not directly extend to the saddle-point structure of an MMZSMG.

Due to these challenges, only a few methods have been proposed for BOSMG. Existing approaches are either heuristic, updating UL parameters via policy-gradient steps with respect to each objective without a corresponding convergence result, as in the meta-gradient incentive design with pipelining method (Meta-Gradient) of Yang et al. (2022); or they rely on second-order information, such as Hessian inverses, which is typically computationally prohibitive at scale, as in the differentiable arbitrating (DA) algorithm of Wang et al. (2023). While a penalty-based BRL gradient-descent (PBRL) algorithm proposed in Shen et al. (2025) uses only first-order oracles, their guarantees establish convergence to a stationary point of the penalized surrogate rather than to a stationary point of the original bilevel problem. Moreover, much of the existing theoretical analysis is restricted to deterministic regimes, leaving open the stochastic setting that is standard in RL, where gradients must be estimated from sampled trajectories and sample complexity is a primary concern.

This motivates the following research question:

> *Can stochastic first-order methods solve BOSMG with provably efficient iteration and sample complexity?*

## 1.1. Related Works

**Bilevel Optimization**. Recent years have seen significant progress in developing first-order algorithms for bilevel optimization with rigorous theoretical convergence guarantees (Lu & Mei, 2024; Shen & Chen, 2023; Kwon et al., 2024; Huang, 2024; Liu et al., 2024). In particular, penalty-based methods have attracted particular attention due to their simplicity and strong empirical performance (Shen & Chen, 2023; Kwon et al., 2024; Chen et al., 2025; Lu, 2025).

When the LL problem is strongly convex, Kwon et al. (2023) reformulate the LL optimality condition as a constraint via the function value gap, and incorporate this gap into the UL objective through a penalty term. Their method alternates between optimizing the LL variables using an aggregated surrogate objective and updating the UL variables using the resulting surrogate that is approximately optimized with respect to the LL variables. Building on this nested structure, Chen et al. (2025) further show that, when the penalty coefficient is sufficiently large, this approach converges to a hypergradient-based stationary point of the original bilevel problem with near-optimal iteration complexity.

For nonconvex LL problems, a widely studied regime assumes that the LL objective satisfies the Polyak-Łojasiewicz (PŁ) condition (Lu, 2023; Xiao et al., 2023; Huang, 2024). Under this assumption, Kwon et al. (2024) and Chen et al. (2024) establish that if both the LL objective and the surrogate objective satisfy the PŁ condition, then their proposed penalty-based algorithms converge to hypergradient-based stationary points of the original bilevel problem. More recently, Jiang et al. (2025) show that if the UL objective additionally satisfies a stronger regularity requirement than PŁ, namely a flatness condition, then one can design a single-loop algorithm that still provably finds these hypergradient-based stationary points.

**Bilevel Reinforcement Learning**. Compared to single-level RL, which has been extensively studied (Sutton et al., 1999; Nachum et al., 2017; Sutton & Barto, 2018; Geist et al., 2019; Mei et al., 2020; Agarwal et al., 2021; Xiao, 2022), BRL has recently attracted growing attention for its ability to model hierarchical decision-making, particularly in the development of efficient algorithms (Chen et al., 2022; Chakraborty et al., 2024; Li et al., 2024). For example, motivated by reinforcement learning with human feedback

(RLHF), Chakraborty et al. (2024) propose an algorithm for policy alignment in RL (PARL), which requires computing second-order derivatives of the loss functions. HPGD proposed Thoma et al. (2024) and SoBiRL proposed in Yang et al. (2025) leverage closed-form characterizations of optimal policies in regularized LL MDPs, and use first-order information for estimating hypergradients, thereby avoiding second-order oracles.

In parallel, penalty-based BRL methods provide an alternative that bypasses hypergradient computation entirely while using only first-order information from the underlying objective functions. In particular, Shen et al. (2025) propose a penalty-based approach PBRL in the spirit of penalty methods in bilevel optimization. Under PŁ-type conditions, Gaur et al. (2025) propose First-Order BRL, and establish the best-known iteration and sample complexity guarantees for penalty-based BRL. More recently, Zeng et al. (2025) develop SLAC for BRL and provide convergence guarantees to hypergradient-based stationary solutions of the original BRL formulation.

**Min–Max Zero-Sum Markov Games**. Min–max zero-sum Markov games are a canonical model for competitive sequential decision making, providing a principled extension from single-agent MDPs to adversarial multi-agent settings, and their theoretical foundations are well established (Littman, 1994; Bai et al., 2020; Bai & Jin, 2020; Yang & Ma, 2023; Kalogiannis et al., 2025). Recent advances for regularized zero-sum Markov games further deliver strong algorithmic and statistical guarantees, including linear convergence to Nash equilibria under suitable regularization (Zeng et al., 2022; Cen et al., 2023; 2024; Nayak et al., 2025). Besides, a range of recent and practically important applications can be modeled through such games. For example, several lines of work in RLHF formulate preference learning via two-player zero-sum Markov games coupled with a preference model (Swamy et al., 2024; Rosset et al., 2024; Munos et al., 2024; Ye et al., 2024; Zhang et al., 2025; Zhou et al., 2025).

**Bilevel Optimization with Min–Max Structured Games at LL**. Yet, when such competitive games appear as the LL component of a bilevel problem, the landscape changes dramatically: the UL must optimize through an equilibrium induced by coupled min-max dynamics. A representative application is incentive design, where a principal at the UL shapes incentives to influence the behavior of competing agents at the LL (Yang et al., 2020; Liu et al., 2022; Yang et al., 2022).

Provably efficient methods for this setting remain scarce. Existing attempts highlight this gap. Meta-Gradient (Yang et al., 2022) targets this setting but is heuristic and provides no convergence guarantees. Hypergradient-based approaches such as Liu et al. (2022) require inverting the LL Hessian, relying on the strong assumption that second-

order information can be accurately estimated. Similarly, DA (Wang et al., 2023) estimates the Markov-game value-function Hessian via policy gradients and still requires Hessian inversion to form the hypergradient, which can be prohibitive in large-scale RL. In addition, Yao et al. (2024) also consider related bilevel problems. However, their main focus is constrained bilevel optimization, where the LL problem involves functional constraints and is reformulated via a Lagrangian-based gap function, leading to a convex-linear primal-dual saddle-point structure rather than a zero-sum Markov game. Closest in spirit to our approach, Shen et al. (2025) propose a first-order penalty method by aggregating the UL objective with a Nikaido–Isoda (NI) gap term constructed from the LL loss.

## 1.2. Main Contributions of This Work

In this work, we propose penalty-augmented Nikaido–Isoda descent–ascent (PANDA), a first-order method for solving BRL problems in which the LL is a regularized MMZSMG and the UL optimizes through the induced saddle-point equilibrium. Using a penalty-based reformulation, PANDA relies only on first-order information and avoids UL hypergradient computation, thereby eliminating the need for second-order derivatives. Moreover, by exploiting the intrinsic structure of regularized MMZSMG, we prove that PANDA converges to approximate stationary points of the original bilevel problem without imposing restrictive conditions, such as strong convexity, on either the UL or LL objectives. To the best of our knowledge, PANDA is the first stochastic first-order algorithm for bilevel problems with a regularized MMZSMG at the LL that achieves $\tilde{\mathcal{O}}(\epsilon^{-1})$ iteration complexity and $\tilde{\mathcal{O}}(\epsilon^{-3})$ sample complexity, matching the state-of-the-art rates for BRL with a single-policy MDP at the LL (Gaur et al., 2025).

The main contributions of this work are highlighted as follows:

▶ We propose PANDA, a first-order, stochastic, policy-gradient-based algorithm for BOSMG, making it readily applicable to large-scale and sample-based settings.

▶ To the best of our knowledge, this is the first theoretical result showing that a stochastic first-order method can find an $\epsilon$-stationary point of the original bilevel problem in $\tilde{\mathcal{O}}(\epsilon^{-1})$ iterations with $\tilde{\mathcal{O}}(\epsilon^{-3})$ samples, matching the best-known rates for BRL with a single-policy LL MDP.

▶ We establish several structural results for the underlying BRL formulation, including the uniqueness of LL saddle points under general regularization, as well as smoothness and a non-uniform PŁ property of the NI function. These properties yield new insights into regularized MMZSMGs and may serve as useful tools for future work.

▶ We validate PANDA across multiple environments, where it consistently outperforms closely related baseline methods.

## 2. Bilevel Optimization over Saddle Points of Zero-Sum Markov Games

In this work, we study the following BRL problem, where the LL problem is a regularized MMZSMG:

$$\min_{x,\phi,\psi} f(x,\phi,\psi) \tag{1a}$$

$$\text{s.t. } (\phi,\psi) \in \arg\min_{\phi'} \max_{\psi'} J(x,\phi',\psi') \tag{1b}$$

where $f$ denotes the UL objective, $\phi$ and $\psi$ are learnable policy parameters, and $J$ is the LL objective. Here, the UL objective $f$ is evaluated at the UL decision variable $x$ and the optimal policy pair $(\pi_\phi, \pi_\psi)$ induced by the LL problem. We nevertheless write it as $f(x,\phi,\psi)$ for notational convenience. In particular, $J$ corresponds to the regularized value function, which we define explicitly below.

### 2.1. Regularized Min–Max Zero-Sum Markov Games at LL

Let $\mathcal{M}(x) \triangleq \{\mathcal{S}, \mathcal{A}, \mathcal{B}, r_x, \mathcal{P}, \gamma, h\}$ be a regularized MMZSMG, where $\mathcal{S}$ is a finite state space; $\mathcal{A}$ and $\mathcal{B}$ are finite action spaces for the min-player and max-player, respectively; $r_x : \mathcal{S} \times \mathcal{A} \times \mathcal{B} \mapsto \mathbb{R}$ is the parameterized reward function; $\mathcal{P} : \mathcal{S} \times \mathcal{A} \times \mathcal{B} \mapsto \Delta_{|\mathcal{S}|}$ is the transition kernel; $\gamma \in (0,1)$ is the discount factor; and $h = (h_s)_{s \in \mathcal{S}}$ is a regularizer, where each $h_s : \Delta_{|\mathcal{A}|} \times \Delta_{|\mathcal{B}|} \mapsto \mathbb{R}_+$ is strongly convex in the min-player's mixed strategy and strongly concave in the max-player's mixed strategy at state $s$. Furthermore, let $\pi_\phi(\cdot|s) \in \Delta_{|\mathcal{A}|}$ denote the min-player's policy parameterized by $\phi$ for each state $s$, and similarly let $\pi_\psi(\cdot|s) \in \Delta_{|\mathcal{B}|}$ denote the max-player's policy parameterized by $\psi$ for each state $s$. We consider policy parameterizations that are expressive enough to represent all the mixed policies in the policy class.

For any policy pair $(\pi_\phi, \pi_\psi)$, the regularized state-value function at state $s$ is defined as

$$V_{\mathcal{M}(x)}^{\pi_\phi,\pi_\psi}(s) \triangleq \mathbb{E}\left[ \sum_{t=0}^{\infty} \gamma^t \left( r_x(s_t, a_t, b_t) \right.\right.$$

$$\left.\left. + h_{s_t}\left( \pi_\phi(\cdot|s_t), \pi_\psi(\cdot|s_t) \right) \right) \middle| s_0 = s, \phi, \psi, \mathcal{M}(x) \right]. \tag{2}$$

For an initial state distribution $\rho \in \Delta_{|\mathcal{S}|}$ with full support, i.e., $\min_{s \in \mathcal{S}} \rho(s) > 0$, we define $V_{\mathcal{M}(x)}^{\pi_\phi,\pi_\psi}(\rho) \triangleq \mathbb{E}_{s \sim \rho} V_{\mathcal{M}(x)}^{\pi_\phi,\pi_\psi}(s)$. For notational simplicity, we write $J(x,\phi,\psi)$ to denote $V_{\mathcal{M}(x)}^{\pi_\phi,\pi_\psi}(\rho)$ whenever the dependence is clear from context.

Due to the strong convexity–concavity of the regularizer, the LL min–max problem admits a unique optimal policy pair, which coincides with the saddle point of the game in the policy space.

**Proposition 1.** *For any given $x$, the regularized MMZSMG admits a unique equilibrium policy pair $(\pi_\phi^*(x), \pi_\psi^*(x))$*

*such that for any $\phi$ and $\psi$,*

$$V_{\mathcal{M}(x)}^{\pi_\phi^*(x),\pi_\psi}(\rho) \leq V_{\mathcal{M}(x)}^{\pi_\phi^*(x),\pi_\psi^*(x)}(\rho) \leq V_{\mathcal{M}(x)}^{\pi_\phi,\pi_\psi^*(x)}(\rho). \tag{3}$$

*Moreover, we have*

$$V_{\mathcal{M}(x)}^{\pi_\phi^*(x),\pi_\psi^*(x)}(\rho) = \min_\phi \max_\psi V_{\mathcal{M}(x)}^{\pi_\phi,\pi_\psi}(\rho)$$

$$= \max_\psi \min_\phi V_{\mathcal{M}(x)}^{\pi_\phi,\pi_\psi}(\rho).$$

This optimal policy pair is also referred to as the Nash equilibrium (NE) of the regularized MMZSMG. Proposition 1 establishes, to the best of our knowledge, the most general uniqueness guarantee for equilibrium policies in regularized MMZSMG, in the sense that the regularizer $h_s$ can be *any* state-wise strongly convex–concave function with arbitrary strong convexity/concavity *moduli*. For concreteness, throughout the remainder of this work we focus on entropy regularization, i.e.,

$$h_s(\pi_\phi(\cdot|s),\pi_\psi(\cdot|s)) = -\tau_\phi H(\pi_\phi(\cdot|s)) + \tau_\psi H(\pi_\psi(\cdot|s)), \tag{4}$$

where $\tau_\phi > 0$ and $\tau_\psi > 0$ are the regularization coefficients. *Remark* 1. Other regularization choices are also covered by Proposition 1. For example, a KL-based regularizer, $h_s(\pi_\phi(\cdot|s),\pi_\psi(\cdot|s)) = -\tau_\phi \text{KL}(\pi_\phi(\cdot|s) \| \pi_{\text{ref}}(\cdot|s)) + \tau_\psi \text{KL}(\pi_\psi(\cdot|s) \| \pi_{\text{ref}}(\cdot|s))$, is a special case that is often useful in practice, where $\pi_{\text{ref}}$ denotes a reference policy.

Consequently, problem (1) can be equivalently reformulated as

$$\min_x F(x) \triangleq f(x, \phi^*(x), \psi^*(x)). \tag{5}$$

Here, $\phi^*(x) \in \{\phi : \pi_\phi = \pi_\phi^*(x)\}$ and $\psi^*(x) \in \{\psi : \pi_\psi = \pi_\psi^*(x)\}$ denote parameters that induce the optimal policy pair of the LL min–max game for a given $x$. We note that the UL objective $f$ is evaluated at the equilibrium policy pair induced by the LL problem. Although there may exist multiple optimal parameter pairs, they all induce the same unique equilibrium policy pair $(\pi_\phi^*(x), \pi_\psi^*(x))$ and hence yield the same value of $f$.

### 2.2. Penalty-Based Reformulation of BOSMG

However, computing the hypergradient of $F(x)$ with respect to $x$ via the chain rule is computationally expensive, since the parameter $x$ couples the UL and LL problems. By Proposition 1, the policy pair induced by $(\phi^*(x), \psi^*(x))$ coincides with a saddle point of the LL min–max problem. This observation motivates the use of the classical NI function (Nikaidô & Isoda, 1955) to quantify how far the policy pair induced by $(\phi, \psi)$ is from equilibrium in the LL MMZSMG. Specifically, the NI function for our problem is defined as

$$g(x,\phi,\psi) \triangleq \max_{\psi'} J(x,\phi,\psi') - \min_{\phi'} J(x,\phi',\psi), \tag{6}$$

which is always nonnegative and equals zero if and only if the policy pair induced by $(\phi, \psi)$ is an NE of the MMZSMG $\mathcal{M}(x)$ (Von Heusinger & Kanzow, 2009).

Therefore, it is natural to use the NI function to reformulate the original BRL problem (1) as

$$\min_{x, \phi, \psi} f(x, \phi, \psi) \quad \text{s.t.} \quad g(x, \phi, \psi) \le 0. \qquad (7)$$

One of the most direct approaches to solving the resulting single-constraint optimization problem (7) is to employ a penalty method (Wright et al., 1999; Kwon et al., 2024; Chen et al., 2025).

Subsequently, we define the following penalty-based objective function:

$$\mathcal{L}_\lambda(x, \phi, \psi) \triangleq f(x, \phi, \psi) + \lambda g(x, \phi, \psi) \qquad (8)$$

for some penalty parameter $\lambda > 0$. We then define the associated hyper-objective as the minimum value of $\mathcal{L}_\lambda(x, \phi, \psi)$ for a given $x$:

$$\mathcal{L}_\lambda^*(x) \triangleq \min_{\phi, \psi} f(x, \phi, \psi) + \lambda g(x, \phi, \psi), \qquad (9)$$

which can serve as a surrogate for $F(x)$ in (5). This leads to a simple strategy for solving (1): for a given $\lambda > 0$, first solve the inner penalty-based problem to obtain $(\phi_\lambda^*(x), \psi_\lambda^*(x)) \in \arg\min_{\phi, \psi} \mathcal{L}_\lambda(x, \phi, \psi)$, and then update $x$ by minimizing the resulting surrogate objective $\mathcal{L}_\lambda^*(x)$, i.e., $\min_x \mathcal{L}_\lambda^*(x)$. Next, we propose a policy-gradient-based algorithm for solving this class of BOSMG.

### 2.3. Policy Gradient Methods for BOSMG

Motivated by the nested structure induced by the penalty-based reformulation of (1), our algorithm consists of three components.

**Step 1: Best-response approximation.** (lines 7 – 10 in PANDA) Solving (8) requires estimating the NI function, which in turn entails approximately solving the two best response problems in (6). To this end, we apply policy-gradient descent/ascent to obtain $\tilde{\phi}$ and $\tilde{\psi}$ as effective approximations of the best-response parameters.

To compute policy gradients $\nabla_\phi J(x, \phi, \psi)$ and $\nabla_\psi J(x, \phi, \psi)$, we further define the joint action-value function as

$$Q_{\mathcal{M}(x)}^{\pi_\phi, \pi_\psi}(s, a, b) = r_x(s, a, b) + \gamma \mathbb{E}_{s'} V_{\mathcal{M}(x)}^{\pi_\phi, \pi_\psi}(s'). \qquad (10)$$

Then the policy gradient $\nabla_\phi J(x, \phi, \psi)$ (cf. Lemma B.6 in Appendix) is given by

$$\nabla_\phi J(x, \phi, \psi) = \mathbb{E}\Big[ \sum_{t=0}^{\infty} \gamma^t \nabla_\phi \log \pi_\phi(a_t|s_t)$$

$$\cdot \big( Q_{\mathcal{M}(x)}^{\pi_\phi, \pi_\psi}(s_t, a_t, b_t) - \tau_\psi \log \pi_\psi(b_t|s_t) + \tau_\phi \log \pi_\phi(a_t|s_t) \big) \Big],$$

where the expectation is taken over trajectories generated by the policy pair $(\pi_\phi, \pi_\psi)$ in the Markov game $\mathcal{M}(x)$.

In implementation, we sample a batch of $B_J$ trajectories with truncated horizon length $H$ and approximate the expectation by the sample average. The resulting sample-based gradient estimators are given by

$$\nabla_\phi \hat{J}(x, \phi, \psi; B_J, H) = \frac{1}{B_J} \sum_{i=1}^{B_J} \nabla_\phi \hat{J}_i(x, \phi, \psi; H), \quad (11)$$

and the per-trajectory gradient estimator is given by

$$\nabla_\phi \hat{J}_i(x, \phi, \psi; H) = \sum_{t=0}^{H-1} \gamma^t \nabla_\phi \log \pi_\phi(a_{i,t}|s_{i,t})$$

$$\cdot \Big( \widehat{Q}_{\mathcal{M}(x)}^{\pi_\phi, \pi_\psi}(s_{i,t}, a_{i,t}, b_{i,t}) - \tau_\psi \log \pi_\psi(b_{i,t}|s_{i,t})$$

$$+ \tau_\phi \log \pi_\phi(a_{i,t}|s_{i,t}) \Big), \qquad (12)$$

where $\widehat{Q}_{\mathcal{M}(x)}^{\pi_\phi, \pi_\psi}(s, a, b)$ denotes an empirical estimate of the Q-function, and $\{s_{i,t}, a_{i,t}, b_{i,t}\}_{t=0}^{H-1}$ is the $i$th sampled trajectory of length $H$ generated by $s_0 \sim \rho$, $a_t \sim \pi_\phi(\cdot|s_t)$, $b_t \sim \pi_\psi(\cdot|s_t)$, and $s_{t+1} \sim \mathcal{P}(\cdot|s_t, a_t, b_t)$. In this work, we use Monte Carlo roll-outs to estimate the policy gradients. The gradients $\nabla_\psi J(x, \phi, \psi)$ and $\nabla_\psi \hat{J}(x, \phi, \psi; B_J, H)$ are defined and estimated analogously.

**Step 2: Penalty-subproblem approximation.** (lines 11 – 12 in PANDA) The next step is to obtain $\phi_\lambda^*(x)$ and $\psi_\lambda^*(x)$ by solving the inner minimization problem

$$\min_{\phi, \psi} h(x, \phi, \psi) \triangleq \frac{1}{\lambda} f(x, \phi, \psi) + g(x, \phi, \psi). \qquad (13)$$

Using the updated $\tilde{\phi}$ and $\tilde{\psi}$ obtained from the best-response approximation step, we are able to estimate the NI function via

$$\tilde{g}(x, \phi, \psi, \tilde{\phi}, \tilde{\psi}) \triangleq J(x, \phi, \tilde{\psi}) - J(x, \tilde{\phi}, \psi), \qquad (14)$$

and use it as a surrogate of $g(x, \phi, \psi)$. We then update $(\phi, \psi)$ via stochastic gradient descent using the following estimated gradient:

$$\nabla_{(\phi, \psi)} \tilde{h}(x, \phi, \psi, \tilde{\phi}, \tilde{\psi})$$

$$= \frac{1}{\lambda} \nabla_{(\phi, \psi)} f(x, \phi, \psi) + \nabla_{(\phi, \psi)} \tilde{g}(x, \phi, \psi, \tilde{\phi}, \tilde{\psi}). \quad (15)$$

Here, $\nabla_{(\phi, \psi)} f(x, \phi, \psi)$ is estimated from a batch of size $B$ in implementation as $\nabla_{(\phi, \psi)} \hat{f}(x, \phi, \psi; B) = \frac{1}{B} \sum_{i=1}^{B} \nabla_{(\phi, \psi)} \hat{f}_i(x, \phi, \psi)$, where $\hat{f}_i(x, \phi, \psi)$ denotes the per-sample objective estimate. After $K$ update steps, we obtain $(\phi^K, \psi^K)$, which is expected to be close to the optimal solution $(\phi_\lambda^*(x), \psi_\lambda^*(x))$.

**Step 3: Hypergradient step.** (lines 16 – 17 in PANDA) Finally, we apply stochastic gradient descent to update the

**Algorithm 1** PANDA: Penalty-Augmented Nikaido–Isoda based Descent–Ascent

---

1: **Input:** Initial parameters $x_0, \phi_0, \psi_0$, trajectory sample batch size $B_J$, UL sample batch size $B$, horizon length $H$.
2: Init $(y_0, z_0) \leftarrow (\phi_0, \psi_0)$ and $(\tilde{y}_0, \tilde{z}_0) \leftarrow (\phi_0, \psi_0)$
3: **for** $t = 0$ **to** $T - 1$ **do**
4: $\quad (\phi_t^0, \psi_t^0) \leftarrow (y_t, z_t)$
5: $\quad (\tilde{\phi}_t^0, \tilde{\psi}_t^0) \leftarrow (\tilde{y}_t, \tilde{z}_t)$
6: $\quad$ **for** $k = 0$ **to** $K - 1$ **do**
7: $\qquad u_t^k \leftarrow \nabla_\phi \hat{J}(x_t, \tilde{\phi}_t^k, \psi_t^k; B_J, H)$
8: $\qquad v_t^k \leftarrow \nabla_\psi \hat{J}(x_t, \phi_t^k, \tilde{\psi}_t^k; B_J, H)$
9: $\qquad \tilde{\phi}_t^{k+1} \leftarrow \tilde{\phi}_t^k - \eta_\phi u_t^k$
10: $\qquad \tilde{\psi}_t^{k+1} \leftarrow \tilde{\psi}_t^k + \eta_\psi v_t^k$
11: $\qquad g_t^k \leftarrow \nabla_{(\phi,\psi)} \widehat{h}(x_t, \phi_t^k, \psi_t^k, \tilde{\phi}_t^{k+1}, \tilde{\psi}_t^{k+1}; B_J, B, H)$
12: $\qquad (\phi_t^{k+1}, \psi_t^{k+1}) \leftarrow (\phi_t^k, \psi_t^k) - \eta_\theta g_t^k$
13: $\quad$ **end for**
14: $\quad (y_{t+1}, z_{t+1}) \leftarrow (\phi_t^K, \psi_t^K)$
15: $\quad (\tilde{y}_{t+1}, \tilde{z}_{t+1}) \leftarrow (\tilde{\phi}_t^K, \tilde{\psi}_t^K)$
16: $\quad \ell_t \leftarrow \nabla_x \widehat{\mathcal{L}}_\lambda(x_t, y_{t+1}, z_{t+1}, \tilde{y}_{t+1}, \tilde{z}_{t+1}; B, B_J, H)$
17: $\quad x_{t+1} \leftarrow x_t - \eta_x \ell_t$
18: **end for**

---

UL parameter $x$ using the following estimate of the hypergradient $\nabla_x \mathcal{L}_\lambda^*(x)$ in (18):

$$\nabla_x \tilde{\mathcal{L}}_\lambda(x, \phi^K, \psi^K, \tilde{\phi}^K, \tilde{\psi}^K)$$
$$= \nabla_x f(x, \phi^K, \psi^K) + \lambda \nabla_x \tilde{g}(x, \phi^K, \psi^K, \tilde{\phi}^K, \tilde{\psi}^K), \quad (16)$$

where $\nabla_x f(x, \phi^K, \psi^K)$ is estimated analogously to $\nabla_{(\phi,\psi)} f(x, \phi, \psi)$. Moreover, the gradient $\nabla_x J(x, \phi, \psi)$ (cf. Lemma B.5 in Appendix) is given by $\nabla_x J(x, \phi, \psi) = \mathbb{E}\left[\sum_{t=0}^\infty \gamma^t \nabla_x r_x(s_t, a_t, b_t)\right]$, which we estimate using Monte Carlo roll-outs as $\nabla_x \hat{J}(x, \phi, \psi; B_J, H) = \frac{1}{B_J} \sum_{i=1}^{B_J} \nabla_x \hat{J}_i(x, \phi, \psi; H)$ with per-sample gradient $\nabla_x \hat{J}_i(x, \phi, \psi; H) = \sum_{t=0}^{H-1} \gamma^t \nabla_x r_x(s_{i,t}, a_{i,t}, b_{i,t})$. The detailed implementation of this policy-gradient-based algorithm is summarized in Algorithm 1.

## 3. Theoretical Analysis of PANDA

In this section, we present the main theoretical convergence results for the proposed PANDA algorithm. We begin by stating the assumptions used in our analysis.

First, we impose the following blanket assumptions on $\mathcal{M}(x)$ to ensure the well-posedness of the LL problem and to facilitate our convergence analysis.

**Assumption 1.** Suppose the following conditions hold:

1. The reward function $r_x$ is bounded by $B_r$, i.e., $|r_x(s, a, b)| \leq B_r$ for any $x, s, a,$ and $b$.

2. The reward function $r_x(s, a, b)$ is $C_r$-Lipschitz continuous and $L_r$-smooth in $x$, for any state $s$, action $a$, and action $b$.
3. There exists some constant $\delta_\pi > 0$ such that $\pi_\phi(a|s) \geq \delta_\pi$ for any $(s, a) \in \mathcal{S} \times \mathcal{A}$ and $\pi_\psi(b|s) \geq \delta_\pi$ for any $(s, b) \in \mathcal{S} \times \mathcal{B}$.
4. There exists some constant $\delta_\rho > 0$ such that the initial distribution satisfies $\rho(s) \geq \delta_\rho$ for any state $s$.

*Remark* 2. These conditions are commonly adopted in the analysis of Markov games and BRL (Zeng et al., 2022; Gaur et al., 2025). Next, we turn to the assumptions on the objective functions.

**Assumption 2.** (Lipschitz continuity and PŁ condition) Suppose the following conditions hold:

1. $f(x, \phi, \psi)$ is twice differentiable, $C_f$-Lipschitz continuous, $L_{f,1}$-smooth and $L_{f,2}$-Hessian Lipschitz continuous in $(x, \phi, \psi)$.
2. $J(x, \phi, \psi)$ is twice differentiable, $C_J$-Lipschitz continuous, $L_{J,1}$-smooth and $L_{J,2}$-Hessian Lipschitz continuous in $(x, \phi, \psi)$.
3. There exists $\lambda_0 > 0$ such that for any given $x$ and $c \in (0, \lambda_0^{-1}]$, $cf(x, \phi, \psi) + g(x, \phi, \psi)$ is $\mu_h$-PŁ in $(\phi, \psi)$.

*Remark* 3. The first two conditions are standard Lipschitz-continuity assumptions on the UL and LL objective functions, and are widely used in bilevel optimization and BRL (Chen et al., 2024; Gaur et al., 2025). Under Assumption 1, the Lipschitz continuity and smoothness of $J$ can be established using Lemmas 5 and 6 in Zeng et al. (2022), together with straightforward derivations. These conditions do not introduce any additional restrictions beyond Assumption 1. However, the Lipschitz continuity of the Hessian of $J$ does not follow from Assumption 1, and is required to ensure that stationary points are well-defined.

*Remark* 4. The last condition is essential for establishing the correspondence between stationary points of the original bilevel problem and those of the penalty reformulation when $\lambda$ is sufficiently large. It also guarantees that the hyper-objective $\mathcal{L}_\lambda^*(x)$ is differentiable (Kwon et al., 2024; Chen et al., 2024), even when the loss functions $f$, $g$, or their combination (e.g., $cf + g$) are possibly nonconvex. This assumption is broadly adopted in the bilevel optimization literature (Kwon et al., 2024; Chen et al., 2024; Jiang et al., 2025) to facilitate convergence analysis, and has also been used in recent BRL works (Gaur et al., 2025; Zeng et al., 2025).

*Remark* 5. It is worth noting that this assumption is not required when $c = 0$. In particular, we can show that the NI function $g(x, \phi, \psi)$ automatically satisfies a non-uniform PŁ condition with respect to $(\phi, \psi)$; see Lemma 1.

**Policy parameterization.** We adopt a tabular softmax parameterization for both players' policies. This parameterization is commonly used in solving RL problems (Mei et al., 2020; Zeng et al., 2022; 2025).

We further make the following standard assumption on the stochastic gradient estimators of the UL objective function.

**Assumption 3.** Suppose the gradient estimator of the UL objective is unbiased and has bounded variance $\sigma_f^2$.

### 3.1. Properties in BOSMG

**Properties of the NI function.** We next show that the NI function $g(x, \phi, \psi)$ satisfies a non-uniform PŁ property that arises intrinsically from the regularized MMZSMG structure.

**Lemma 1.** *For any $x$ and any $(\phi, \psi)$, $g(x, \phi, \psi)$ satisfies the following condition:*

$$\frac{1}{2}\|\nabla_{(\phi,\psi)}g(x,\phi,\psi)\|_2^2 \geq \mu(\phi,\psi)g(x,\phi,\psi), \quad (17)$$

*where $\mu(\phi,\psi) = (1 - \gamma)\frac{\min\{\tau_\phi, \tau_\psi\}}{|\mathcal{S}|}\min_s \rho^2(s)$ $\min\{\min_{s,a}\pi_\phi^2(a|s), \min_{s,b}\pi_\psi^2(b|s)\} > 0.$*

*Remark 6.* Lemma 1 extends the non-uniform PŁ condition of the soft value function established in Mei et al. (2020) to the regularized MMZSMG setting. This property is crucial for analyzing the convergence of our proposed PANDA algorithm.

In addition to the non-uniform PŁ property above, we can show that the NI function $g(x, \phi, \psi)$ also satisfies the following Lipschitz-type regularity properties.

**Lemma 2.** *Under Assumptions 1 and 2, the NI function $g(x, \phi, \psi)$ defined in (6) is $\mu_g$-PŁ in $(\phi, \psi)$ for some constant $\mu_g > 0$, $C_g$-Lipschitz continuous, $L_{g,1}$-smooth and $L_{g,2}$-Hessian Lipschitz continuous in $(x, \phi, \psi)$ for some constants $C_g > 0$, $L_{g,1} > 0$ and $L_{g,2} > 0$.*

**Properties of the hypergradient.** Under the above assumptions and results, the gradient of $\mathcal{L}_\lambda^*(x)$ is given by Chen et al. (2024) as

$$\nabla_x \mathcal{L}_\lambda^*(x)$$
$$= \nabla_x f(x, \phi_\lambda^*(x), \psi_\lambda^*(x)) + \lambda \nabla_x g(x, \phi_\lambda^*(x), \psi_\lambda^*(x))$$
$$= \nabla_x f(x, \phi_\lambda^*(x), \psi_\lambda^*(x)) + \lambda \nabla_x J(x, \phi_\lambda^*(x), \tilde{\psi}^*(x))$$
$$- \lambda \nabla_x J(x, \tilde{\phi}^*(x), \psi_\lambda^*(x)), \quad (18)$$

where $\tilde{\phi}^*(x) = \arg\min_\phi J(x, \phi, \psi_\lambda^*(x))$ and $\tilde{\psi}^*(x) = \arg\max_\psi J(x, \phi_\lambda^*(x), \psi)$.

*Remark 7.* By Lemma 4.3 of Chen et al. (2024), the hypergradient $\nabla F(x)$ exists and is defined as $\lim_{\lambda \to +\infty} \nabla \mathcal{L}_\lambda^*(x)$. Moreover, the gradient mismatch between $\nabla F(x)$ and $\nabla \mathcal{L}_\lambda^*(x)$ is bounded as $\|\nabla F(x) - \nabla \mathcal{L}_\lambda^*(x)\| = \mathcal{O}(\lambda^{-1})$. Consequently, to obtain an $\mathcal{O}(\epsilon)$-stationary point of the original hyper-objective $F(x)$, where stationarity is measured by the squared gradient norm $\|\nabla F(x)\|^2$, it suffices to compute an $\mathcal{O}(\epsilon)$-stationary point of the surrogate objective $\mathcal{L}_\lambda^*(x)$, measured by $\|\nabla \mathcal{L}_\lambda^*(x)\|^2$, with a sufficiently large penalty parameter $\lambda = \mathcal{O}(\epsilon^{-1/2})$.

### 3.2. Convergence Results of PANDA

For clarity, we first summarize the additional constants used in the following theorem. $L_F$ is the smoothness constant of the hyper-objective $F$, $L_{h,1}$ is the smoothness constant of the penalized objective $h$ for $\lambda \geq \lambda_0$. Their existence under Assumptions 1 and 2 is established in Appendix C.3. We also define $\kappa \triangleq \frac{\max\{L_{J,1}, L_{g,1}, L_{h,1}\}}{\min\{\mu_h, \mu_g\}} > 0$.

We now present the following convergence guarantees for Algorithm 1.

**Theorem 1.** *Suppose Assumptions 1 to 3 hold. Let $\lambda \geq \lambda_0$, and let the step sizes satisfy*

$$\eta_x \leq \frac{1}{2L_F}, \eta_\theta \asymp \kappa^{-5}, \eta_\phi, \eta_\psi \leq \min\left\{\frac{1}{L_{J,1}}, \frac{1}{\mu_g + 1}, 1\right\}.$$

*Then, for Algorithm 1 with $K = \mathcal{O}(\log \lambda)$, we have*

$$\frac{1}{T}\sum_{t=0}^{T-1}\mathbb{E}\left[\|\nabla F(x_t)\|^2\right] \leq \mathcal{O}\left(\frac{\Delta_F}{T}\right) + \mathcal{O}\left(\frac{\sigma_f^2}{B}\right)$$
$$+ \mathcal{O}(\lambda^{-2}) + \mathcal{O}(\lambda^2 \gamma^{2H} H^2) + \mathcal{O}\left(\frac{\lambda^2}{B_J}\right), \quad (19)$$

*where $\Delta_F \triangleq \mathbb{E}[F(x_0) - F(x_T)]$.*

**Corollary 1.** *Under the same conditions as Theorem 1, choose $T = \mathcal{O}(\epsilon^{-1})$, $\lambda = \mathcal{O}(\epsilon^{-1/2})$, $K = \mathcal{O}(\log \epsilon^{-1})$, $H = \mathcal{O}(\log \epsilon^{-1})$, $B_J = \mathcal{O}(\epsilon^{-2})$, and $B = \mathcal{O}(\epsilon^{-1})$. Then the iterates generated by Algorithm 1 satisfy*

$$\frac{1}{T}\sum_{t=0}^{T-1}\mathbb{E}\left[\|\nabla F(x_t)\|^2\right] \leq \mathcal{O}(\epsilon). \quad (20)$$

*Remark 8.* Theorem 1 implies that, to obtain an $\epsilon$-stationary point measured by $T^{-1}\sum_{t=0}^{T-1}\mathbb{E}[\|\nabla F(x_t)\|^2] \leq \mathcal{O}(\epsilon)$, the required number of outer iterations is $T = \mathcal{O}(\epsilon^{-1})$. Moreover, since each outer iteration performs $K$ inner updates and uses $B$ and $B_J$ samples, the total sample complexity of PANDA scales as $T \cdot K \cdot (B + B_J) = \tilde{\mathcal{O}}(\epsilon^{-3})$, where $\tilde{\mathcal{O}}(\cdot)$ hides logarithmic factors.

*Remark 9.* The resulting sample complexity bound matches the best-known guarantees in the BRL literature (Gaur et al., 2025; Zeng et al., 2025); however, these works consider LL problems with a single optimization direction (either minimization or maximization), rather than a coupled min–max game.

## 4. Numerical Experiments

In this section, we present numerical experiments comparing the proposed PANDA algorithm for solving BOSMG problems with closely related baselines, including META (Yang et al., 2022), DA (Wang et al., 2023), and PBRL (Shen et al., 2025).

### 4.1. Synthetic Problem

We first consider a synthetic problem motivated by incentive design (Yang et al., 2020), where the UL objective

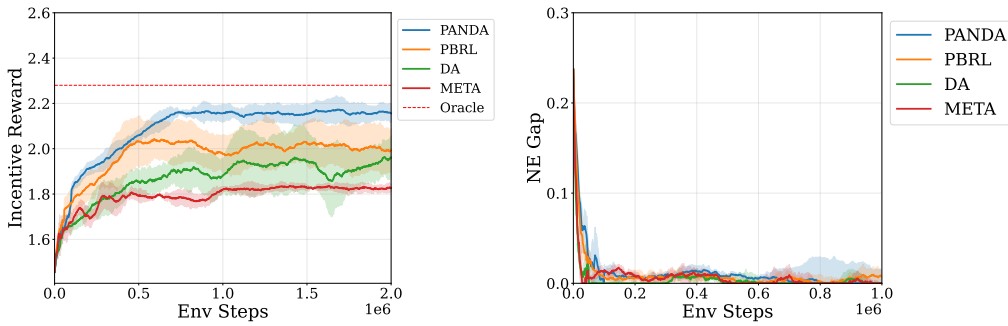

*Figure 1.* Synthetic problem results, averaged over three random seeds. Left: UL incentive reward vs. environment sample steps. Right: LL NE gap vs. environment sample steps.

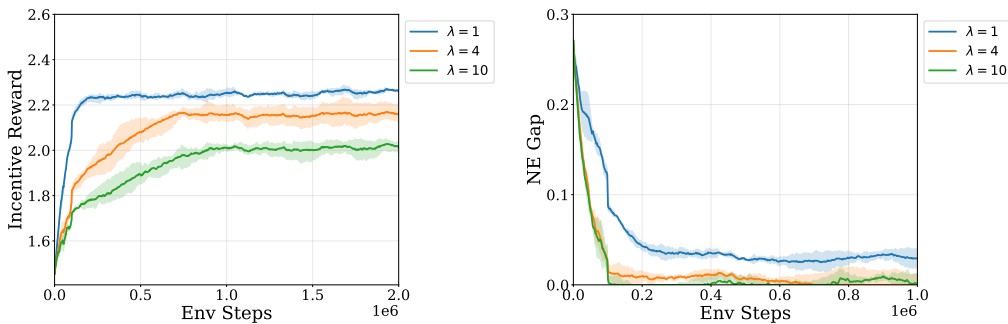

*Figure 2.* Ablation study on penalty parameter $\lambda$. The results are averaged over three random seeds. Left: UL incentive reward vs. environment sample steps. Right: LL NE gap vs. environment sample steps.

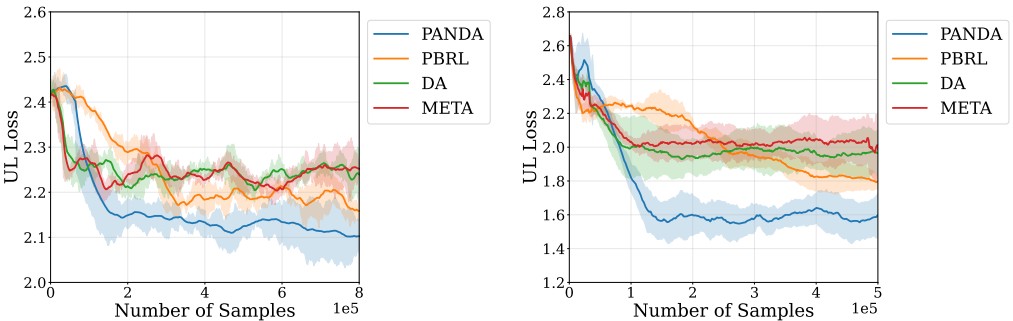

*Figure 3.* Sentinel-Intruder results averaged over three random seeds. UL loss vs. number of sampled trajectories. Left: $5 \times 5$ grid. Right: $20 \times 20$ grid.

encourages two players to cooperate by maximizing the incentive designer's cumulative reward in a fixed MDP $\mathcal{M}_{\mathrm{id}} = \{\mathcal{S}, \mathcal{A}, \mathcal{B}, r_{\mathrm{id}}, \mathcal{P}_{\mathrm{id}}, \gamma\}$, while the LL problem is a regularized MMZSMG $\mathcal{M}(x)$. The overall problem can be written as

$$\max_{x, \phi, \psi} \quad \mathbb{E}_{\pi_\phi, \pi_\psi} \left[ \sum_{t=0}^{T-1} \gamma^t r_{\mathrm{id}}(s_t, a_t, b_t) \Big| \mathcal{M}_{\mathrm{id}} \right]$$
$$\text{s.t.} \quad (\phi, \psi) \in \arg\min_{\phi'} \max_{\psi'} V^{\pi_{\phi'}, \pi_{\psi'}}_{\mathcal{M}(x)}(\rho).$$

In the experiments, we set $|\mathcal{S}| = 5$ and $|\mathcal{A}| = |\mathcal{B}| = 3$, and use tabular softmax policies for both players. The designer reward $r_{\mathrm{id}}$ is uniformly sampled from $[0, 1]$. The base

reward $r_{\mathrm{base}}$ is uniformly sampled from $[0, 1]$, and the incentive reward is $\mathrm{sigmoid}(x(s, a, b))$, where $x \in \mathbb{R}^{|\mathcal{S}| \times |\mathcal{A}| \times |\mathcal{B}|}$ is the incentive parameter to be optimized. The transition dynamics $\mathcal{P}_{\mathrm{id}}$ and $\mathcal{P}$ are randomly generated. Detailed settings are provided in Appendix D.

We compare PANDA with META, DA and PBRL in terms of the UL incentive reward and the LL NE gap, measured by the NI function. In addition, we construct a strong oracle baseline that uses dynamic programming to compute the exact value function and leverages second-order information of the objective functions to obtain exact hypergradients. We regard this oracle as an approximate upper bound on the achievable algorithmic performance. For fairness, we plot

performance against the number of environment sample steps in Figure 1. With the same number of steps, higher incentive reward indicates better UL maximization, while a smaller NE gap indicates that the learned LL policy pair is closer to the NE. The NE gaps of all methods approach zero, suggesting that each can effectively solve the LL MMZSMG. Among them, PANDA attains the highest incentive reward, showing that it more effectively steers the LL equilibrium toward higher designer reward. Moreover, the performance gap between PANDA and the oracle baseline remains small, suggesting that PANDA achieves a solution quality close to that of the oracle.

To examine the effect of the penalty parameter $\lambda$ on PANDA, we also conduct an ablation study by varying $\lambda$ while keeping all other experimental settings unchanged. As shown in Figure 2, when $\lambda = 1$, PANDA achieves a relatively high UL objective value, but the NE gap remains large, indicating that the LL solution is far from equilibrium. In contrast, when $\lambda = 4$ or $\lambda = 10$, the NE gap is close to zero, suggesting that the LL equilibrium constraint is approximately satisfied. Nevertheless, the stronger penalization with $\lambda = 10$ slightly compromises the UL objective. These results indicate that $\lambda$ controls a trade-off between enforcing LL equilibrium accuracy and optimizing the UL objective in practice.

### 4.2. Sentinel-Intruder

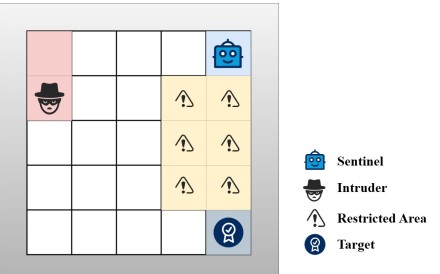

*Figure 4.* Sentinel-Intruder setup. The sentinel aims to capture the intruder before it reaches the target at the bottom-right corner, while avoiding the restricted areas (yellow cells on the right side of the map). The sentinel spawns at the top-right cell, and the intruder spawns uniformly at random within its spawn region (red cells in the top-left corner).

Figure 4 illustrates the Sentinel-Intruder environment, a $5 \times 5$ grid world with two agents: a sentinel and an intruder. The intruder aims to reach the target at the bottom-right corner without being captured, while the sentinel attempts to capture the intruder first. At each step, both agents choose one of five actions (up, down, left, right, stay). An episode terminates when the sentinel captures the intruder, the intruder reaches the target, or the maximum number of steps is reached. If the sentinel captures first, the sentinel (intruder) receives $+10$ ($-10$); if the intruder reaches the target, the intruder (sentinel) receives $+10$ ($-10$).

The map also contains six restricted cells on the right side. Our goal is to discourage the sentinel from entering these cells while still competing against the intruder, which models real-world scenarios where the sentinel must avoid dangerous or privacy-sensitive regions. We formulate this as a BOSMG problem: the UL minimizes the total number of sentinel visits to restricted cells, while the LL computes an NE under a reward parameterized by the UL variable $x$. Specifically, the LL solves $\min_\phi \max_\psi V_{\mathcal{M}(x)}^{\pi_\phi, \pi_\psi}(\rho)$, where $\pi_\phi$ and $\pi_\psi$ denote the intruder and sentinel policies, respectively, and the LL reward is $r_{\text{env}}(s, a, b) + 0.05 \cdot r_x(s, a, b)$. Here, $r_{\text{env}}$ is the terminal environment reward and $r_x$ is an additional learnable reward model with parameter $x$. The UL objective is $\min_x \mathbb{E}_{\tau \sim \pi_\phi^*, \pi_\psi^*}[\mathbb{C}(\tau)]$, where $\mathbb{C}(\tau)$ counts the number of restricted cells visited by the sentinel along trajectory $\tau = \{s_t, a_t, b_t\}_{t=0}^{T-1}$, and $(\pi_\phi^*, \pi_\psi^*)$ is the LL NE.

As shown in Figure 3 (Left), in this more complex and realistic setting, PANDA achieves the lowest UL loss under the same number of sampled trajectories in the environment, indicating that PANDA effectively solves the LL adversarial game and improves the UL objective.

To validate the effectiveness of PANDA in larger-scale environments, we further conduct experiments on a $20 \times 20$ grid map. The results in Figure 3 (Right) show that PANDA continues to perform effectively in this setting and outperforms existing baselines. Full details are deferred to Appendix D.

## 5. Concluding Remarks

In this work, we proposed PANDA, a penalty-based policy-gradient algorithm for BOSMG. To the best of our knowledge, PANDA is the first stochastic first-order method with convergence guarantees for finding an $\epsilon$-stationary point of the original BOSMG formulation, achieving $\tilde{\mathcal{O}}(\epsilon^{-1})$ iteration complexity and $\tilde{\mathcal{O}}(\epsilon^{-3})$ sample complexity. Notably, these rates match the state-of-the-art rates previously known only for BRL with single-policy LL optimization, despite the additional challenges posed by MMZSMGs. Extensive experiments further demonstrate that PANDA consistently outperforms competitive baselines.

Our current work focuses on the setting where the LL problem is a regularized MMZSMG. Extending the proposed framework to general min–max games and broader multi-agent settings is a promising direction for future work.

## Acknowledgments

The work of Zihao Zheng and Songtao Lu is supported in part by project #MMT-8115077 of the Shun Hing Institute of Advanced Engineering, The Chinese University of Hong Kong (CUHK), and in part by the CUHK Direct Grant (Project No. 4055259). The work of Irwin King is supported in part by the General Research Fund (Project No. RGC GRF 2151317).

## Impact Statement

This work contributes to the foundations of bilevel optimization with a regularized MMZSMG at the LL. It may have positive impacts on principled algorithm design for hierarchical and multi-agent decision-making problems. Since the paper is primarily theoretical and evaluated in controlled environments, we do not foresee immediate negative societal impacts. Potential risks may arise only through downstream applications of such methods in strategic decision-making systems, where fairness, safety, and robustness should be carefully considered.

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

# A. Preliminaries

## A.1. Notation

Unless otherwise specified, we use the following notations throughout the appendix.

| Notation | Definition |
|---|---|
| $\mathcal{S}$ | finite state space |
| $\mathcal{A}, \mathcal{B}$ | finite action spaces for the min-player and max-player |
| $r_x(s, a, b)$ | reward function parameterized by $x$ |
| $\mathcal{P}(s'\mid s, a, b)$ | transition probability from state $s$ to $s'$ under actions $a, b$ |
| $\gamma$ | discount factor |
| $h(\cdot)$ | regularization function |
| $\pi_\phi$ | policy of the min-player parameterized by $\phi$ |
| $\pi_\psi$ | policy of the max-player parameterized by $\psi$ |
| $\tau_\phi, \tau_\psi$ | regularization coefficients for $\pi_\phi$ and $\pi_\psi$ |
| $V_{\mathcal{M}(x)}^{\pi_\phi,\pi_\psi}(\rho)$ | value function of policies $(\pi_\phi, \pi_\psi)$ in the MMZSMG $\mathcal{M}(x)$ under initial distribution $\rho$ |
| $\mathcal{M}(x)$ | regularized MMZSMG $\{\mathcal{S}, \mathcal{A}, \mathcal{B}, r_x, \mathcal{P}, \gamma, h\}$ parameterized by $x$ |
| $\pi_\phi^*(x), \pi_\psi^*(x)$ | optimal policies of the min-player and max-player in $\mathcal{M}(x)$ |
| $\phi^*(x), \psi^*(x)$ | parameters of the optimal policies $\pi_\phi^*(x), \pi_\psi^*(x)$ |
| $\pi_\phi^*(x, \psi), \pi_\psi^*(x, \phi)$ | best-response policies given $x$ and the opponent's policy |
| $\phi^*(x, \psi), \psi^*(x, \phi)$ | parameters of the best-response policies |
| $\theta$ | concatenation of $\phi$ and $\psi$ |
| $\theta^*(x)$ | concatenation of $\phi^*(x)$ and $\psi^*(x)$ |
| $\theta^*(x, \theta)$ | concatenation of $\phi^*(x, \psi)$ and $\psi^*(x, \phi)$, where $\theta = (\phi, \psi)$ |
| $J(x, \phi, \psi)$ | abbreviated notation of $V_{\mathcal{M}(x)}^{\pi_\phi,\pi_\psi}(\rho)$ |
| $f(x, \phi, \psi)$ | UL objective function |
| $g(x, \phi, \psi)$ | NI function at LL |
| $\tilde{g}(x, \phi, \psi, \tilde{\phi}, \tilde{\psi})$ | NI function estimator $J(x, \phi, \tilde{\psi}) - J(x, \tilde{\phi}, \psi)$ |
| $J_1(x, \phi)$ | best-response value function of the min-player $J_1(x, \phi) \triangleq \max_\psi J(x, \phi, \psi)$ |
| $J_2(x, \psi)$ | best-response value function of the max-player $J_2(x, \psi) \triangleq \min_\phi J(x, \phi, \psi)$ |
| $\mathcal{L}_\lambda(x, \phi, \psi)$ | $f(x, \phi, \psi) + \lambda g(x, \phi, \psi)$ |
| $\tilde{\mathcal{L}}_\lambda(x, \phi, \psi, \tilde{\phi}, \tilde{\psi})$ | $f(x, \phi, \psi) + \lambda \tilde{g}(x, \phi, \psi, \tilde{\phi}, \tilde{\psi})$ |
| $h(x, \phi, \psi)$ | $\frac{1}{\lambda} f(x, \phi, \psi) + g(x, \phi, \psi)$ |
| $\tilde{h}(x, \phi, \psi, \tilde{\phi}, \tilde{\psi})$ | $\frac{1}{\lambda} f(x, \phi, \psi) + \tilde{g}(x, \phi, \psi, \tilde{\phi}, \tilde{\psi})$ |
| $F(x)$ | UL hyper-objective function $F(x) \triangleq f(x, \phi^*(x), \psi^*(x))$ |
| $B, B_J$ | batch sizes for estimating $f(\cdot)$ and $J(\cdot)$ |
| $H$ | length of truncated trajectories for estimating $J(\cdot)$ |
| $J(\cdot; B_J, H), \tilde{g}(\cdot; B_J, H)$ | estimators of $J(\cdot)$ and $\tilde{g}(\cdot)$ with $B_J$ samples and $H$-step truncated trajectories |
| $f(\cdot; B)$ | estimator of $f(\cdot)$ with $B$ samples |
| $\tilde{h}(\cdot; B, B_J, H), \tilde{\mathcal{L}}_\lambda(\cdot; B, B_J, H)$ | estimators of $\tilde{h}(\cdot)$ and $\tilde{\mathcal{L}}_\lambda(\cdot)$ with $B, B_J$ samples and $H$-step truncated trajectories |
| $\phi_\lambda^*(x), \psi_\lambda^*(x)$ | the optimal solution of the penalized problem $\min_{\phi,\psi} \mathcal{L}_\lambda(x, \phi, \psi)$ |
| $\theta_\lambda^*(x)$ | concatenation of $\phi_\lambda^*(x)$ and $\psi_\lambda^*(x)$ |
| $\mathrm{dist}(X, Y)$ | the distance between two sets $X$ and $Y$, defined in Definition 1 |
| $\delta_\rho, \delta_\pi$ | lower bounds of the initial state distribution and policy |
| $B_r$ | bound of the reward function: $\lvert r_x(s, a, b)\rvert \le B_r$ |
| $C_f, C_J, C_g, C_{\tilde{g}}$ | Lipschitz continuity constants of $f, J, g, \tilde{g}$ |
| $C_\pi$ (Lemma C.10) | Lipschitz continuity constant of the best-response parameter set $\phi^*(x, \psi)$ and $\psi^*(x, \phi)$ |
| $L_{f,1}, L_{g,1}, L_{\tilde{g},1}, L_{h,1}$ | Lipschitz smoothness constants of $f, g, \tilde{g}, h$ |
| $L_{J,1}, L_{J_1,1}, L_{J_2,1}$ | Lipschitz smoothness constants of $J, J_1, J_2$ |
| $L_F$ (Lemma C.14) | Lipschitz smoothness constant of $F$ |
| $L_{J,2}, L_{g,2}, L_{f,2}$ | second-order Lipschitz smoothness constants of $J, g, f$ |
| $\mu_J, \mu_g, \mu_h$ | PŁ-constants of $J, g, h$ in $\theta$ |
| $\sigma_f, \sigma_J$ (Lemma C.16) | variance bounds of the stochastic gradient of $f$ and $J$ |
| $\lVert \cdot \rVert$ | if not specified otherwise, we use the Euclidean norm (or the Frobenius norm for matrices) |

## A.2. Mathematical Preliminaries

**Definition 1** (Distance)**.** The distance between two sets $X$ and $Y$ is defined as $\operatorname{dist}(X, Y) = \inf_{x \in X, y \in Y} \|x - y\|$. Moreover, we also define the distance between a point $x$ and a set $Y$ as $\operatorname{dist}(x, Y) = \inf_{y \in Y} \|x - y\|$.

**Definition 2** (Lipschitz continuity)**.** We say that a function $f(x, y)$ is $C_f$-Lipschitz continuous if, for any $x, x', y, y'$, we have $\|f(x, y) - f(x', y')\| \leq C_f(\|x - x'\| + \|y - y'\|)$.

Similarly, we say that $f$ is $L_{f,1}$-smooth if for any $x, x', y, y'$, we have $\|\nabla f(x, y) - \nabla f(x', y')\| \leq L_{f,1}(\|x - x'\| + \|y - y'\|)$.

We say that $f$ is $L_{f,2}$-Lipschitz continuous Hessian if for any $x, x', y, y'$, we have $\|\nabla^2 f(x, y) - \nabla^2 f(x', y')\| \leq L_{f,2}(\|x - x'\| + \|y - y'\|)$.

**Definition 3** (PŁ condition)**.** We say that a function $f(x, y)$ satisfies $\mu$-Polyak–Łojasiewicz (PŁ) condition with respect to $y$ if for any fixed $x$ and any $y$, we have $\|\nabla_y f(x, y)\|^2 \geq 2\mu(f(x, y) - \min_{y'} f(x, y'))$.

**Lemma A.1** (Theorem 2 of Karimi et al. (2016))**.** *If $f(x, y)$ is $L_{f,1}$-Lipschitz smooth and $\mu$-PŁ in $y$, then it satisfies the error bound (EB) condition with $\mu$, i.e.,*

$$\|\nabla_y f(x, y)\| \geq \mu \operatorname{dist}(y, y^*(x)). \tag{22}$$

*Moreover, it also satisfies the quadratic growth (QG) condition with $\mu$, i.e.,*

$$f(x, y) - \min_{y'} f(x, y') \geq \frac{\mu}{2} \operatorname{dist}^2(y, y^*(x)). \tag{23}$$

**Lemma A.2** (Generalized Danskin's Theorem (Shen et al., 2025; Clarke, 1975))**.** *Let $\mathcal{F}$ be a compact set and let a continuous function $\ell : \mathbb{R}^d \times \mathcal{F} \to \mathbb{R}$ satisfy: 1) $\nabla_x \ell(x, y)$ is continuous in $(x, y)$; and 2) for any $x$ and for any $y, y' \in \arg\max_{y \in \mathcal{F}} \ell(x, y)$, $\nabla_x \ell(x, y) = \nabla_x \ell(x, y')$. Define $h(x) \triangleq \max_{y \in \mathcal{F}} \ell(x, y)$. Then we have $\nabla h(x) = \nabla_x \ell(x, y^*)$ for any $y^* \in \arg\max_{y \in \mathcal{F}} \ell(x, y)$.*

# B. Proofs in Section 2

## B.1. Proofs of Proposition 1

Consider the regularizers $(h_s)_{s \in \mathcal{S}}$, where $h_s(y_s, z_s) : \Delta_{|\mathcal{A}|} \times \Delta_{|\mathcal{B}|} \to \mathbb{R}$. Here, for notational simplicity in the proof, we use $y_s$ and $z_s$ to denote the mixed strategies of the min-player and max-player at state $s$, respectively, and we use $|\mathcal{A}|$ and $|\mathcal{B}|$ to denote the dimensions of the action spaces. For each state $s$, $h_s(y_s, z_s)$ is defined as a continuous function that is strongly convex in $y_s$ and strongly concave in $z_s$.

We first define $H_s : \Delta_{|\mathcal{A}|} \times \Delta_{|\mathcal{B}|} \mapsto \mathbb{R}$ as follows:

$$H_s(y_s, z_s) = y_s^T Q_s z_s + h_s(y_s, z_s),$$

where $Q_s \in \mathbb{R}^{|\mathcal{A}| \times |\mathcal{B}|}$ is a given matrix. Later, we will see that $Q_s$ is related to the Q-function of the Markov game.

It is easy to verify that $H_s(y_s, z_s)$ is also strongly convex in $y_s$ and strongly concave in $z_s$. Consider two optimization problems with $H_s(y_s, z_s)$ as the objective function:

$$\min_{y_s \in \Delta_{|\mathcal{A}|}} \max_{z_s \in \Delta_{|\mathcal{B}|}} H_s(y_s, z_s),$$

and

$$\max_{z_s \in \Delta_{|\mathcal{B}|}} \min_{y_s \in \Delta_{|\mathcal{A}|}} H_s(y_s, z_s).$$

By the strong convexity-concavity of $H_s(y_s, z_s)$, we know that there exists a unique $(y_s^*, z_s^*) \in \Delta_{|\mathcal{A}|} \times \Delta_{|\mathcal{B}|}$ such that

$$H_s(y_s^*, z_s^*) = \min_{y_s \in \Delta_{|\mathcal{A}|}} \max_{z_s \in \Delta_{|\mathcal{B}|}} H_s(y_s, z_s) = \max_{z_s \in \Delta_{|\mathcal{B}|}} \min_{y_s \in \Delta_{|\mathcal{A}|}} H_s(y_s, z_s),$$

which is the unique saddle point of $H_s(y_s, z_s)$ (Rockafellar, 1997; Ekeland & Témam, 1999).

We next consider the envelope function $h_s^* : \mathbb{R}^{|\mathcal{A}| \times |\mathcal{B}|} \to \mathbb{R}$, defined by $h_s^*(Q_s) \triangleq \min_{y_s \in \Delta_{|\mathcal{A}|}} \max_{z_s \in \Delta_{|\mathcal{B}|}} \{y_s^T Q_s z_s + h_s(y_s, z_s)\}$. The following results establish several basic properties of $h_s^*$.

**Lemma B.1.** *For each state $s \in \mathcal{S}$, the envelope function $h_s^*$ has the following properties:*

i. *(Uniqueness of the solution) For any $Q_s \in \mathbb{R}^{|\mathcal{A}| \times |\mathcal{B}|}$, there exists a unique $(y_s^*, z_s^*) \in \Delta_{|\mathcal{A}|} \times \Delta_{|\mathcal{B}|}$ such that $h_s^*(Q_s) = (y_s^*)^T Q_s z_s^* + h_s(y_s^*, z_s^*) = \min_{y_s \in \Delta_{|\mathcal{A}|}} \max_{z_s \in \Delta_{|\mathcal{B}|}} \{y_s^T Q_s z_s + h_s(y_s, z_s)\} = \max_{z_s \in \Delta_{|\mathcal{B}|}} \min_{y_s \in \Delta_{|\mathcal{A}|}} \{y_s^T Q_s z_s + h_s(y_s, z_s)\}.$*

ii. *(Monotonicity) For any $Q_s, Q_s' \in \mathbb{R}^{|\mathcal{A}| \times |\mathcal{B}|}$ such that $Q_s \leq Q_s'$ (element-wise), we have $h_s^*(Q_s) \leq h_s^*(Q_s')$.*

*Proof.* The first property follows from the strong convexity-concavity of $y_s^T Q_s z_s + h_s(y_s, z_s)$ for any $Q_s$.

Now we prove the monotonicity. For any $Q_s, Q_s' \in \mathbb{R}^{|\mathcal{A}| \times |\mathcal{B}|}$ such that $Q_s \leq Q_s'$, since $y_s \in \Delta_{|\mathcal{A}|}$ and $z_s \in \Delta_{|\mathcal{B}|}$ are probability distributions, we have for any $y_s$ and $z_s$,

$$y_s^T Q_s z_s \leq y_s^T Q_s' z_s.$$

Fix $y_s$, for any $z_s \in \Delta_{|\mathcal{B}|}$, we have

$$
\begin{aligned}
& y_s^T Q_s z_s \leq y_s^T Q_s' z_s, \quad \forall z_s \in \Delta_{|\mathcal{B}|} \\
\Longrightarrow{} & y_s^T Q_s z_s + h_s(y_s, z_s) \leq y_s^T Q_s' z_s + h_s(y_s, z_s), \quad \forall z_s \in \Delta_{|\mathcal{B}|} \\
\Longrightarrow{} & \max_{z_s \in \Delta_{|\mathcal{B}|}} \{y_s^T Q_s z_s + h_s(y_s, z_s)\} \leq \max_{z_s \in \Delta_{|\mathcal{B}|}} \{y_s^T Q_s' z_s + h_s(y_s, z_s)\},
\end{aligned}
\tag{24}
$$

which implies

$$
\begin{aligned}
h_s^*(Q_s) &= \min_{y_s \in \Delta_{|\mathcal{A}|}} \max_{z_s \in \Delta_{|\mathcal{B}|}} \{y_s^T Q_s z_s + h_s(y_s, z_s)\} \\
&\leq \min_{y_s \in \Delta_{|\mathcal{A}|}} \max_{z_s \in \Delta_{|\mathcal{B}|}} \{y_s^T Q_s' z_s + h_s(y_s, z_s)\} = h_s^*(Q_s').
\end{aligned}
\tag{25}
$$

The monotonicity of $h_s^*$ is proved. $\qquad\square$

Now we consider the regularized MMZSMG $\mathcal{M}(x)$ defined in Section 2. Inspired by the proof strategy in Geist et al. (2019), we analyze the properties of MMZSMGs by defining the corresponding Bellman operators. We then establish the contraction property of these Bellman operators to prove the existence and uniqueness of the equilibrium in regularized MMZSMGs.

Since we only focus on the proof of the uniqueness of the equilibrium at LL, for simplicity, we may omit the UL variable $x$ in the notations, i.e., $x$ is fixed in the following proof if not specified otherwise.

**Definition 4** (Min–max Bellman Operator). For any $V \in \mathbb{R}^{|\mathcal{S}|}$, define $Q \in \mathbb{R}^{|\mathcal{S}| \times |\mathcal{A}| \times |\mathcal{B}|}$ as $Q_{sab} = r_x(s, a, b) + \gamma \mathbb{E}_{s' \sim \mathcal{P}(\cdot|s,a,b)} V(s')$, where $V(s')$ is the entry of $V$ corresponding to state $s'$. Let $y \in \Delta_{|\mathcal{A}|}^{|\mathcal{S}|}$ and $z \in \Delta_{|\mathcal{B}|}^{|\mathcal{S}|}$ be the policies of the two players respectively.

The min–max Bellman operator for a given policy pair $(y, z)$ is defined as $T_{y,z} : V \in \mathbb{R}^{|\mathcal{S}|} \mapsto T_{y,z}V \in \mathbb{R}^{|\mathcal{S}|}$, where for each state $s \in \mathcal{S}$, $(T_{y,z}V)(s) \triangleq y_s^T Q_s z_s$.

The soft min–max Bellman operator for a given policy pair $(y, z)$ is defined as $T_{y,z,h} : V \in \mathbb{R}^{|\mathcal{S}|} \mapsto T_{y,z,h}V \in \mathbb{R}^{|\mathcal{S}|}$, where for each state $s \in \mathcal{S}$, $(T_{y,z,h}V)(s) \triangleq (T_{y,z}V)(s) + h_s(y_s, z_s)$.

Also define the min–max Bellman optimality operator $T_* : V \in \mathbb{R}^{|\mathcal{S}|} \mapsto T_*V \in \mathbb{R}^{|\mathcal{S}|}$ state-wise as follows: for each state $s \in \mathcal{S}$, $(T_*V)(s) \triangleq \min_{y_s \in \Delta_{|\mathcal{A}|}} \max_{z_s \in \Delta_{|\mathcal{B}|}} (T_{y_s,z_s}V)(s)$. For simplicity, we can write $T_*$ as $T_*V = \min_{y \in \Delta_{|\mathcal{A}|}^{|\mathcal{S}|}} \max_{z \in \Delta_{|\mathcal{B}|}^{|\mathcal{S}|}} T_{y,z}V$, where the min–max is taken over all state-wise policies.

The soft min–max Bellman optimality operator $T_{*,h} : V \in \mathbb{R}^{|\mathcal{S}|} \mapsto T_{*,h}V \in \mathbb{R}^{|\mathcal{S}|}$ is similarly defined as follows: for each state $s \in \mathcal{S}$, $(T_{*,h}V)(s) \triangleq \min_{y_s \in \Delta_{|\mathcal{A}|}} \max_{z_s \in \Delta_{|\mathcal{B}|}} (T_{y_s,z_s,h}V)(s)$. For simplicity, we can write $T_{*,h}$ as $T_{*,h}V = \min_{y \in \Delta_{|\mathcal{A}|}^{|\mathcal{S}|}} \max_{z \in \Delta_{|\mathcal{B}|}^{|\mathcal{S}|}} T_{y,z,h}V$, where the min–max is taken over all state-wise policies.

For each state $s \in \mathcal{S}$, consider the min–max problem $\min_{y_s \in \Delta_{|\mathcal{A}|}} \max_{z_s \in \Delta_{|\mathcal{B}|}} \{y_s^T Q_s z_s + h_s(y_s, z_s)\}$, by Lemma B.1, we know that there exists a unique optimal policy pair $(y_s^*, z_s^*)$ for this problem. Therefore, for any $V \in \mathbb{R}^{|\mathcal{S}|}$, there exists a

unique policy pair $(y^*, z^*)$ such that $T_{*,h}V = T_{y^*,z^*,h}V$, which means that the soft min–max Bellman optimality operator $T_{*,h}$ can be viewed as a special case of the soft min–max Bellman operator $T_{y,z,h}$ with the policy pair $(y^*, z^*)$.

Then we will show some properties of the min–max Bellman operators.

**Lemma B.2** (Properties of min–max Bellman Operators). *The min–max Bellman operators $T_{y,z}$ for any given policy pair $(y, z)$ and the optimality operator $T_*$ have the following properties:*

  i. *(Monotonicity) For any $V, V' \in \mathbb{R}^{|\mathcal{S}|}$ such that $V \leq V'$, we have $T_{y,z}V \leq T_{y,z}V'$ and $T_*V \leq T_*V'$. Here, the inequality is element-wise.*

  ii. *(Contraction) For any $V, V' \in \mathbb{R}^{|\mathcal{S}|}$, we have $\|T_{y,z}V - T_{y,z}V'\|_\infty \leq \gamma\|V - V'\|_\infty$ and $\|T_*V - T_*V'\|_\infty \leq \gamma\|V - V'\|_\infty$.*

  iii. *(Distributivity) For any constant $c \in \mathbb{R}$, we have $T_{y,z}(V + c\mathbf{1}) = T_{y,z}V + \gamma c\mathbf{1}$ and $T_*(V + c\mathbf{1}) = T_*V + \gamma c\mathbf{1}$, where $\mathbf{1} \in \mathbb{R}^{|\mathcal{S}|}$ is a vector with all elements equal to 1.*

*Proof.* We can write $T_{y,z}V$ as

$$T_{y,z}V = r_{y,z} + \gamma\mathcal{P}_{y,z}V,$$

where $r_{y,z} \in \mathbb{R}^{|\mathcal{S}|}$ is defined as $r_{y,z}(s) = \sum_{a\in\mathcal{A}}\sum_{b\in\mathcal{B}} y_s(a)z_s(b)r(s, a, b)$ and $\mathcal{P}_{y,z} \in \mathbb{R}^{|\mathcal{S}|\times|\mathcal{S}|}$ is defined as $\mathcal{P}_{y,z}(s, s') = \sum_{a\in\mathcal{A}}\sum_{b\in\mathcal{B}} y_s(a)z_s(b)\mathcal{P}(s'|s, a, b)$. Now for any $V, V' \in \mathbb{R}^{|\mathcal{S}|}$ such that $V \leq V'$, we have

$$T_{y,z}V - T_{y,z}V' = \gamma\mathcal{P}_{y,z}(V - V') \leq 0, \tag{26}$$

which proves the monotonicity of $T_{y,z}$. Similarly, by the definition of $T_*$ in Definition 4 we have

$$
\begin{aligned}
V \leq V' &\implies \mathcal{P}_{y,z}V \leq \mathcal{P}_{y,z}V', \\
&\implies Q_s \leq Q'_s, \quad \forall s \in \mathcal{S} \\
&\implies y_s^T Q_s z_s \leq y_s^T Q'_s z_s, \quad \forall y_s \in \Delta_{|\mathcal{A}|}, z_s \in \Delta_{|\mathcal{B}|}, \forall s \in \mathcal{S} \\
&\implies \max_{z_s} y_s^T Q_s z_s \leq \max_{z_s} y_s^T Q'_s z_s, \quad \forall y_s \in \Delta_{|\mathcal{A}|}, \forall s \in \mathcal{S} \\
&\implies \min_{y_s}\max_{z_s} y_s^T Q_s z_s \leq \min_{y_s}\max_{z_s} y_s^T Q'_s z_s, \quad \forall s \in \mathcal{S} \\
&\iff T_*V \leq T_*V', 
\end{aligned}
\tag{27}
$$

where the first implication follows from the fact that $\mathcal{P}_{y,z}$ is a stochastic matrix, the third implication follows from the fact that $y_s$ and $z_s$ are probability distributions.

Next we prove the contraction property. For any $V, V' \in \mathbb{R}^{|\mathcal{S}|}$, we have

$$
\begin{aligned}
\|T_{y,z}V - T_{y,z}V'\|_\infty &= \|\gamma\mathcal{P}_{y,z}(V - V')\|_\infty \\
&\leq \gamma\|V - V'\|_\infty.
\end{aligned}
\tag{28}
$$

Since this holds for any given policy pair $(y, z)$, we know that

$$\max_{y,z}\|T_{y,z}V - T_{y,z}V'\|_\infty \leq \gamma\|V - V'\|_\infty. \tag{29}$$

Then we have for any $V, V' \in \mathbb{R}^{|\mathcal{S}|}$,

$$
\begin{aligned}
\|T_*V - T_*V'\|_\infty &= \|\min_y\max_z T_{y,z}V - \min_y\max_z T_{y,z}V'\|_\infty \\
&\leq \max_{y,z,s}|(T_{y,z}V)(s) - (T_{y,z}V')(s)| \\
&= \max_{y,z}\|T_{y,z}V - T_{y,z}V'\|_\infty \\
&\leq \gamma\|V - V'\|_\infty.
\end{aligned}
\tag{30}
$$

Now we prove the distributivity property. The distributivity of $T_{y,z}$ is straightforward:

$$
\begin{aligned}
T_{y,z}(V + c\mathbf{1}) &= r_{y,z} + \gamma \mathcal{P}_{y,z}(V + c\mathbf{1}) \\
&= r_{y,z} + \gamma \mathcal{P}_{y,z}V + \gamma c \mathcal{P}_{y,z}\mathbf{1} \\
&= T_{y,z}V + \gamma c\mathbf{1}.
\end{aligned}
\tag{31}
$$

Similarly, we have

$$
\begin{aligned}
T_*(V + c\mathbf{1}) &= \min_y \max_z T_{y,z}(V + c\mathbf{1}) \\
&= \min_y \max_z \{T_{y,z}V + \gamma c\mathbf{1}\} \\
&= \min_y \max_z T_{y,z}V + \gamma c\mathbf{1} \\
&= T_*V + \gamma c\mathbf{1}.
\end{aligned}
\tag{32}
$$

$\square$

Now we will also show some properties of the soft min–max Bellman operators.

**Lemma B.3** (Properties of Soft Min–Max Bellman Operators)**.** *The soft min–max Bellman operators $T_{y,z,h}$ for any given $(y, z)$ and the soft optimality operator $T_{*,h}$ have the following properties:*

   i. *(Monotonicity) For any $V, V' \in \mathbb{R}^{|\mathcal{S}|}$ such that $V \leq V'$ (element-wise), we have $T_{y,z,h}V \leq T_{y,z,h}V'$ and $T_{*,h}V \leq T_{*,h}V'$.*

   ii. *(Contraction) For any $V, V' \in \mathbb{R}^{|\mathcal{S}|}$, we have $\|T_{y,z,h}V - T_{y,z,h}V'\|_\infty \leq \gamma\|V - V'\|_\infty$ and $\|T_{*,h}V - T_{*,h}V'\|_\infty \leq \gamma\|V - V'\|_\infty$, where $\|\cdot\|_\infty$ takes the maximum absolute value among all elements.*

   iii. *(Distributivity) For any constant $c \in \mathbb{R}$, we have $T_{y,z,h}(V + c\mathbf{1}) = T_{y,z,h}V + \gamma c\mathbf{1}$ and $T_{*,h}(V + c\mathbf{1}) = T_{*,h}V + \gamma c\mathbf{1}$, where $\mathbf{1} \in \mathbb{R}^{|\mathcal{S}|}$ is a vector with all elements equal to 1.*

*Proof.* For any $V, V' \in \mathbb{R}^{|\mathcal{S}|}$ such that $V \leq V'$, by the monotonicity of $T_{y,z}$ in Lemma B.2, we obtain

$$
\begin{aligned}
T_{y,z,h}V - T_{y,z,h}V' &= T_{y,z}V + h(y, z) - (T_{y,z}V' + h(y, z)) \\
&= T_{y,z}V - T_{y,z}V' \leq 0,
\end{aligned}
\tag{33}
$$

where $h(y, z) = (h_s(y, z))_{s \in \mathcal{S}} \in \mathbb{R}^{|\mathcal{S}|}$ is the vector of regularization terms for all states. This proves the monotonicity of $T_{y,z,h}$.

Similarly, since $T_{*,h}V = \min_{y \in \Delta_{|\mathcal{A}|}^{|\mathcal{S}|}} \max_{z \in \Delta_{|\mathcal{B}|}^{|\mathcal{S}|}} T_{y,z,h}V$, we can obtain

$$
\begin{aligned}
V \leq V' &\implies Q_s \leq Q'_s, \quad \forall s \in \mathcal{S} \\
&\implies y_s^T Q_s z_s + h_s(y_s, z_s) \leq y_s^T Q'_s z_s + h_s(y_s, z_s), \quad \forall y_s \in \Delta_{|\mathcal{A}|}, z_s \in \Delta_{|\mathcal{B}|}, \forall s \in \mathcal{S} \\
&\implies \max_{z_s}\{y_s^T Q_s z_s + h_s(y_s, z_s)\} \leq \max_{z_s}\{y_s^T Q'_s z_s + h_s(y_s, z_s)\}, \quad \forall y_s \in \Delta_{|\mathcal{A}|}, \forall s \in \mathcal{S} \\
&\implies \min_{y_s} \max_{z_s}\{y_s^T Q_s z_s + h_s(y_s, z_s)\} \leq \min_{y_s} \max_{z_s}\{y_s^T Q'_s z_s + h_s(y_s, z_s)\}, \quad \forall s \in \mathcal{S} \\
&\iff T_{*,h}V \leq T_{*,h}V'.
\end{aligned}
\tag{34}
$$

This proves the monotonicity of $T_{*,h}$. Next we prove the contraction property. For any $V, V' \in \mathbb{R}^{|\mathcal{S}|}$, we have

$$
\|T_{y,z,h}V - T_{y,z,h}V'\|_\infty = \|T_{y,z}V - T_{y,z}V'\|_\infty \leq \gamma\|V - V'\|_\infty.
\tag{35}
$$

Similarly, we can get

$$
\begin{aligned}
\|T_{*,h}V - T_{*,h}V'\|_\infty &= \|\min_y \max_z T_{y,z,h}V - \min_y \max_z T_{y,z,h}V'\|_\infty \\
&\leq \max_{y,z,s} |(T_{y,z,h}V)(s) - (T_{y,z,h}V')(s)| \\
&= \max_{y,z} \|T_{y,z,h}V - T_{y,z,h}V'\|_\infty \\
&\leq \gamma\|V - V'\|_\infty.
\end{aligned}
\tag{36}
$$

Thus, $T_{y,z,h}$ and $T_{*,h}$ are both $\gamma$-contractions.

Now we prove the distributivity property. The distributivity of $T_{y,z,h}$ is straightforward:

$$
\begin{aligned}
T_{y,z,h}(V + c\mathbf{1}) &= T_{y,z}(V + c\mathbf{1}) + h(y,z) \\
&= T_{y,z}V + \gamma c\mathbf{1} + h(y,z) \\
&= T_{y,z,h}V + \gamma c\mathbf{1}.
\end{aligned}
\tag{37}
$$

Similarly, we can obtain the distributivity of $T_{*,h}$ as follows:

$$
\begin{aligned}
T_{*,h}(V + c\mathbf{1}) &= \min_y \max_z T_{y,z,h}(V + c\mathbf{1}) \\
&= \min_y \max_z \{T_{y,z,h}V + \gamma c\mathbf{1}\} \\
&= \min_y \max_z T_{y,z,h}V + \gamma c\mathbf{1} \\
&= T_{*,h}V + \gamma c\mathbf{1}.
\end{aligned}
\tag{38}
$$

$\square$

By the contraction property in Lemma B.3, we know that the Bellman operators defined in Definition 4 all have unique fixed points.

**Definition 5** (On-Policy and Optimal Value Functions)**.** We have the following definitions:

1. For any policy pair $(y,z) \in \Delta_{|\mathcal{A}|}^{|\mathcal{S}|} \times \Delta_{|\mathcal{B}|}^{|\mathcal{S}|}$, $V_h^{y,z} \in \mathbb{R}^{|\mathcal{S}|}$ is defined as the unique fixed point of the soft min–max Bellman operator $T_{y,z,h}$, i.e., $V_h^{y,z} = T_{y,z,h}V_h^{y,z}$.

2. $V_h^* \in \mathbb{R}^{|\mathcal{S}|}$ is defined as the unique fixed point of the soft min–max Bellman optimality operator $T_{*,h}$, i.e., $V_h^* = T_{*,h}V_h^*$.

Then we can prove the following lemma.

**Lemma B.4** (Unique Equilibrium)**.** *There exists a unique* $(y_h^*, z_h^*) \in \Delta_{|\mathcal{A}|}^{|\mathcal{S}|} \times \Delta_{|\mathcal{B}|}^{|\mathcal{S}|}$ *such that* $V_h^{y_h^*, z_h^*} = V_h^*$, *and for any* $y \in \Delta_{|\mathcal{A}|}^{|\mathcal{S}|}$ *and* $z \in \Delta_{|\mathcal{B}|}^{|\mathcal{S}|}$,

$$
V_h^{y_h^*, z} \leq V_h^* \leq V_h^{y, z_h^*},
$$

*where the inequalities are element-wise.*

*Proof.* From Definition 5, we know that $V_h^*$ is the unique fixed point of $T_{*,h}$, i.e.,

$$
V_h^* = T_{*,h}V_h^*.
$$

By Definition 4 of $T_{*,h}$ and Lemma B.1, for the fixed value vector $V_h^*$, there exists a unique policy pair $(y_h^*, z_h^*)$ such that

$$
T_{*,h}V_h^* = T_{y_h^*, z_h^*, h}V_h^*.
$$

Therefore,

$$V_h^* = T_{*,h} V_h^* = T_{y_h^*, z_h^*, h} V_h^*. \tag{39}$$

Hence $V_h^*$ is a fixed point of $T_{y_h^*, z_h^*, h}$. Since $T_{y_h^*, z_h^*, h}$ is a contraction, its fixed point is unique, and is defined as $V_h^{y_h^*, z_h^*}$. Thus, we obtain

$$V_h^{y_h^*, z_h^*} = V_h^*. \tag{40}$$

Next, we prove the saddle inequalities. For any $z \in \Delta_{|\mathcal{B}|}^{|\mathcal{S}|}$, since $(y_h^*, z_h^*)$ is the unique saddle point of the statewise regularized problem $\min_y \max_z T_{y,z,h} V_h^*$, we have, for every state $s \in \mathcal{S}$,

$$\left( T_{y_h^*, z, h} V_h^* \right)(s) \leq \left( T_{y_h^*, z_h^*, h} V_h^* \right)(s).$$

Thus, element-wise,

$$T_{y_h^*, z, h} V_h^* \leq T_{y_h^*, z_h^*, h} V_h^* = V_h^*.$$

Since $T_{y_h^*, z, h}$ is monotone by Lemma B.3, applying it repeatedly gives

$$T_{y_h^*, z, h}^{(2)} V_h^* \leq T_{y_h^*, z, h} V_h^* \leq V_h^*. \tag{41}$$

By induction, for any $k \geq 1$,

$$T_{y_h^*, z, h}^{(k)} V_h^* \leq V_h^*.$$

Taking $k \to \infty$ and using the contraction property of $T_{y_h^*, z, h}$, we obtain

$$V_h^{y_h^*, z} = \lim_{k \to \infty} T_{y_h^*, z, h}^{(k)} V_h^* \leq V_h^*. \tag{42}$$

This proves the left inequality.

Similarly, for any $y \in \Delta_{|\mathcal{A}|}^{|\mathcal{S}|}$, since $(y_h^*, z_h^*)$ is the statewise saddle point, we have

$$\left( T_{y_h^*, z_h^*, h} V_h^* \right)(s) \leq \left( T_{y, z_h^*, h} V_h^* \right)(s), \qquad \forall s \in \mathcal{S}.$$

Hence,

$$V_h^* = T_{y_h^*, z_h^*, h} V_h^* \leq T_{y, z_h^*, h} V_h^*.$$

By monotonicity of $T_{y, z_h^*, h}$,

$$V_h^* \leq T_{y, z_h^*, h} V_h^* \leq T_{y, z_h^*, h}^{(2)} V_h^* \leq \cdots \leq T_{y, z_h^*, h}^{(k)} V_h^*.$$

Taking $k \to \infty$ and using the contraction property of $T_{y, z_h^*, h}$ gives

$$V_h^* \leq \lim_{k \to \infty} T_{y, z_h^*, h}^{(k)} V_h^* = V_h^{y, z_h^*}. \tag{43}$$

Therefore,

$$V_h^{y_h^*, z} \leq V_h^* = V_h^{y_h^*, z_h^*} \leq V_h^{y, z_h^*}. \tag{44}$$

Finally, uniqueness follows from the uniqueness of $V_h^*$. Given this fixed point $V_h^*$, the state-wise regularized min–max problem admits a unique saddle point at each state due to the strong convexity–concavity of the regularized objective. Therefore, the policy pair $(y_h^*, z_h^*)$ is unique. $\qquad \square$

Based on these results, we can prove Proposition 1 about the existence and uniqueness of the equilibrium in the regularized two-player zero-sum Markov game.

**Proposition B.1** (Proposition 1 in Section 2.3). *For any given $x$, there exists a unique $(\pi_\phi^*(x), \pi_\psi^*(x))$ such that for any $\phi$ and $\psi$,*

$$V_{\mathcal{M}(x)}^{\pi_\phi^*(x), \pi_\psi}(\rho) \leq V_{\mathcal{M}(x)}^{\pi_\phi^*(x), \pi_\psi^*(x)}(\rho) \leq V_{\mathcal{M}(x)}^{\pi_\phi, \pi_\psi^*(x)}(\rho).$$

*Moreover, we have*

$$V_{\mathcal{M}(x)}^{\pi_\phi^*(x), \pi_\psi^*(x)}(\rho) = \min_\phi \max_\psi V_{\mathcal{M}(x)}^{\pi_\phi, \pi_\psi}(\rho) = \max_\psi \min_\phi V_{\mathcal{M}(x)}^{\pi_\phi, \pi_\psi}(\rho).$$

*Proof.* By the definition of the joint state-value function $V_{\mathcal{M}(x)}^{\pi_\phi, \pi_\psi}(s)$ in (2), we have

$$V_{\mathcal{M}(x)}^{\pi_\phi, \pi_\psi}(s) = \mathbb{E}_{\pi_\phi, \pi_\psi, \mathcal{P}} \left[ \sum_{t=0}^\infty \gamma^t \left( r_x(s_t, a_t, b_t) + h_{s_t}(\pi_\phi(\cdot|s_t), \pi_\psi(\cdot|s_t)) \right) \Big| s_0 = s \right]$$

$$= \pi_\phi(\cdot|s)^T \left( r(x, s, \cdot, \cdot) + \gamma \sum_{s'} \mathcal{P}(s'|s, \cdot, \cdot) V_{\mathcal{M}(x)}^{\pi_\phi, \pi_\psi}(s') \right) \pi_\psi(\cdot|s) + h_s(\pi_\phi(\cdot|s), \pi_\psi(\cdot|s)).$$

By the definition of $T_{y, z, h}$ in Definition 4, for any policy pair $(\pi_\phi, \pi_\psi)$, we have

$$V_{\mathcal{M}(x)}^{\pi_\phi, \pi_\psi}(s) = (T_{\pi_\phi, \pi_\psi, h} V_{\mathcal{M}(x)}^{\pi_\phi, \pi_\psi})(s), \tag{45}$$

which implies that $V_{\mathcal{M}(x)}^{\pi_\phi, \pi_\psi}$ is the unique fixed point of $T_{\pi_\phi, \pi_\psi, h}$, i.e., the on-policy soft value function $V_{\mathcal{M}(x)}^{\pi_\phi, \pi_\psi}$ is the fixed point $V_h^{\pi_\phi, \pi_\psi}$ defined in Definition 5.

Then, by Lemma B.4, there exists a unique policy pair $(\pi_\phi^*(x), \pi_\psi^*(x))$ such that for any state $s$,

$$V_{\mathcal{M}(x)}^{\pi_\phi^*(x), \pi_\psi}(s) = V_h^{\pi_\phi^*(x), \pi_\psi}(s) \leq V_h^*(s) \leq V_h^{\pi_\phi, \pi_\psi^*(x)}(s) = V_{\mathcal{M}(x)}^{\pi_\phi, \pi_\psi^*(x)}(s), \tag{46}$$

and $V_h^*(s) = V_h^{\pi_\phi^*(x), \pi_\psi^*(x)}(s) = V_{\mathcal{M}(x)}^{\pi_\phi^*(x), \pi_\psi^*(x)}(s)$.

Since this holds for every state $s$, for the given initial state distribution $\rho$ with $\min_s \rho(s) > 0$, we have

$$V_{\mathcal{M}(x)}^{\pi_\phi^*(x), \pi_\psi}(\rho) \leq V_{\mathcal{M}(x)}^{\pi_\phi^*(x), \pi_\psi^*(x)}(\rho) \leq V_{\mathcal{M}(x)}^{\pi_\phi, \pi_\psi^*(x)}(\rho). \tag{47}$$

Therefore, $(\pi_\phi^*(x), \pi_\psi^*(x))$ is the unique saddle point of $\min_{\pi_\phi} \max_{\pi_\psi} V_{\mathcal{M}(x)}^{\pi_\phi, \pi_\psi}(\rho)$. Hence,

$$V_{\mathcal{M}(x)}^{\pi_\phi^*(x), \pi_\psi^*(x)}(\rho) = \min_{\pi_\phi} \max_{\pi_\psi} V_{\mathcal{M}(x)}^{\pi_\phi, \pi_\psi}(\rho) = \max_{\pi_\psi} \min_{\pi_\phi} V_{\mathcal{M}(x)}^{\pi_\phi, \pi_\psi}(\rho). \tag{48}$$

Since $\phi$ and $\psi$ are the parameters of the policies $\pi_\phi$ and $\pi_\psi$ respectively, and the parameterization is expressive enough to represent all mixed policies in the policy class, this further implies that

$$V_{\mathcal{M}(x)}^{\pi_\phi^*(x), \pi_\psi^*(x)}(\rho) = \min_\phi \max_\psi V_{\mathcal{M}(x)}^{\pi_\phi, \pi_\psi}(\rho) = \max_\psi \min_\phi V_{\mathcal{M}(x)}^{\pi_\phi, \pi_\psi}(\rho). \tag{49}$$

$\square$

## B.2. Proofs of Other Technical Lemmas

For the Markov game $\mathcal{M}(x) = \{\mathcal{S}, \mathcal{A}, \mathcal{B}, r_x, \mathcal{P}, \gamma, h\}$ defined in Section 2, we define the state-visitation distribution under policies $\pi_\phi$ and $\pi_\psi$ as

$$d_\rho^{\pi_\phi, \pi_\psi}(s) \triangleq (1 - \gamma) \sum_{t=0}^\infty \gamma^t P(s_t = s | s_0 \sim \rho, \pi_\phi, \pi_\psi, \mathcal{M}(x)).$$

**Lemma B.5** (Gradient of $J(x, \phi, \psi)$ with respect to $x$). *For any $x$, $\phi$ and $\psi$, the gradient of $J(x, \phi, \psi)$ with respect to $x$ is given by*

$$\nabla_x J(x, \phi, \psi) = \frac{1}{1-\gamma} \mathbb{E}_{s \sim d_\rho^{\pi_\phi, \pi_\psi}, a \sim \pi_\phi(\cdot|s), b \sim \pi_\psi(\cdot|s)} \left[ \nabla_x r_x(s, a, b) \right] = \mathbb{E} \left[ \sum_{t=0}^{\infty} \gamma^t \nabla_x r_x(s_t, a_t, b_t) \right],$$

*where the expectation is taken over the trajectory $\{s_t, a_t, b_t\}_{t \geq 0}$ generated by $s_0 \sim \rho$, $a_t \sim \pi_\phi(\cdot|s_t)$, $b_t \sim \pi_\psi(\cdot|s_t)$ and $s_{t+1} \sim P(\cdot|s_t, a_t, b_t)$.*

*Proof.* Since $J(x, \phi, \psi)$ is defined as the value function $V_{\mathcal{M}(x)}^{\pi_\phi, \pi_\psi}(\rho)$ for the given initial distribution $\rho$, we have

$$\begin{aligned}
J(x, \phi, \psi) &= \mathbb{E}_{\substack{s_0 \sim \rho, a_t \sim \pi_\phi(\cdot|s_t) \\ s_{t+1} \sim P^{\pi_\psi}(\cdot|s_t, a_t)}} \left[ \sum_{t=0}^{\infty} \gamma^t \left( r_x(s_t, a_t, b_t) - \tau_\psi \log \pi_\psi(b_t|s_t) + \tau_\phi \log \pi_\phi(a_t|s_t) \right) \right] \\
&= \frac{1}{1-\gamma} \sum_s d_\rho^{\pi_\phi, \pi_\psi}(s) \sum_{a,b} \pi_\phi(a|s) \pi_\psi(b|s) \left( r_x(s, a, b) - \tau_\psi \log \pi_\psi(b|s) + \tau_\phi \log \pi_\phi(a|s) \right)
\end{aligned}$$

Taking gradient with respect to $x$ on both sides, we have

$$\begin{aligned}
\nabla_x J(x, \phi, \psi) &= \frac{1}{1-\gamma} \sum_s d_\rho^{\pi_\phi, \pi_\psi}(s) \sum_{a,b} \pi_\phi(a|s) \pi_\psi(b|s) \nabla_x r_x(s, a, b) \\
&= \mathbb{E}_{\substack{s_0 \sim \rho, a_t \sim \pi_\phi(\cdot|s_t) \\ b_t \sim \pi_\psi(\cdot|s_t) \\ s_{t+1} \sim P(\cdot|s_t, a_t, b_t)}} \left[ \sum_{t=0}^{\infty} \gamma^t \nabla_x r_x(s_t, a_t, b_t) \right],
\end{aligned} \tag{50}$$

which completes the proof. $\qquad\square$

**Lemma B.6** (Policy Gradient of $J(x, \phi, \psi)$). *For any $x$, $\phi$ and $\psi$, the policy gradients of $J(x, \phi, \psi)$ with respect to $\phi$ and $\psi$ are given by*

$$\begin{aligned}
&\nabla_\phi J(x, \phi, \psi) \\
&= \mathbb{E}_{\substack{s_0 \sim \rho, a_t \sim \pi_\phi(\cdot|s_t) \\ b_t \sim \pi_\psi(\cdot|s_t) \\ s_{t+1} \sim P(\cdot|s_t, a_t, b_t)}} \left[ \sum_{t=0}^{\infty} \gamma^t \left( \nabla_\phi \log \pi_\phi(a_t|s_t) \left( Q_{\mathcal{M}(x)}^{\pi_\phi, \pi_\psi}(s_t, a_t, b_t) - \tau_\psi \log \pi_\psi(b_t|s_t) + \tau_\phi \log \pi_\phi(a_t|s_t) \right) \right) \right], \\
&\nabla_\psi J(x, \phi, \psi) \\
&= \mathbb{E}_{\substack{s_0 \sim \rho, a_t \sim \pi_\phi(\cdot|s_t) \\ b_t \sim \pi_\psi(\cdot|s_t) \\ s_{t+1} \sim P(\cdot|s_t, a_t, b_t)}} \left[ \sum_{t=0}^{\infty} \gamma^t \left( \nabla_\psi \log \pi_\psi(b_t|s_t) \left( Q_{\mathcal{M}(x)}^{\pi_\phi, \pi_\psi}(s_t, a_t, b_t) - \tau_\psi \log \pi_\psi(b_t|s_t) + \tau_\phi \log \pi_\phi(a_t|s_t) \right) \right) \right],
\end{aligned}$$

*where $Q_{\mathcal{M}(x)}^{\pi_\phi, \pi_\psi}(s_t, a_t, b_t)$ is the Q-function defined in* (10).

*Proof.* By definition, we have

$$
\begin{aligned}
& J(x, \phi, \psi) \\
& = \mathop{\mathbb{E}}_{\substack{s_0 \sim \rho, a_t \sim \pi_\phi(\cdot|s_t) \\ b_t \sim \pi_\psi(\cdot|s_t) \\ s_{t+1} \sim P(\cdot|s_t, a_t, b_t)}} \left[ \sum_{t=0}^\infty \gamma^t \left[ r_x(s_t, a_t, b_t) + \tau_\phi \log \pi_\phi(a_t|s_t) - \tau_\psi \log \pi_\psi(b_t|s_t) \right] \right] \\
& = \mathop{\mathbb{E}}_{\substack{s_0 \sim \rho, a_t \sim \pi_\phi(\cdot|s_t) \\ s_{t+1} \sim P^{\pi_\psi}(\cdot|s_t, a_t)}} \left[ \sum_{t=0}^\infty \gamma^t \left[ \mathbb{E}_{b_t \sim \pi_\psi(\cdot|s_t)}[r_x(s_t, a_t, b_t)] + \tau_\phi \log \pi_\phi(a_t|s_t) - \tau_\psi \mathbb{E}_{b_t \sim \pi_\psi(\cdot|s_t)}[\log \pi_\psi(b_t|s_t)] \right] \right] \\
& = \mathop{\mathbb{E}}_{\substack{s_0 \sim \rho, a_t \sim \pi_\phi(\cdot|s_t) \\ s_{t+1} \sim P^{\pi_\psi}(\cdot|s_t, a_t)}} \left[ \sum_{t=0}^\infty \gamma^t \left[ r_x^{\pi_\psi}(s_t, a_t) + \tau_\phi \log \pi_\phi(a_t|s_t) + \tau_\psi H(\pi_\psi(\cdot|s_t)) \right] \right]
\end{aligned}
\tag{51}
$$

where $P^{\pi_\psi}(\cdot|s_t, a_t) \triangleq \mathbb{E}_{b_t \sim \pi_\psi(\cdot|s_t)}[P(\cdot|s_t, a_t, b_t)]$ and $r_x^{\pi_\psi}(s_t, a_t) \triangleq \mathbb{E}_{b_t \sim \pi_\psi(\cdot|s_t)}[r_x(s_t, a_t, b_t)]$. Here $H(\pi_\psi(\cdot|s_t))$ does not depend on $\pi_\phi$.

Since $\pi_\phi$ plays the role of the minimizing player, we define a corresponding reward model for $\pi_\phi$ as

$$
\tilde{r}_{(x,\psi)}(s, a) \triangleq -r_x^{\pi_\psi}(s, a) - \tau_\psi H(\pi_\psi(\cdot|s)) = -\mathbb{E}_{b \sim \pi_\psi(\cdot|s)}[r_x(s, a, b) - \tau_\psi \log \pi_\psi(b|s)].
$$

For any fixed $\pi_\psi$, the optimization over $\pi_\phi$ can then be reformulated as a single-policy entropy-regularized MDP $\mathcal{M}^{\pi_\psi}(x) = \{\mathcal{S}, \mathcal{A}, P^{\pi_\psi}, \tilde{r}_{(x,\psi)}, \gamma\}$. Under the transformed reward $\tilde{r}_{(x,\psi)}$, the original minimization over $\pi_\phi$ is equivalent to maximizing the following entropy-regularized value function:

$$
\widetilde{V}_{\mathcal{M}^{\pi_\psi}(x)}^{\pi_\phi}(\rho) = \mathop{\mathbb{E}}_{\substack{s_0 \sim \rho, a_t \sim \pi_\phi(\cdot|s_t) \\ s_{t+1} \sim P^{\pi_\psi}(\cdot|s_t, a_t)}} \left[ \sum_{t=0}^\infty \gamma^t \left[ \tilde{r}_{(x,\psi)}(s_t, a_t) - \tau_\phi \log \pi_\phi(a_t|s_t) \right] \right],
$$

which equals $-J(x, \phi, \psi)$. Accordingly, the state-value function under policy $\pi_\phi$ is given by

$$
\widetilde{V}_{\mathcal{M}^{\pi_\psi}(x)}^{\pi_\phi}(s) = \mathop{\mathbb{E}}_{\substack{s_0 = s, a_t \sim \pi_\phi(\cdot|s_t) \\ s_{t+1} \sim P^{\pi_\psi}(\cdot|s_t, a_t)}} \left[ \sum_{t=0}^\infty \gamma^t \left[ \tilde{r}_{(x,\psi)}(s_t, a_t) - \tau_\phi \log \pi_\phi(a_t|s_t) \right] \right] = -V_{\mathcal{M}(x)}^{\pi_\phi, \pi_\psi}(s).
$$

Define the corresponding Q-function for $\mathcal{M}^{\pi_\psi}(x)$ as

$$
\widetilde{Q}_{\mathcal{M}^{\pi_\psi}(x)}^{\pi_\phi}(s, a) \triangleq \tilde{r}_{(x,\psi)}(s, a) + \gamma \mathbb{E}_{s' \sim P^{\pi_\psi}(\cdot|s, a)} \left[ \widetilde{V}_{\mathcal{M}^{\pi_\psi}(x)}^{\pi_\phi}(s') \right].
\tag{52}
$$

It follows that

$$
\begin{aligned}
\widetilde{Q}_{\mathcal{M}^{\pi_\psi}(x)}^{\pi_\phi}(s, a) & = -\mathbb{E}_{b \sim \pi_\psi(\cdot|s)}[r_x(s, a, b) - \tau_\psi \log \pi_\psi(b|s)] + \gamma \mathop{\mathbb{E}}_{\substack{b \sim \pi_\psi(\cdot|s) \\ s' \sim P(\cdot|s, a, b)}} \left[ \widetilde{V}_{\mathcal{M}^{\pi_\psi}(x)}^{\pi_\phi}(s') \right] \\
& = \mathbb{E}_{b \sim \pi_\psi(\cdot|s)} \left[ -r_x(s, a, b) + \tau_\psi \log \pi_\psi(b|s) + \gamma \mathop{\mathbb{E}}_{s' \sim P(\cdot|s, a, b)} \left[ -V_{\mathcal{M}(x)}^{\pi_\phi, \pi_\psi}(s') \right] \right] \\
& = -\mathbb{E}_{b \sim \pi_\psi(\cdot|s)} \left[ Q_{\mathcal{M}(x)}^{\pi_\phi, \pi_\psi}(s, a, b) - \tau_\psi \log \pi_\psi(b|s) \right].
\end{aligned}
\tag{53}
$$

Then by Lemma 10 of Mei et al. (2020), we can get

$$
\nabla_\phi \widetilde{V}^{\pi_\phi}_{\mathcal{M}^{\pi_\psi}(x)}(\rho)
$$
$$
= \mathop{\mathbb{E}}_{\substack{s_0\sim\rho, a_t\sim\pi_\phi(\cdot|s_t) \\ s_{t+1}\sim P^{\pi_\psi}(\cdot|s_t,a_t)}} \left[ \sum_{t=0}^\infty \gamma^t \left( \nabla_\phi \log \pi_\phi(a_t|s_t) \left( \widetilde{Q}^{\pi_\phi}_{\mathcal{M}^{\pi_\psi}(x)}(s_t,a_t) - \tau_\phi \log \pi_\phi(a_t|s_t) \right) \right) \right]
$$
$$
= \mathop{\mathbb{E}}_{\substack{s_0\sim\rho, a_t\sim\pi_\phi(\cdot|s_t) \\ b_t\sim\pi_\psi(\cdot|s_t) \\ s_{t+1}\sim P(\cdot|s_t,a_t,b_t)}} \left[ \sum_{t=0}^\infty \gamma^t \left( \nabla_\phi \log \pi_\phi(a_t|s_t) \left( \widetilde{Q}^{\pi_\phi}_{\mathcal{M}^{\pi_\psi}(x)}(s_t,a_t) - \tau_\phi \log \pi_\phi(a_t|s_t) \right) \right) \right]
$$
$$
= -\mathop{\mathbb{E}}_{\substack{s_0\sim\rho, a_t\sim\pi_\phi(\cdot|s_t) \\ b_t\sim\pi_\psi(\cdot|s_t) \\ s_{t+1}\sim P(\cdot|s_t,a_t,b_t)}} \left[ \sum_{t=0}^\infty \gamma^t \left( \nabla_\phi \log \pi_\phi(a_t|s_t) \left( \mathop{\mathbb{E}}_{b_t\sim\pi_\psi(b_t|s_t)} \left[ Q^{\pi_\phi,\pi_\psi}_{\mathcal{M}(x)}(s_t,a_t,b_t) - \tau_\psi \log \pi_\psi(b_t|s_t) \right] \right. \right. \right.
$$
$$
\left. \left. \left. + \tau_\phi \log \pi_\phi(a_t|s_t) \right) \right) \right], \tag{54}
$$

where the second equality is to expand $b_t$ from $P^{\pi_\psi}$, and the last equality follows from the definition of $\widetilde{Q}^{\pi_\phi}_{\mathcal{M}^{\pi_\psi}(x)}(s_t,a_t)$.
This implies that

$$
\nabla_\phi J(x,\phi,\psi)
$$
$$
= \mathop{\mathbb{E}}_{\substack{s_0\sim\rho, a_t\sim\pi_\phi(\cdot|s_t) \\ b_t\sim\pi_\psi(\cdot|s_t) \\ s_{t+1}\sim P(\cdot|s_t,a_t,b_t)}} \left[ \sum_{t=0}^\infty \gamma^t \left( \nabla_\phi \log \pi_\phi(a_t|s_t) \left( \mathop{\mathbb{E}}_{b_t\sim\pi_\psi(b_t|s_t)} \left[ Q^{\pi_\phi,\pi_\psi}_{\mathcal{M}(x)}(s_t,a_t,b_t) - \tau_\psi \log \pi_\psi(b_t|s_t) \right] \right. \right. \right.
$$
$$
\left. \left. \left. + \tau_\phi \log \pi_\phi(a_t|s_t) \right) \right) \right]
$$
$$
= \frac{1}{1-\gamma} \sum_s d^{\pi_\phi,\pi_\psi}_\rho(s) \sum_a \pi_\phi(a|s) \left( \nabla_\phi \log \pi_\phi(a|s) \left( \sum_b \pi_\psi(b|s) \left[ Q^{\pi_\phi,\pi_\psi}_{\mathcal{M}(x)}(s,a,b) - \tau_\psi \log \pi_\psi(b|s) \right] \right. \right.
$$
$$
\left. \left. + \tau_\phi \log \pi_\phi(a|s) \right) \right)
$$
$$
= \frac{1}{1-\gamma} \sum_s d^{\pi_\phi,\pi_\psi}_\rho(s) \sum_{a,b} \pi_\phi(a|s)\pi_\psi(b|s) \left( \nabla_\phi \log \pi_\phi(a|s) \left( Q^{\pi_\phi,\pi_\psi}_{\mathcal{M}(x)}(s,a,b) - \tau_\psi \log \pi_\psi(b|s) + \tau_\phi \log \pi_\phi(a|s) \right) \right)
$$
$$
= \mathop{\mathbb{E}}_{\substack{s_0\sim\rho, a_t\sim\pi_\phi(\cdot|s_t) \\ b_t\sim\pi_\psi(\cdot|s_t) \\ s_{t+1}\sim P(\cdot|s_t,a_t,b_t)}} \left[ \sum_{t=0}^\infty \gamma^t \left( \nabla_\phi \log \pi_\phi(a_t|s_t) \left( Q^{\pi_\phi,\pi_\psi}_{\mathcal{M}(x)}(s_t,a_t,b_t) - \tau_\psi \log \pi_\psi(b_t|s_t) + \tau_\phi \log \pi_\phi(a_t|s_t) \right) \right) \right]. \tag{55}
$$

Similarly, we can prove that

$$
\nabla_\psi J(x,\phi,\psi)
$$
$$
= \mathop{\mathbb{E}}_{\substack{s_0\sim\rho, a_t\sim\pi_\phi(\cdot|s_t) \\ b_t\sim\pi_\psi(\cdot|s_t) \\ s_{t+1}\sim P(\cdot|s_t,a_t,b_t)}} \left[ \sum_{t=0}^\infty \gamma^t \left( \nabla_\psi \log \pi_\psi(b_t|s_t) \left( Q^{\pi_\phi,\pi_\psi}_{\mathcal{M}(x)}(s_t,a_t,b_t) - \tau_\psi \log \pi_\psi(b_t|s_t) + \tau_\phi \log \pi_\phi(a_t|s_t) \right) \right) \right]. \tag{56}
$$

$\square$

## C. Proofs in Section 3

### C.1. Useful Properties of Single-Policy MDPs

The notation in this subsection **differs** from that in the rest of the paper, as we focus exclusively on the properties of single-policy MDPs in this section.

Let $\mathcal{M}(x) = \{\mathcal{S}, \mathcal{A}, r_x, \mathcal{P}_x, \gamma, h\}$ be a parameterized single-policy MDP with entropy regularization. For any policy $\pi$, the soft state-value function at state $s$ is defined as

$$V_{\mathcal{M}(x)}^{\pi}(s) \triangleq \mathbb{E}\left[\sum_{t=0}^{\infty} \gamma^t \left(r_x(s_t, a_t) - \tau \log \pi(a_t|s_t)\right) \bigg| s_0 = s, \pi, \mathcal{M}(x)\right],$$

where $\tau > 0$ is the regularization coefficient. Then for a full-support initial state distribution $\rho$, we can define the soft value function of policy $\pi$ as $V_{\mathcal{M}(x)}^{\pi}(\rho) \triangleq \mathbb{E}_{s \sim \rho}[V_{\mathcal{M}(x)}^{\pi}(s)]$. These are all standard definitions and notations in single-policy RL.

For any given $x$, let $V^*(x)$ be the soft optimal state-value such that $V_s^*(x) = \max_\pi V_{\mathcal{M}(x)}^{\pi}(s)$. By Nachum et al. (2017) and Yang et al. (2025), $V^*(x)$ is the unique fixed point of the softmax Bellman operator $\mathcal{T}_x : \mathbb{R}^{|\mathcal{S}|} \mapsto \mathbb{R}^{|\mathcal{S}|}$ defined as

$$(\mathcal{T}_x V)(s) = \tau \log \sum_a \exp\left(\frac{r_x(s, a) + \gamma \sum_{s'} \mathcal{P}_x(s'|s, a) V(s')}{\tau}\right),$$

which is $\gamma$-contraction in $V$ w.r.t. $\|\cdot\|_\infty$. Define the optimal soft Q-value as

$$Q_{sa}^*(x) \triangleq r_x(s, a) + \gamma \sum_{s'} \mathcal{P}_x(s'|s, a) V_{s'}^*(x),$$

then the optimal policy $\pi_x^*$ is given by

$$\pi_x^*(a|s) = \frac{\exp(Q_{sa}^*(x)/\tau)}{\sum_{a'} \exp(Q_{sa'}^*(x)/\tau)}.$$

Now we make some standard assumptions on the parameterized MDP $\mathcal{M}(x)$.

**Assumption C.1.** Suppose the following conditions hold for $\mathcal{M}(x)$:

1. Bounded reward: $|r_x(s, a)| \leq R_{\max}$ for any $(s, a) \in \mathcal{S} \times \mathcal{A}$ and any $x \in \mathbb{R}^{d_x}$.

2. Lipschitz reward: $\sup_{s,a} |r_{x_1}(s, a) - r_{x_2}(s, a)| \leq C_r \|x_1 - x_2\|$ for any $x_1, x_2 \in \mathbb{R}^{d_x}$.

3. Lipschitz transition: $\sup_{s,a} \|\mathcal{P}_{x_1}(\cdot|s, a) - \mathcal{P}_{x_2}(\cdot|s, a)\|_1 \leq C_P \|x_1 - x_2\|$ for any $x_1, x_2 \in \mathbb{R}^{d_x}$.

Then we can establish the following Lipschitz continuity of the optimal soft state-value and soft Q-value w.r.t. $x$.

**Lemma C.1.** *Under Assumption C.1, the optimal soft state-value $V^*(x)$ is $C_V$-Lipschitz continuous in $x$ for some constant $C_V > 0$. The optimal soft Q-value $Q^*(x) \in \mathbb{R}^{|\mathcal{S}||\mathcal{A}|}$ is $C_Q$-Lipschitz continuous in $x$ for some constant $C_Q > 0$.*

*Proof.* For any $x_1, x_2 \in \mathbb{R}^{d_x}$ and any $(s, a) \in \mathcal{S} \times \mathcal{A}$, we have

$$|\mathcal{P}_{x_1}(\cdot|s, a)^T V - \mathcal{P}_{x_2}(\cdot|s, a)^T V| \leq \|\mathcal{P}_{x_1}(\cdot|s, a) - \mathcal{P}_{x_2}(\cdot|s, a)\|_1 \|V\|_\infty \leq C_P \|x_1 - x_2\| \|V\|_\infty.$$

Thus, we can obtain

$$\sup_{s,a} \left|\left(r_{x_1}(s, a) + \gamma \sum_{s'} \mathcal{P}_{x_1}(s'|s, a) V_{s'}\right) - \left(r_{x_2}(s, a) + \gamma \sum_{s'} \mathcal{P}_{x_2}(s'|s, a) V_{s'}\right)\right|$$
$$\leq (C_r + \gamma C_P \|V\|_\infty) \|x_1 - x_2\|. \tag{57}$$

By the 1-Lipschitz continuity of the log-sum-exp function in $\|\cdot\|_\infty$, we have for any $v, w \in \mathbb{R}^{|\mathcal{A}|}$,

$$|\tau \log \sum_a e^{v_a/\tau} - \tau \log \sum_a e^{w_a/\tau}| \leq \|v - w\|_\infty = \sup_a |v_a - w_a|,$$

and hence

$$\begin{aligned}
\|\mathcal{T}_{x_1} V - \mathcal{T}_{x_2} V\|_\infty &= \sup_s |(\mathcal{T}_{x_1} V)(s) - (\mathcal{T}_{x_2} V)(s)| \\
&\leq \sup_{s,a} \left| \left( r_{x_1}(s,a) + \gamma \sum_{s'} \mathcal{P}_{x_1}(s'|s,a) V_{s'} \right) - \left( r_{x_2}(s,a) + \gamma \sum_{s'} \mathcal{P}_{x_2}(s'|s,a) V_{s'} \right) \right| \\
&\leq (C_r + \gamma C_P \|V\|_\infty) \|x_1 - x_2\|.
\end{aligned} \tag{58}$$

Since for any $x$, $V^*(x)$ is the unique fixed point of $\mathcal{T}_x$ (Nachum et al., 2017), i.e., $\mathcal{T}_x V^*(x) = V^*(x)$, then

$$\begin{aligned}
\|V^*(x_1) - V^*(x_2)\|_\infty &= \|\mathcal{T}_{x_1} V^*(x_1) - \mathcal{T}_{x_2} V^*(x_2)\|_\infty \\
&\leq \|\mathcal{T}_{x_1} V^*(x_1) - \mathcal{T}_{x_1} V^*(x_2)\|_\infty + \|\mathcal{T}_{x_1} V^*(x_2) - \mathcal{T}_{x_2} V^*(x_2)\|_\infty \\
&\leq \gamma \|V^*(x_1) - V^*(x_2)\|_\infty + (C_r + \gamma C_P \|V^*(x_2)\|_\infty) \|x_1 - x_2\|,
\end{aligned}$$

which implies

$$\|V^*(x_1) - V^*(x_2)\|_\infty \leq \frac{C_r + \gamma C_P \|V^*(x_2)\|_\infty}{1 - \gamma} \|x_1 - x_2\|. \tag{59}$$

Since for any $x \in \mathbb{R}^{d_x}$ and any $s \in \mathcal{S}$,

$$|(\mathcal{T}_x \mathbf{0})(s)| = \left| \tau \log \sum_a \exp\left( \frac{r_x(s,a)}{\tau} \right) \right| \leq R_{\max} + \tau \log |\mathcal{A}|,$$

then by $\gamma$-contraction of $\mathcal{T}_x$ with respect to $\|\cdot\|_\infty$, we can obtain,

$$\begin{aligned}
\|V^*(x)\|_\infty &= \|\mathcal{T}_x V^*(x)\|_\infty \\
&\leq \|\mathcal{T}_x V^*(x) - \mathcal{T}_x \mathbf{0}\|_\infty + \|\mathcal{T}_x \mathbf{0}\|_\infty \\
&\leq \gamma \|V^*(x)\|_\infty + R_{\max} + \tau \log |\mathcal{A}|,
\end{aligned}$$

which implies $\|V^*(x)\|_\infty \leq \frac{R_{\max} + \tau \log |\mathcal{A}|}{1-\gamma}$ for any $x \in \mathbb{R}^{d_x}$. We use $B_V$ here to denote $\frac{R_{\max} + \tau \log |\mathcal{A}|}{1-\gamma}$ for brevity. Therefore, we have

$$\|V^*(x_1) - V^*(x_2)\|_\infty \leq \frac{C_r + \gamma C_P B_V}{1 - \gamma} \|x_1 - x_2\|, \tag{60}$$

which completes the proof for the Lipschitz continuity of $V^*(x)$ by applying $\|\cdot\|_2 \leq \sqrt{|\mathcal{S}|}\|\cdot\|_\infty$.

Finally, we prove the Lipschitz continuity of $Q^*(x)$. For any $x_1, x_2 \in \mathbb{R}^{d_x}$, we have

$$\begin{aligned}
&|Q^*_{sa}(x_1) - Q^*_{sa}(x_2)| \\
&= |r_{x_1}(s,a) - r_{x_2}(s,a) + \gamma \sum_{s'} (\mathcal{P}_{x_1}(s'|s,a) V^*_{s'}(x_1) - \mathcal{P}_{x_2}(s'|s,a) V^*_{s'}(x_2))| \\
&\leq |r_{x_1}(s,a) - r_{x_2}(s,a)| + \gamma \sum_{s'} |\mathcal{P}_{x_1}(s'|s,a) V^*_{s'}(x_1) - \mathcal{P}_{x_2}(s'|s,a) V^*_{s'}(x_2)| \\
&\leq C_r \|x_1 - x_2\| + \gamma \sum_{s'} |\mathcal{P}_{x_1}(s'|s,a)(V^*_{s'}(x_1) - V^*_{s'}(x_2))| + \gamma \sum_{s'} |(\mathcal{P}_{x_1}(s'|s,a) - \mathcal{P}_{x_2}(s'|s,a)) V^*_{s'}(x_2)| \\
&\leq C_r \|x_1 - x_2\| + \gamma \|V^*(x_1) - V^*(x_2)\|_\infty + \gamma \|\mathcal{P}_{x_1}(\cdot|s,a) - \mathcal{P}_{x_2}(\cdot|s,a)\|_1 \|V^*(x_2)\|_\infty \\
&\leq C_r \|x_1 - x_2\| + \gamma \frac{C_r + \gamma C_P B_V}{1 - \gamma} \|x_1 - x_2\| + \gamma C_P B_V \|x_1 - x_2\|,
\end{aligned} \tag{61}$$

which completes the proof by applying $\|Q^*(x_1) - Q^*(x_2)\| \leq \sqrt{|\mathcal{S}||\mathcal{A}|} \max_{s,a} |Q_{sa}^*(x_1) - Q_{sa}^*(x_2)|$.

$\square$

**Lemma C.2.** *Under Assumption C.1, let $\pi^*(x) = \arg\max_\pi V_{\mathcal{M}(x)}^\pi(\rho)$ be the optimal soft policy for the MDP $\mathcal{M}(x)$. Then there exists a constant $C_{\pi^*} > 0$ such that $\pi^*(x)$ is $C_{\pi^*}$-Lipschitz continuous in $x$.*

*Proof.* By Nachum et al. (2017), the optimal soft policy for any $s \in \mathcal{S}$ and $a \in \mathcal{A}$ can be represented as

$$\pi_{sa}^*(x) = \frac{\exp(Q_{sa}^*(x)/\tau)}{\sum_{a'} \exp(Q_{sa'}^*(x)/\tau)}, \tag{62}$$

where $Q^*(x)$ is the optimal soft Q-value defined in Lemma C.1.

Therefore, for any $x_1, x_2 \in \mathbb{R}^{d_x}$, we have

$$\|\pi^*(x_1) - \pi^*(x_2)\| \leq \frac{1}{2\tau}\|Q^*(x_1) - Q^*(x_2)\| \leq \frac{C_Q}{2\tau}\|x_1 - x_2\|, \tag{63}$$

which completes the proof. $\square$

**Lemma C.3.** *Under Assumption C.1, suppose the reward model $r_x$ and the transition model $\mathcal{P}_x$ also satisfy the following conditions:*

1. *$r_x(s, a)$ is differentiable w.r.t. $x$ and there exists a constant $L_r$ such that for any $x_1$ and $x_2$, we have $\sup_{s,a} \|\nabla r_{x_1}(s, a) - \nabla r_{x_2}(s, a)\| \leq L_r\|x_1 - x_2\|$.*

2. *$\mathcal{P}_x(s'|s, a)$ is differentiable w.r.t. $x$ and there exists a constant $L_P$ such that for any $x_1$ and $x_2$, we have $\sup_{s,a,s'} \|\nabla \mathcal{P}_{x_1}(s'|s, a) - \nabla \mathcal{P}_{x_2}(s'|s, a)\| \leq L_P\|x_1 - x_2\|$.*

*Let $\pi^*(x) = \arg\max_\pi V_{\mathcal{M}(x)}^\pi(\rho)$, then there exists a constant $L_\pi$ such that for any $x_1$ and $x_2$, we have*

$$\|\nabla\pi^*(x_1) - \nabla\pi^*(x_2)\| \leq L_\pi\|x_1 - x_2\|.$$

*Proof.* For any $s \in \mathcal{S}$ and $a \in \mathcal{A}$,

$$\pi_{sa}^*(x) = \frac{\exp(Q_{sa}^*(x)/\tau)}{\sum_{a'} \exp(Q_{sa'}^*(x)/\tau)} = \exp\left(\frac{Q_{sa}^*(x)}{\tau} - \log\sum_{a'} \exp\left(\frac{Q_{sa'}^*(x)}{\tau}\right)\right).$$

By the definition of $Q_{sa}^*(x)$, we have

$$Q_{sa}^*(x) = r_x(s, a) + \gamma\sum_{s'} \mathcal{P}_x(s'|s, a)V_{s'}^*(x).$$

This implies that

$$\log\sum_{a'} \exp\left(\frac{Q_{sa'}^*(x)}{\tau}\right) = \log\sum_{a'} \exp\left(\frac{r_x(s, a') + \gamma\sum_{s'} \mathcal{P}_x(s'|s, a')V_{s'}^*(x)}{\tau}\right)$$

$$= \tau^{-1}(\mathcal{T}_x V^*(x))(s) = \tau^{-1}V_s^*(x).$$

Thus, we can represent $\pi_{sa}^*(x)$ as

$$\pi_{sa}^*(x) = \exp\left(\frac{1}{\tau}(Q_{sa}^*(x) - V_s^*(x))\right). \tag{64}$$

Taking the gradient with respect to $x$ on both sides, we can obtain

$$\nabla\pi_{sa}^*(x) = \frac{\pi_{sa}^*(x)}{\tau}\left(\nabla r_x(s, a) + \gamma\sum_{s'} \nabla\mathcal{P}_x(s'|s, a)V_{s'}^*(x) + \gamma\sum_{s'} \mathcal{P}_x(s'|s, a)\nabla V_{s'}^*(x) - \nabla V_s^*(x)\right).$$

In matrix form, this can be written as

$$\nabla\pi^*(x) = \frac{1}{\tau}\,\text{diag}(\pi^*(x))\left(\nabla r_x + \gamma D_P(x) - U(x)\nabla V^*(x)\right), \tag{65}$$

where each element $U \in \mathbb{R}^{|\mathcal{S}||\mathcal{A}|\times|\mathcal{S}|}$ is defined as $U_{sas'} \triangleq 1 - \gamma\mathcal{P}_{sas'}(x)$ for $s = s'$ and $U_{sas'} \triangleq -\gamma\mathcal{P}_{sas'}(x)$ otherwise, and each element of $D_P(x) \in \mathbb{R}^{|\mathcal{S}||\mathcal{A}|\times d_x}$ is defined as

$$D_P(x)_{sa} \triangleq \sum_{s'} V_{s'}^*(x)\nabla\mathcal{P}_x(s'|s,a)^T.$$

Consider the matrix $U(x)$, we have

$$\|U(x)\| = \sqrt{\sum_{\substack{s,a,s'\\s\neq s'}}\gamma^2\mathcal{P}_{sas'}(x)^2 + \sum_{s,a}(1-\gamma\mathcal{P}_{sas}(x))^2} \leq \sqrt{|\mathcal{S}||\mathcal{A}|} \triangleq B_U, \tag{66}$$

and

$$\|U(x_1) - U(x_2)\| \leq \sqrt{|\mathcal{S}||\mathcal{A}|}\sup_{s,a}\sum_{s'}|U_{sas'}(x_1) - U_{sas'}(x_2)| \leq \gamma C_P\sqrt{|\mathcal{S}||\mathcal{A}|}\|x_1 - x_2\| \triangleq C_U\|x_1 - x_2\|. \tag{67}$$

Consider the transition gradient term $D_P(x)$, we have for any state $s$ and action $a$,

$$\|D_P(x)_{sa}\| = \|\sum_{s'} V_{s'}^*(x)\nabla\mathcal{P}_x(s'|s,a)^T\|$$

$$\leq \sum_{s'}|V_{s'}^*(x)|\|\nabla\mathcal{P}_x(s'|s,a)\|$$

$$\leq |\mathcal{S}|B_V C_P.$$

Thus, we can obtain

$$\|D_P(x)\| \leq \sqrt{|\mathcal{S}||\mathcal{A}|}\sup_{s,a}\|D_P(x)_{sa}\| \leq |\mathcal{S}|^{\frac{3}{2}}|\mathcal{A}|^{\frac{1}{2}}B_V C_P \triangleq B_{D_P}, \tag{68}$$

and for any $x_1, x_2 \in \mathbb{R}^{d_x}$, any state $s$ and action $a$, we have

$$\|D_P(x_1)_{sa} - D_P(x_2)_{sa}\|$$

$$= \left\|\sum_{s'} V_{s'}^*(x_1)\nabla\mathcal{P}_{x_1}(s'|s,a)^T - \sum_{s'} V_{s'}^*(x_2)\nabla\mathcal{P}_{x_2}(s'|s,a)^T\right\|$$

$$\leq \sum_{s'}\|V_{s'}^*(x_1)\left(\nabla\mathcal{P}_{x_1}(s'|s,a)^T - \nabla\mathcal{P}_{x_2}(s'|s,a)^T\right)\| + \sum_{s'}\|(V_{s'}^*(x_1) - V_{s'}^*(x_2))\nabla\mathcal{P}_{x_2}(s'|s,a)^T\|$$

$$\leq |\mathcal{S}|B_V\sup_{s'}\|\nabla\mathcal{P}_{x_1}(s'|s,a) - \nabla\mathcal{P}_{x_2}(s'|s,a)\| + |\mathcal{S}|C_P\|V^*(x_1) - V^*(x_2)\|_\infty$$

$$\leq \left(|\mathcal{S}|B_V L_P + |\mathcal{S}|C_P\frac{C_r + \gamma C_P B_V}{1-\gamma}\right)\|x_1 - x_2\|$$

$$\triangleq C_{D_P}\|x_1 - x_2\|, \tag{69}$$

where the second last inequality is due to the Lipschitz continuity of $\nabla\mathcal{P}_x$ and $V^*(x)$ w.r.t. $x$.

Moreover, define the mapping $\Phi : \mathbb{R}^{d_x} \times \mathbb{R}^{|\mathcal{S}|} \mapsto \mathbb{R}^{|\mathcal{S}|}$ as

$$\Phi(x,v) = \tau\log\left(\exp\left(\frac{r(x) + \gamma\mathcal{P}(x)v}{\tau}\right)\cdot\mathbf{1}\right),$$

where $\log(\cdot)$ and $\exp(\cdot)$ are element-wise operations, and $\mathbf{1} \in \mathbb{R}^{|\mathcal{A}|}$ is an all-one vector. Then $V^*(x)$ is the fixed point of this mapping.

By Proposition 4.1 in Yang et al. (2025), we know that $V^*(x)$ is the unique fixed point of the mapping $\Phi(x, \cdot)$ for any $x \in \mathbb{R}^{d_x}$, and $V^*(x)$ is differentiable w.r.t. $x$ with

$$\nabla V^*(x) = (I - \gamma \mathcal{P}_x^{\pi^*(x)})^{-1} \nabla_1 \Phi(x, V^*(x))$$
$$= \sum_{t=0}^{\infty} \gamma^t \left( \mathcal{P}_x^{\pi^*(x)} \right)^t \nabla_1 \Phi(x, V^*(x)), \tag{70}$$

where $\mathcal{P}_x^{\pi^*(x)} \in \mathbb{R}^{|\mathcal{S}| \times |\mathcal{S}|}$ is the transition kernel induced by the optimal policy $\pi^*(x)$, with each element defined as $\left( \mathcal{P}_x^{\pi^*(x)} \right)_{ss'} = \sum_a \pi_{sa}^*(x) \mathcal{P}_x(s'|s, a)$.

Consequently, for any $x_1, x_2 \in \mathbb{R}^{d_x}$, we obtain

$$\|\nabla V^*(x_1) - \nabla V^*(x_2)\| = \left\| \sum_{t=0}^{\infty} \gamma^t \left( \left( \mathcal{P}_{x_1}^{\pi^*(x_1)} \right)^t \nabla_1 \Phi(x_1, V^*(x_1)) - \left( \mathcal{P}_{x_2}^{\pi^*(x_2)} \right)^t \nabla_1 \Phi(x_2, V^*(x_2)) \right) \right\|$$
$$\leq \left\| \sum_{t=0}^{\infty} \gamma^t \left( \left( \mathcal{P}_{x_1}^{\pi^*(x_1)} \right)^t \nabla_1 \Phi(x_1, V^*(x_1)) - \left( \mathcal{P}_{x_2}^{\pi^*(x_2)} \right)^t \nabla_1 \Phi(x_1, V^*(x_1)) \right) \right\|$$
$$+ \left\| \sum_{t=0}^{\infty} \gamma^t \left( \left( \mathcal{P}_{x_2}^{\pi^*(x_2)} \right)^t \nabla_1 \Phi(x_1, V^*(x_1)) - \left( \mathcal{P}_{x_2}^{\pi^*(x_2)} \right)^t \nabla_1 \Phi(x_2, V^*(x_2)) \right) \right\|.$$

Moreover, we have for any state $s$,

$$\|\mathcal{P}_{x_1}^{\pi^*(x_1)}(\cdot|s) - \mathcal{P}_{x_2}^{\pi^*(x_2)}(\cdot|s)\|_{\infty}$$
$$= \| \sum_a \pi^*(a|s; x_1) \mathcal{P}_{x_1}(\cdot|s, a) - \sum_a \pi^*(a|s; x_2) \mathcal{P}_{x_2}(\cdot|s, a) \|_{\infty}$$
$$\leq \| \sum_a (\pi^*(a|s; x_1) - \pi^*(a|s; x_2)) \mathcal{P}_{x_1}(\cdot|s, a) \|_{\infty} + \| \sum_a \pi^*(a|s; x_1) (\mathcal{P}_{x_1}(\cdot|s, a) - \mathcal{P}_{x_2}(\cdot|s, a)) \|_{\infty}$$
$$\leq \sum_a |\pi^*(a|s; x_1) - \pi^*(a|s; x_2)| \|\mathcal{P}_{x_1}(\cdot|s, a)\|_{\infty} + \sum_a \pi^*(a|s; x_1) \|\mathcal{P}_{x_1}(\cdot|s, a) - \mathcal{P}_{x_2}(\cdot|s, a)\|_{\infty}$$
$$\leq \|\pi^*(\cdot|s; x_1) - \pi^*(\cdot|s; x_2)\|_1 + \sup_{s,a} \|\mathcal{P}_{x_1}(\cdot|s, a) - \mathcal{P}_{x_2}(\cdot|s, a)\|_{\infty}$$
$$\leq \sqrt{|\mathcal{A}|} \|\pi^*(x_1) - \pi^*(x_2)\| + \sup_{s,a} \|\mathcal{P}_{x_1}(\cdot|s, a) - \mathcal{P}_{x_2}(\cdot|s, a)\|_1$$
$$\leq \left( \sqrt{|\mathcal{A}|} C_{\pi^*} + C_P \right) \|x_1 - x_2\|, \tag{71}$$

therefore, we obtain

$$\|\mathcal{P}_{x_1}^{\pi^*(x_1)} - \mathcal{P}_{x_2}^{\pi^*(x_2)}\|_{\infty} \leq |\mathcal{S}| \sup_s \|\mathcal{P}_{x_1}^{\pi^*(x_1)}(\cdot|s) - \mathcal{P}_{x_2}^{\pi^*(x_2)}(\cdot|s)\|_{\infty} \leq |\mathcal{S}| \left( \sqrt{|\mathcal{A}|} C_{\pi^*} + C_P \right) \|x_1 - x_2\|.$$

Hence, denote $A = \mathcal{P}_{x_1}^{\pi^*(x_1)} \in \mathbb{R}^{|\mathcal{S}| \times |\mathcal{S}|}$, $B = \mathcal{P}_{x_2}^{\pi^*(x_2)} \in \mathbb{R}^{|\mathcal{S}| \times |\mathcal{S}|}$. We obtain

$$\left\| \left( \mathcal{P}_{x_1}^{\pi^*(x_1)} \right)^t - \left( \mathcal{P}_{x_2}^{\pi^*(x_2)} \right)^t \right\|_{\infty} = \|A^t - B^t\|_{\infty}$$
$$= \left\| \sum_{i=0}^{t-1} A^{t-1-i}(A - B)B^i \right\|_{\infty}$$
$$\leq \sum_{i=0}^{t-1} \|A^{t-1-i}\|_{\infty} \|A - B\|_{\infty} \|B^i\|_{\infty}$$
$$\leq t \|A - B\|_{\infty}$$
$$\leq t |\mathcal{S}| \left( \sqrt{|\mathcal{A}|} C_{\pi^*} + C_P \right) \|x_1 - x_2\|, \tag{72}$$

which implies that there exists a constant $L_{pow} > 0$ such that

$$\left\| \left( \mathcal{P}_{x_1}^{\pi^*(x_1)} \right)^t - \left( \mathcal{P}_{x_2}^{\pi^*(x_2)} \right)^t \right\| \leq |\mathcal{S}| \left\| \left( \mathcal{P}_{x_1}^{\pi^*(x_1)} \right)^t - \left( \mathcal{P}_{x_2}^{\pi^*(x_2)} \right)^t \right\|_\infty \leq t L_{pow} \|x_1 - x_2\|. \tag{73}$$

Meanwhile, we have

$$\frac{\partial \Phi_s(x, v)}{\partial x_i} = \frac{\sum_a \left( \partial_{x_i} r_x(s, a) + \gamma \sum_{s'} \partial_{x_i} \mathcal{P}_x(s'|s, a) v_{s'} \right) \exp \left( \tau^{-1} \left( r_x(s, a) + \gamma \sum_{s'} \mathcal{P}_x(s'|s, a) v_{s'} \right) \right)}{\sum_a \exp \left( \tau^{-1} \left( r_x(s, a) + \gamma \sum_{s'} \mathcal{P}_x(s'|s, a) v_{s'} \right) \right)},$$

and substituting $v = V^*(x)$, we obtain

$$\nabla_1 \Phi_s(x, V^*(x)) = \sum_a \pi_{sa}^*(x) \left( \nabla r_x(s, a) + \gamma \sum_{s'} \nabla \mathcal{P}_x(s'|s, a) V_{s'}^*(x) \right). \tag{74}$$

Then for any $x$, we have

$$\|\nabla_1 \Phi_s(x, V^*(x))\| \leq \sum_a \pi_{sa}^*(x) \left( \|\nabla r_x(s, a)\| + \gamma \sum_{s'} \|\nabla \mathcal{P}_x(s'|s, a)\| |V_{s'}^*(x)| \right)$$
$$\leq C_r + \gamma |\mathcal{S}| C_P B_V \triangleq B_\Phi, \tag{75}$$

thus, we can arrive at

$$\|\nabla V^*(x)\| = \left\| \sum_{t=0}^{\infty} \gamma^t \left( \mathcal{P}_x^{\pi^*(x)} \right)^t \nabla_1 \Phi(x, V^*(x)) \right\|$$
$$\leq \sum_{t=0}^{\infty} \gamma^t \left\| \left( \mathcal{P}_x^{\pi^*(x)} \right)^t \right\| \|\nabla_1 \Phi(x, V^*(x))\|$$
$$\leq \frac{\sqrt{|\mathcal{S}|} B_\Phi}{1 - \gamma}. \tag{76}$$

We also have for any state $s$ and for any $x_1, x_2$,

$$\|\nabla_1 \Phi_s(x_1, V^*(x_1)) - \nabla_1 \Phi_s(x_2, V^*(x_2))\|$$
$$\leq \left\| \sum_a \pi_{sa}^*(x_1) \nabla r_{x_1}(s, a) - \sum_a \pi_{sa}^*(x_2) \nabla r_{x_2}(s, a) \right\|$$
$$+ \gamma \left\| \sum_a \pi_{sa}^*(x_1) \sum_{s'} \nabla \mathcal{P}_{x_1}(s'|s, a) V_{s'}^*(x_1) - \sum_a \pi_{sa}^*(x_2) \sum_{s'} \nabla \mathcal{P}_{x_2}(s'|s, a) V_{s'}^*(x_2) \right\|$$
$$\leq \left\| \sum_a (\pi_{sa}^*(x_1) - \pi_{sa}^*(x_2)) \nabla r_{x_1}(s, a) \right\| + \left\| \sum_a \pi_{sa}^*(x_2) (\nabla r_{x_1}(s, a) - \nabla r_{x_2}(s, a)) \right\|$$
$$+ \gamma \left\| \sum_a (\pi_{sa}^*(x_1) - \pi_{sa}^*(x_2)) \sum_{s'} \nabla \mathcal{P}_{x_1}(s'|s, a) V_{s'}^*(x_1) \right\|$$
$$+ \gamma \left\| \sum_a \pi_{sa}^*(x_2) \sum_{s'} (\nabla \mathcal{P}_{x_1}(s'|s, a) - \nabla \mathcal{P}_{x_2}(s'|s, a)) V_{s'}^*(x_1) \right\|$$
$$+ \gamma \left\| \sum_a \pi_{sa}^*(x_2) \sum_{s'} \nabla \mathcal{P}_{x_2}(s'|s, a) (V_{s'}^*(x_1) - V_{s'}^*(x_2)) \right\|$$
$$\leq C_r \sqrt{|\mathcal{A}|} C_{\pi^*} \|x_1 - x_2\| + L_r \|x_1 - x_2\| + \gamma |\mathcal{S}| C_P B_V \sqrt{|\mathcal{A}|} C_{\pi^*} \|x_1 - x_2\|$$
$$+ \gamma |\mathcal{S}| L_P B_V \|x_1 - x_2\| + \gamma |\mathcal{S}| C_P C_V \|x_1 - x_2\|$$
$$= \left( \sqrt{|\mathcal{A}|} C_r C_{\pi^*} + L_r + \gamma |\mathcal{S}| C_P B_V \sqrt{|\mathcal{A}|} C_{\pi^*} + \gamma |\mathcal{S}| L_P B_V + \gamma |\mathcal{S}| C_P C_V \right) \|x_1 - x_2\|, \tag{77}$$

which implies that there exists a constant $L_\Phi > 0$ such that

$$\|\nabla_1 \Phi(x_1, V^*(x_1)) - \nabla_1 \Phi(x_2, V^*(x_2))\| \leq \sqrt{|\mathcal{S}|} \max_s \|\nabla_1 \Phi_s(x_1, V^*(x_1)) - \nabla_1 \Phi_s(x_2, V^*(x_2))\|$$
$$\leq L_\Phi \|x_1 - x_2\|. \tag{78}$$

Subsequently, we can arrive at for any $x_1, x_2$,

$$\|\nabla V^*(x_1) - \nabla V^*(x_2)\| \leq \sum_{t=0}^\infty \gamma^t \left\| \left(\mathcal{P}_{x_1}^{\pi^*(x_1)}\right)^t - \left(\mathcal{P}_{x_2}^{\pi^*(x_2)}\right)^t \right\| \|\nabla_1 \Phi(x_1, V^*(x_1))\|$$
$$+ \sum_{t=0}^\infty \gamma^t \left\| \left(\mathcal{P}_{x_2}^{\pi^*(x_2)}\right)^t \right\| \|\nabla_1 \Phi(x_1, V^*(x_1)) - \nabla_1 \Phi(x_2, V^*(x_2))\|$$
$$\leq \sum_{t=0}^\infty \gamma^t t L_{pow} \|x_1 - x_2\| B_\Phi + \sum_{t=0}^\infty \gamma^t \sqrt{|\mathcal{S}|} L_\Phi \|x_1 - x_2\|$$
$$= \left( \frac{\gamma L_{pow} B_\Phi}{(1-\gamma)^2} + \frac{\sqrt{|\mathcal{S}|} L_\Phi}{1-\gamma} \right) \|x_1 - x_2\| \triangleq L_V \|x_1 - x_2\|. \tag{79}$$

Therefore, we have for any $x_1, x_2$,

$$\|\nabla \pi^*(x_1) - \nabla \pi^*(x_2)\|$$
$$= \left\| \tau^{-1} \operatorname{diag}(\pi^*(x_1)) \left(\nabla r(x_1) + \gamma D_\mathcal{P}(x_1) - U(x_1)\nabla V^*(x_1)\right) \right.$$
$$\left. - \tau^{-1} \operatorname{diag}(\pi^*(x_2)) \left(\nabla r(x_2) + \gamma D_\mathcal{P}(x_2) - U(x_2)\nabla V^*(x_2)\right) \right\|$$
$$\leq \tau^{-1} \|\operatorname{diag}(\pi^*(x_1))\nabla r(x_1) - \operatorname{diag}(\pi^*(x_2))\nabla r(x_2)\|$$
$$+ \tau^{-1}\gamma \|\operatorname{diag}(\pi^*(x_1))D_\mathcal{P}(x_1) - \operatorname{diag}(\pi^*(x_2))D_\mathcal{P}(x_2)\|$$
$$+ \tau^{-1} \|\operatorname{diag}(\pi^*(x_1))U(x_1)\nabla V^*(x_1) - \operatorname{diag}(\pi^*(x_2))U(x_2)\nabla V^*(x_2)\|$$
$$\leq \tau^{-1} \|\operatorname{diag}(\pi^*(x_1)) \left(\nabla r(x_1) - \nabla r(x_2)\right)\| + \tau^{-1} \|\left(\operatorname{diag}(\pi^*(x_1)) - \operatorname{diag}(\pi^*(x_2))\right) \nabla r(x_2)\|$$
$$+ \tau^{-1}\gamma \|\operatorname{diag}(\pi^*(x_1)) \left(D_\mathcal{P}(x_1) - D_\mathcal{P}(x_2)\right)\| + \tau^{-1}\gamma \|\left(\operatorname{diag}(\pi^*(x_1)) - \operatorname{diag}(\pi^*(x_2))\right) D_\mathcal{P}(x_2)\|$$
$$+ \tau^{-1} \|\operatorname{diag}(\pi^*(x_1))U(x_1) \left(\nabla V^*(x_1) - \nabla V^*(x_2)\right)\|$$
$$+ \tau^{-1} \|\operatorname{diag}(\pi^*(x_1)) \left(U(x_1) - U(x_2)\right) \nabla V^*(x_2)\|$$
$$+ \tau^{-1} \|\left(\operatorname{diag}(\pi^*(x_1)) - \operatorname{diag}(\pi^*(x_2))\right) U(x_2)\nabla V^*(x_2)\|$$
$$\leq \tau^{-1} \|\operatorname{diag}(\pi^*(x_1))\|\|\nabla r(x_1) - \nabla r(x_2)\| + \tau^{-1} \|\operatorname{diag}(\pi^*(x_1)) - \operatorname{diag}(\pi^*(x_2))\|\|\nabla r(x_2)\|$$
$$+ \tau^{-1}\gamma \|\operatorname{diag}(\pi^*(x_1))\| \|D_\mathcal{P}(x_1) - D_\mathcal{P}(x_2)\| + \tau^{-1}\gamma \|\operatorname{diag}(\pi^*(x_1)) - \operatorname{diag}(\pi^*(x_2))\| \|D_\mathcal{P}(x_2)\|$$
$$+ \tau^{-1} \|\operatorname{diag}(\pi^*(x_1))\|\|U(x_1)\|\|\nabla V^*(x_1) - \nabla V^*(x_2)\|$$
$$+ \tau^{-1} \|\operatorname{diag}(\pi^*(x_1))\|\|U(x_1) - U(x_2)\|\|\nabla V^*(x_2)\|$$
$$+ \tau^{-1} \|\operatorname{diag}(\pi^*(x_1)) - \operatorname{diag}(\pi^*(x_2))\|\|U(x_2)\|\|\nabla V^*(x_2)\|$$
$$\leq \tau^{-1}\sqrt{|\mathcal{S}||\mathcal{A}|}L_r\|x_1 - x_2\| + \tau^{-1}\sqrt{|\mathcal{S}||\mathcal{A}|}C_{\pi^*}C_r\|x_1 - x_2\|$$
$$+ \tau^{-1}\gamma C_{D_\mathcal{P}}\|x_1 - x_2\| + \tau^{-1}\gamma C_{\pi^*}B_{D_\mathcal{P}}\|x_1 - x_2\|$$
$$+ \tau^{-1}B_U L_V\|x_1 - x_2\| + \tau^{-1}\frac{\sqrt{|\mathcal{S}|}C_U B_\Phi}{1-\gamma}\|x_1 - x_2\| + \tau^{-1}\frac{B_U B_\Phi C_{\pi^*}}{1-\gamma}\|x_1 - x_2\|$$
$$= \tau^{-1}\left(\sqrt{|\mathcal{S}||\mathcal{A}|}(L_r + C_{\pi^*}C_r) + \gamma C_{D_\mathcal{P}} + \gamma C_{\pi^*}B_{D_\mathcal{P}} + B_U L_V + \frac{\sqrt{|\mathcal{S}|}C_U B_\Phi}{1-\gamma} + \frac{B_U B_\Phi C_{\pi^*}}{1-\gamma}\right)\|x_1 - x_2\| \tag{80}$$
$$\triangleq L_\pi \|x_1 - x_2\|, \tag{81}$$

where the second inequality is to separate the differences of each term, and the third inequality follows from the sub-multiplicative property of matrix norms. $\square$

### C.2. Useful Properties of Min–Max Zero-Sum Markov Games

For the regularized MMZSMG $\mathcal{M}(x) = \{\mathcal{S}, \mathcal{A}, \mathcal{B}, r_x, \mathcal{P}, \gamma\}$ defined in Section 2, we define the state-visitation distribution under policies $\pi_\phi$ and $\pi_\psi$ as

$$d_\rho^{\pi_\phi, \pi_\psi}(s) \triangleq (1-\gamma) \sum_{t=0}^\infty \gamma^t P(s_t = s | s_0 \sim \rho, \pi_\phi, \pi_\psi, \mathcal{M}(x)).$$

Consider the following two best-response optimization problems: $\max_{\pi_\psi \in \Delta_{|\mathcal{S}|}^{|\mathcal{B}|}} V_{\mathcal{M}(x)}^{\pi_\phi, \pi_\psi}(\rho)$ and $\min_{\pi_\phi \in \Delta_{|\mathcal{S}|}^{|\mathcal{A}|}} V_{\mathcal{M}(x)}^{\pi_\phi, \pi_\psi}(\rho)$. We have the following lemma.

**Lemma C.4.** *For any $x$ and any fixed $\pi_\phi \in \Delta_{|\mathcal{S}|}^{|\mathcal{A}|}$, the problem $\max_{\pi_\psi \in \Delta_{|\mathcal{S}|}^{|\mathcal{B}|}} V_{\mathcal{M}(x)}^{\pi_\phi, \pi_\psi}(\rho)$ admits a unique maximizer. Similarly, for any fixed $\pi_\psi \in \Delta_{|\mathcal{S}|}^{|\mathcal{B}|}$, the problem $\min_{\pi_\phi \in \Delta_{|\mathcal{S}|}^{|\mathcal{A}|}} V_{\mathcal{M}(x)}^{\pi_\phi, \pi_\psi}(\rho)$ admits a unique minimizer.*

*Proof.* By definition, we have

$$
\begin{aligned}
V_{\mathcal{M}(x)}^{\pi_\phi, \pi_\psi}(\rho) &= \mathop{\mathbb{E}}_{\substack{s_0 \sim \rho, a_t \sim \pi_\phi(\cdot|s_t) \\ b_t \sim \pi_\psi(\cdot|s_t) \\ s_{t+1} \sim P(\cdot|s_t, a_t, b_t)}} \left[ \sum_{t=0}^\infty \gamma^t \left[ r_x(s_t, a_t, b_t) + \tau_\phi \log \pi_\phi(a_t|s_t) - \tau_\psi \log \pi_\psi(b_t|s_t) \right] \right] \\
&= \mathop{\mathbb{E}}_{\substack{s_0 \sim \rho, a_t \sim \pi_\phi(\cdot|s_t) \\ s_{t+1} \sim P^{\pi_\psi}(\cdot|s_t, a_t)}} \left[ \sum_{t=0}^\infty \gamma^t \left[ r_x^{\pi_\psi}(s_t, a_t) + \tau_\phi \log \pi_\phi(a_t|s_t) + \tau_\psi H(\pi_\psi(\cdot|s_t)) \right] \right],
\end{aligned}
\tag{82}
$$

where $P^{\pi_\psi}(\cdot|s_t, a_t) = \mathbb{E}_{b_t \sim \pi_\psi(\cdot|s_t)}[P(\cdot|s_t, a_t, b_t)]$ and $r_x^{\pi_\psi}(s_t, a_t) = \mathbb{E}_{b_t \sim \pi_\psi(\cdot|s_t)}[r_x(s_t, a_t, b_t)]$.

Here $H(\pi_\psi(\cdot|s_t))$ does not depend on $\pi_\phi$. Since $\pi_\phi$ is the min-player, we can define a new reward model for $\pi_\phi$ as:

$$\tilde{r}_{(x,\psi)}(s, a) \triangleq -r_x^{\pi_\psi}(s, a) - \tau_\psi H(\pi_\psi(\cdot|s)).$$

Then for any given $x$ and $\psi$, $\mathcal{M}(x)$ is equivalent to a single-policy entropy-regularized MDP as $\mathcal{M}^{\pi_\psi}(x) = \{\mathcal{S}, \mathcal{A}, P^{\pi_\psi}, \tilde{r}_{(x,\psi)}, \gamma\}$, and $\pi_\phi$ is the policy that wants to maximize the soft value function:

$$J^{\pi_\psi}(x, \phi) = \mathop{\mathbb{E}}_{\substack{s_0 \sim \rho, a_t \sim \pi_\phi(\cdot|s_t) \\ s_{t+1} \sim P^{\pi_\psi}(\cdot|s_t, a_t)}} \left[ \sum_{t=0}^\infty \gamma^t \left[ \tilde{r}_{(x,\psi)}(s_t, a_t) - \tau_\phi \log \pi_\phi(a_t|s_t) \right] \right],
\tag{83}$$

which is equivalent to $-V_{\mathcal{M}(x)}^{\pi_\phi, \pi_\psi}(\rho)$.

Since we know that in the single-policy setting, the optimal policy for the soft value function exists and is unique (Geist et al., 2019), then for any given $x$ and $\psi$, the optimal $\pi_\phi$ that maximizes the soft value function $J^{\pi_\psi}(x, \phi)$ also exists and is unique. Therefore, the minimizer of the problem $\min_{\pi_\phi \in \Delta_{|\mathcal{S}|}^{|\mathcal{A}|}} V_{\mathcal{M}(x)}^{\pi_\phi, \pi_\psi}(\rho)$ for any fixed $x$ and $\pi_\psi$ also exists and is unique.

Similarly, $V_{\mathcal{M}(x)}^{\pi_\phi, \pi_\psi}(\rho)$ can also be written as

$$V_{\mathcal{M}(x)}^{\pi_\phi, \pi_\psi}(\rho) = \mathop{\mathbb{E}}_{\substack{s_0 \sim \rho, b_t \sim \pi_\psi(\cdot|s_t) \\ s_{t+1} \sim P^{\pi_\phi}(\cdot|s_t, b_t)}} \left[ \sum_{t=0}^\infty \gamma^t \left[ r_x^{\pi_\phi}(s_t, b_t) - \tau_\psi \log \pi_\psi(b_t|s_t) - \tau_\phi H(\pi_\phi(\cdot|s_t)) \right] \right],
\tag{84}$$

where $P^{\pi_\phi}(\cdot|s_t, b_t) = \mathbb{E}_{a_t \sim \pi_\phi(\cdot|s_t)}[P(\cdot|s_t, a_t, b_t)]$ and $r_x^{\pi_\phi}(s_t, b_t) = \mathbb{E}_{a_t \sim \pi_\phi(\cdot|s_t)}[r_x(s_t, a_t, b_t)]$.

Define

$$\tilde{r}_{(x,\phi)}(s, b) \triangleq r_x^{\pi_\phi}(s, b) - \tau_\phi H(\pi_\phi(\cdot|s)).$$

Then, for any fixed $x$ and $\pi_\phi$, by absorbing the effect of $\pi_\phi$ into the transition kernel and the reward, the optimization over $\pi_\psi$ can be reformulated as a single-policy entropy-regularized MDP $\mathcal{M}^{\pi_\phi}(x) = \{\mathcal{S}, \mathcal{B}, P^{\pi_\phi}, \tilde{r}_{(x,\phi)}, \gamma\}$. The corresponding objective for $\pi_\psi$ is to maximize the following soft value function:

$$J^{\pi_\phi}(x, \psi) = \mathop{\mathbb{E}}_{\substack{s_0 \sim \rho, b_t \sim \pi_\psi(\cdot|s_t) \\ s_{t+1} \sim P^{\pi_\phi}(\cdot|s_t, b_t)}} \left[ \sum_{t=0}^{\infty} \gamma^t \left[ \tilde{r}_{(x,\phi)}(s_t, b_t) - \tau_\psi \log \pi_\psi(b_t|s_t) \right] \right], \tag{85}$$

which is equivalent to $V^{\pi_\phi, \pi_\psi}_{\mathcal{M}(x)}(\rho)$. Then for any given $x$ and $\phi$, the optimal $\pi_\psi$ that maximizes $J^{\pi_\phi}(x, \psi)$ exists and is unique. Therefore, the maximizer of the problem $\max_{\pi_\psi \in \Delta^{|\mathcal{B}|}_{|\mathcal{S}|}} V^{\pi_\phi, \pi_\psi}_{\mathcal{M}(x)}(\rho)$ for any $x$ and $\phi$ also exists and is unique. $\qquad\square$

Denote the best-response policies as $\pi_\phi^*(x, \psi) = \arg\min_{\pi_\phi \in \Delta^{|\mathcal{A}|}_{|\mathcal{S}|}} V^{\pi_\phi, \pi_\psi}_{\mathcal{M}(x)}(\rho)$, $\pi_\psi^*(x, \phi) = \arg\max_{\pi_\psi \in \Delta^{|\mathcal{B}|}_{|\mathcal{S}|}} V^{\pi_\phi, \pi_\psi}_{\mathcal{M}(x)}(\rho)$. Denote the corresponding best-response parameters as $\phi^*(x, \psi) = \arg\min_\phi J(x, \phi, \psi)$, and $\psi^*(x, \phi) = \arg\max_\psi J(x, \phi, \psi)$. Then the optimal softmax parameters satisfy

$$\phi^*(x, \psi) = \{\phi \in \mathbb{R}^{|\mathcal{S}||\mathcal{A}|} : \pi_\phi = \pi_\phi^*(x, \psi)\} = \{\log \pi_\phi^*(x, \psi) + \mathrm{diag}(c)\mathbf{1}_{|\mathcal{S}| \times |\mathcal{A}|} : c \in \mathbb{R}^{|\mathcal{S}|}\},$$

and

$$\psi^*(x, \phi) = \{\psi \in \mathbb{R}^{|\mathcal{S}||\mathcal{B}|} : \pi_\psi = \pi_\psi^*(x, \phi)\} = \{\log \pi_\psi^*(x, \phi) + \mathrm{diag}(c)\mathbf{1}_{|\mathcal{S}| \times |\mathcal{B}|} : c \in \mathbb{R}^{|\mathcal{S}|}\},$$

where $\mathbf{1}_{|\mathcal{S}| \times |\mathcal{A}|}$ is a $|\mathcal{S}| \times |\mathcal{A}|$ matrix with all entries being 1.

Similarly, we also define the best-response state-visitation distribution for policy $\pi_\phi$ as $d_\rho^{\pi_\phi, \dagger}(s) \triangleq d_\rho^{\pi_\phi, \pi_\psi}(s)\big|_{\pi_\psi \in \pi_\psi^*(x, \phi)}$, and the best-response state-visitation distribution for policy $\pi_\psi$ as $d_\rho^{\dagger, \pi_\psi}(s) \triangleq d_\rho^{\pi_\phi, \pi_\psi}(s)\big|_{\pi_\phi \in \pi_\phi^*(x, \psi)}$.

By Proposition 1, we know that the optimal policy pair $(\pi_\phi^*, \pi_\psi^*)$ exists and is unique, therefore we can also define the optimal state-visitation distribution under policies $\pi_\phi^*$ and $\pi_\psi^*$ as $d_\rho^*(s) \triangleq d_\rho^{\pi_\phi^*, \pi_\psi^*}(s)$.

Denote $J^*(x) = \min_\phi \max_\psi J(x, \phi, \psi) = V^\star_{\mathcal{M}(x)}(\rho)$. Then we have the following lemma on the PŁ condition of the best-response value function $J_1(x, \phi) = \max_\psi J(x, \phi, \psi)$ and $J_2(x, \psi) = \min_\phi J(x, \phi, \psi)$.

**Lemma C.5** (Non-uniform PŁ condition on best-response value function). *Suppose the initial distribution $\rho$ has full support, i.e., $\min_s \rho(s) > 0$. Then for any $x$, the best-response value function $J_1(x, \phi) = \max_\psi J(x, \phi, \psi)$ satisfies the following condition in $\phi$:*

$$\frac{1}{2}\|\nabla_\phi J_1(x, \phi)\|_2^2 \geq \mu_1(x, \phi) \left( J_1(x, \phi) - J^*(x) \right),$$

*where*

$$\mu_1(x, \phi) = \frac{\tau_\phi}{|\mathcal{S}|} \min_s \rho(s) \min_{s,a} \pi_\phi^2(a|s) \min_s \frac{d_\rho^{\pi_\phi, \pi_\psi^*(x, \phi)}(s)}{d_\rho^{\dagger, \pi_\psi^*(x, \phi)}(s)},$$

*and $\pi_\psi^*(x, \phi) = \arg\max_{\pi_\psi} V^{\pi_\phi, \pi_\psi}_{\mathcal{M}(x)}(\rho)$.*

*Similarly, for any $x$, the best-response value function $J_2(x, \psi) = \min_\phi J(x, \phi, \psi)$ satisfies the following condition in $\psi$:*

$$\frac{1}{2}\|\nabla_\psi J_2(x, \psi)\|_2^2 \geq \mu_2(x, \psi) \left( J^*(x) - J_2(x, \psi) \right),$$

*where*

$$\mu_2(x, \psi) = \frac{\tau_\psi}{|\mathcal{S}|} \min_s \rho(s) \min_{s,b} \pi_\psi^2(b|s) \min_s \frac{d_\rho^{\pi_\phi^*(x, \psi), \pi_\psi}(s)}{d_\rho^{\pi_\phi^*(x, \psi), \dagger}(s)},$$

*and $\pi_\phi^*(x, \psi) = \arg\min_{\pi_\phi} V^{\pi_\phi, \pi_\psi}_{\mathcal{M}(x)}(\rho)$.*

*Proof.* Denote $V(x, \pi_\phi, \pi_\psi) = V^{\pi_\phi, \pi_\psi}_{\mathcal{M}(x)}(\rho)$ for notational simplicity if there is no confusion.

Define a new reward model for $\pi_\phi$ as

$$\tilde{r}_{(x,\psi)}(s,a) = -r_x^{\pi_\psi}(s,a) - \tau_\psi H(\pi_\psi(\cdot|s)),$$

and the transition probability as $P^{\pi_\psi}(\cdot|s_t,a_t) = \mathbb{E}_{b_t \sim \pi_\psi(\cdot|s_t)}[P(\cdot|s_t,a_t,b_t)]$.

Then, for any fixed $x$ and $\pi_\psi$, the original minimization over $\pi_\phi$ can be equivalently reformulated as a maximization problem in the single-policy entropy-regularized MDP

$$\mathcal{M}^{\pi_\psi}(x) = \{\mathcal{S}, \mathcal{A}, P^{\pi_\psi}, \tilde{r}_{(x,\psi)}, \gamma\}.$$

Under the transformed reward $\tilde{r}_{(x,\psi)}$, the corresponding objective for $\pi_\phi$ is to maximize

$$J^{\pi_\psi}(x,\phi) = \mathbb{E}_{\substack{s_0 \sim \rho, a_t \sim \pi_\phi(\cdot|s_t) \\ s_{t+1} \sim P^{\pi_\psi}(\cdot|s_t,a_t)}} \left[ \sum_{t=0}^{\infty} \gamma^t \left[ \tilde{r}_{(x,\psi)}(s_t,a_t) - \tau_\phi \log \pi_\phi(a_t|s_t) \right] \right], \tag{86}$$

which equals $-V_{\mathcal{M}(x)}^{\pi_\phi,\pi_\psi}(\rho)$. Then the state-visitation distribution under policy $\pi_\phi$ in this MDP is exactly $d_\rho^{\pi_\phi,\pi_\psi}$.

By Lemma 15 in Mei et al. (2020), we have for any $x$, and any given $\psi$,

$$\frac{1}{2}\|\nabla_\phi J^{\pi_\psi}(x,\phi)\|_2^2 \geq \mu^{\pi_\psi}(x,\phi) \left( \max_\phi J^{\pi_\psi}(x,\phi) - J^{\pi_\psi}(x,\phi) \right), \tag{87}$$

which is equivalent to

$$\frac{1}{2}\|\nabla_\phi V(x,\pi_\phi,\pi_\psi)\|_2^2 \geq \mu^{\pi_\psi}(x,\phi) \left( V(x,\pi_\phi,\pi_\psi) - \min_\phi V(x,\pi_\phi,\pi_\psi) \right), \tag{88}$$

where

$$\mu^{\pi_\psi}(x,\phi) := \frac{\tau_\phi}{|\mathcal{S}|} \min_s \rho(s) \min_{s,a} \pi_\phi^2(a|s) \min_s \frac{d_\rho^{\pi_\phi,\pi_\psi}(s)}{d_\rho^{\dagger,\pi_\psi}(s)}.$$

Now for an arbitrary $\pi_\phi$, we have $\pi_\psi^*(x,\phi) = \arg\max_{\pi_\psi} V(x,\pi_\phi,\pi_\psi)$ is unique. Thus, by Lemma A.2, we can obtain

$$\begin{aligned}
\nabla_\phi J_1(x,\phi) &= \nabla_\phi \max_\psi J(x,\phi,\psi) \\
&= \nabla_\phi \max_{\pi_\psi} V(x,\pi_\phi,\pi_\psi) \\
&= \nabla_\phi V(x,\pi_\phi,\pi_\psi^*(x,\phi)).
\end{aligned} \tag{89}$$

Then we can get

$$\begin{aligned}
\frac{1}{2}\|\nabla_\phi J_1(x,\phi)\|_2^2 &= \frac{1}{2}\|\nabla_\phi V(x,\pi_\phi,\pi_\psi^*(x,\phi))\|_2^2 \\
&\geq \mu^{\pi_\psi^*(x,\pi_\phi)}(x,\pi_\phi) \left( V(x,\pi_\phi,\pi_\psi^*(x,\phi)) - \min_{\phi'} V(x,\pi_{\phi'},\pi_\psi^*(x,\phi)) \right) \\
&= \mu_1(x,\phi) \left( J_1(x,\phi) - V(x,\tilde{\pi}_\phi^*,\pi_\psi^*(x,\phi)) \right), 
\end{aligned} \tag{90}$$

where $\tilde{\pi}_\phi^* = \arg\min_{\pi_\phi'} V(x,\pi_\phi',\pi_\psi^*(x,\phi))$ for the given $\phi$ and

$$\mu_1(x,\phi) \triangleq \mu^{\pi_\psi^*(x,\pi_\phi)}(x,\pi_\phi) = \frac{\tau_\phi}{|\mathcal{S}|} \min_s \rho(s) \min_{s,a} \pi_\phi^2(a|s) \min_s \frac{d_\rho^{\pi_\phi,\pi_\psi^*(x,\phi)}(s)}{d_\rho^{\dagger,\pi_\psi^*(x,\phi)}(s)}.$$

Consequently, for any given $\phi$, by Proposition B.1, we can get

$$
\begin{aligned}
\min_{\phi'} J_1(x, \phi') = \min_{\pi'_\phi} \max_{\pi_\psi} V(x, \pi'_\phi, \pi_\psi) \\
= \max_{\pi_\psi} \min_{\pi'_\phi} V(x, \pi'_\phi, \pi_\psi) \\
\geq \min_{\pi'_\phi} V(x, \pi'_\phi, \pi^*_\psi(x, \phi)) \\
= V(x, \tilde{\pi}^*_\phi, \pi^*_\psi(x, \pi_\phi)).
\end{aligned}
\tag{91}
$$

On the other hand, by definition, we have

$$
\min_{\phi'} J_1(x, \phi') = \min_{\pi'_\phi} \max_{\pi_\psi} V(x, \pi'_\phi, \pi_\psi) = J^*(x).
\tag{92}
$$

Therefore, we obtain

$$
\begin{aligned}
\frac{1}{2}\|\nabla_\phi J_1(x, \pi_\phi)\|_2^2 \geq \mu_1(x, \phi)\left(J_1(x, \phi) - V(x, \tilde{\pi}^*_\phi, \pi^*_\psi(x, \phi))\right) \\
\geq \mu_1(x, \phi)\left(J_1(x, \phi) - J^*(x)\right).
\end{aligned}
\tag{93}
$$

Similarly, we have for any $x$

$$
\frac{1}{2}\|\nabla_\psi J_2(x, \psi)\|_2^2 \geq \mu_2(x, \psi)\left(J^*(x) - J_2(x, \psi)\right),
\tag{94}
$$

where

$$
\mu_2(x, \psi) = \frac{\tau_\psi}{|\mathcal{S}|} \min_s \rho(s) \min_{s,b} \pi_\psi^2(b|s) \min_s \frac{d_\rho^{\pi^*_\phi(x,\psi), \pi_\psi}(s)}{d_\rho^{\pi^*_\phi(x,\psi),\dagger}(s)}.
$$

and $\pi^*_\phi(x, \psi) = \arg\min_{\pi_\phi} V(x, \pi_\phi, \pi_\psi)$.

$\square$

**Lemma C.6** (Lemma 1 in the main text). *Let $\theta = (\phi, \psi) \in \mathbb{R}^{|\mathcal{S}||\mathcal{A}|+|\mathcal{S}||\mathcal{B}|}$ denote the concatenation of the policy parameters $\phi$ and $\psi$. Suppose the initial distribution $\rho$ has full support, i.e., $\min_s \rho(s) > 0$. Then, for any $x$ and any $\theta$, $g(x, \theta)$ satisfies the following condition:*

$$
\frac{1}{2}\|\nabla_\theta g(x, \theta)\|_2^2 \geq \mu(x, \theta)g(x, \theta),
\tag{95}
$$

*where*

$$
\mu(x, \theta) = (1-\gamma)\frac{\min\{\tau_\phi, \tau_\psi\}}{|\mathcal{S}|} \min_s \rho^2(s) \min\{\min_{s,a} \pi_\phi^2(a|s), \min_{s,b} \pi_\psi^2(b|s)\} > 0.
$$

*Proof.* By definition, we have

$$
\nabla_\theta g(x, \theta) = \begin{bmatrix} \nabla_\phi J_1(x, \phi) \\ -\nabla_\psi J_2(x, \psi) \end{bmatrix},
\tag{96}
$$

Hence, by Lemma C.5, it follows that

$$
\begin{aligned}
\frac{1}{2}\|\nabla_\theta g(x, \theta)\|_2^2 &= \frac{1}{2}\|\nabla_\phi J_1(x, \phi)\|_2^2 + \frac{1}{2}\|\nabla_\psi J_2(x, \psi)\|_2^2 \\
&\geq \left(\mu_1(x, \phi)(J_1(x, \phi) - J^*(x)) + \mu_2(x, \psi)(J^*(x) - J_2(x, \psi))\right) \\
&\geq \min\{\mu_1(x, \phi), \mu_2(x, \psi)\}\left(J_1(x, \phi) - J_2(x, \psi)\right) \\
&= \min\{\mu_1(x, \phi), \mu_2(x, \psi)\}g(x, \theta).
\end{aligned}
\tag{97}
$$

Since for any $\phi, \psi$, and for any state $s$,

$$d_\rho^{\pi_\phi, \pi_\psi}(s) = (1 - \gamma) \sum_{t=0}^{\infty} \gamma^t \Pr(s_t = s | s_0 \sim \rho, \pi_\phi, \pi_\psi, \mathcal{M}(x))$$

$$\geq (1 - \gamma) \min_s \rho(s) > 0,$$

then for any $\phi, \phi', \psi, \psi'$,

$$\min_s \frac{d_\rho^{\pi_\phi, \pi_\psi}(s)}{d_\rho^{\pi_{\phi'}, \pi_{\psi'}}(s)} \geq (1 - \gamma) \min_s \rho(s) > 0.$$

Therefore, taking

$$\mu(x, \theta) = (1 - \gamma) \frac{\min\{\tau_\phi, \tau_\psi\}}{|\mathcal{S}|} \min_s \rho^2(s) \min\{\min_{s,a} \pi_\phi^2(a|s), \min_{s,b} \pi_\psi^2(b|s)\},$$

we have

$$\frac{1}{2} \|\nabla_\theta g(x, \theta)\|_2^2 \geq \mu(x, \theta) g(x, \theta). \tag{98}$$

$\square$

## C.3. Proofs of Other Technical Lemmas

**Lemma C.7.** *Under Assumption 1, there exists a constant $\mu_J > 0$ such that the joint value function $J(x, \phi, \psi)$ satisfies the following PŁ conditions for any $x$, $\phi$ and $\psi$:*

$$\frac{1}{2} \|\nabla_\phi J(x, \phi, \psi)\|_2^2 \geq \mu_J \left( J(x, \phi, \psi) - J_2(x, \psi) \right),$$

$$\frac{1}{2} \|\nabla_\psi J(x, \phi, \psi)\|_2^2 \geq \mu_J \left( J_1(x, \phi) - J(x, \phi, \psi) \right).$$

*Proof.* From Eq (88) in Lemma C.5, we have for any $x$ and $\psi$,

$$\frac{1}{2} \|\nabla_\phi V(x, \pi_\phi, \pi_\psi)\|_2^2 \geq \mu^{\pi_\phi}(x, \phi) \left( V(x, \pi_\phi, \pi_\psi) - \min_{\phi'} V(x, \pi_{\phi'}, \pi_\psi) \right), \tag{99}$$

which is equivalent to

$$\frac{1}{2} \|\nabla_\phi J(x, \phi, \psi)\|_2^2 \geq \mu^{\pi_\phi}(x, \phi) \left( J(x, \phi, \psi) - J_2(x, \psi) \right), \tag{100}$$

where

$$\mu^{\pi_\psi}(x, \phi) = \frac{\tau_\phi}{|\mathcal{S}|} \min_s \rho(s) \min_{s,a} \pi_\phi^2(a|s) \min_s \frac{d_\rho^{\pi_\phi, \pi_\psi}(s)}{d_\rho^{\dagger, \pi_\psi}(s)}$$

$$\geq \frac{\tau_\phi}{|\mathcal{S}|} \min_s \rho(s) \left( \min_{s,a} \pi_\phi(a|s) \right)^2 (1 - \gamma) \min_s \rho(s)$$

$$\geq (1 - \gamma) \frac{\min\{\tau_\phi, \tau_\psi\}}{|\mathcal{S}|} \delta_\rho^2 \delta_\pi^2. \tag{101}$$

Similarly, for any $x$ and $\phi$, we have

$$\frac{1}{2} \|\nabla_\psi J(x, \phi, \psi)\|_2^2 \geq \mu^{\pi_\psi}(x, \psi) \left( J_1(x, \phi) - J(x, \phi, \psi) \right), \tag{102}$$

where

$$\mu^{\pi_\psi}(x, \psi) = \frac{\tau_\psi}{|\mathcal{S}|} \min_s \rho(s) \min_{s,b} \pi_\psi^2(b|s) \min_s \frac{d_\rho^{\pi_\phi, \pi_\psi}(s)}{d_\rho^{\pi_\phi, \dagger}(s)}$$

$$\geq (1-\gamma) \frac{\min\{\tau_\phi, \tau_\psi\}}{|\mathcal{S}|} \delta_\rho^2 \delta_\pi^2. \tag{103}$$

Thus, by taking $\mu_J = (1-\gamma) \frac{\min\{\tau_\phi, \tau_\psi\}}{|\mathcal{S}|} \delta_\rho^2 \delta_\pi^2 > 0$, we complete the proof. $\qquad\square$

**Lemma C.8.** *Under Assumption 1, the NI function $g(x, \phi, \psi)$ is $\mu_g$-PŁ in $(\phi, \psi)$ for some constant $\mu_g > 0$.*

*Proof.* By Lemma C.6 and Assumption 1, we have for any $x$ and $\theta$,

$$\mu(x, \theta) = (1-\gamma) \frac{\min\{\tau_\phi, \tau_\psi\}}{|\mathcal{S}|} \min_s \rho^2(s) \min\{\min_{s,a} \pi_\phi^2(a|s), \min_{s,b} \pi_\psi^2(b|s)\}$$

$$\geq (1-\gamma) \frac{\min\{\tau_\phi, \tau_\psi\}}{|\mathcal{S}|} \delta_\rho^2 \delta_\pi^2. \tag{104}$$

Taking $\mu_g = (1-\gamma) \frac{\min\{\tau_\phi, \tau_\psi\}}{|\mathcal{S}|} \delta_\rho^2 \delta_\pi^2 > 0$, we complete the proof. $\qquad\square$

**Lemma C.9.** *Under Assumptions 1 and 2, for any $x$ and $\psi$, define $\pi_\phi^*(x, \psi) \triangleq \arg\min_{\pi_\phi} V_{\mathcal{M}(x)}^{\pi_\phi, \pi_\psi}(\rho)$. Then there exists a constant $C_{\pi,1}$ such that for any $x_1$, $x_2$, $\psi_1$, and $\psi_2$, we have*

$$\|\pi_\phi^*(x_1, \psi_1) - \pi_\phi^*(x_2, \psi_2)\| \leq C_{\pi,1} (\|x_1 - x_2\| + \|\psi_1 - \psi_2\|).$$

*Similarly, for any $x$ and $\phi$, define $\pi_\psi^*(x, \phi) \triangleq \arg\max_{\pi_\psi} V_{\mathcal{M}(x)}^{\pi_\phi, \pi_\psi}(\rho)$. Then there exists a constant $C_{\pi,2}$ such that for any $x_1$, $x_2$, $\phi_1$, and $\phi_2$, we have*

$$\|\pi_\psi^*(x_1, \phi_1) - \pi_\psi^*(x_2, \phi_2)\| \leq C_{\pi,2} (\|x_1 - x_2\| + \|\phi_1 - \phi_2\|).$$

*Proof.* By Lemma C.4, we know that $\pi_\phi^*(x, \psi)$ is the unique optimal policy for a single-policy entropy-regularized MDP $\mathcal{M}^{\pi_\psi}(x) = \{\mathcal{S}, \mathcal{A}, \widetilde{r}_{(x,\psi)}, \widetilde{\mathcal{P}}_\psi, \gamma, \tau_\phi\}$ for any given $x$ and $\psi$, where $\widetilde{r}_{(x,\psi)}(s, a)$ is defined as

$$\widetilde{r}_{(x,\psi)}(s, a) \triangleq -\mathbb{E}_{b \sim \psi(\cdot|s)} [r_x(s, a, b) - \tau_\psi \log \pi_\psi(b|s)],$$

and $\widetilde{\mathcal{P}}_\psi$ is defined as

$$\widetilde{\mathcal{P}}_\psi(s'|s, a) \triangleq \mathbb{E}_{b \sim \psi(\cdot|s)} [\mathcal{P}(s'|s, a, b)].$$

Moreover, we have the following properties for $\widetilde{r}_{(x,\psi)}$ and $\widetilde{\mathcal{P}}_\psi(s'|s, a)$:

$$|\widetilde{r}_{(x,\psi)}(s, a)| \leq B_r + \tau_\psi \log |\mathcal{B}|, \tag{105}$$

and

$$\sup_{s,a} \|\widetilde{\mathcal{P}}_{\psi_1}(s'|s, a) - \widetilde{\mathcal{P}}_{\psi_2}(s'|s, a)\|_1$$

$$= \sup_{s,a} \sum_{s'} \left| \sum_b \pi_{\psi_1}(b|s)\mathcal{P}(s'|s, a, b) - \pi_{\psi_2}(b|s)\mathcal{P}(s'|s, a, b) \right|$$

$$= \sup_{s,a} \sum_{s'} \left| \sum_b (\pi_{\psi_1}(b|s) - \pi_{\psi_2}(b|s))\mathcal{P}(s'|s, a, b) \right|$$

$$\leq \sup_{s,a} \sum_b |\pi_{\psi_1}(b|s) - \pi_{\psi_2}(b|s)| \sum_{s'} \mathcal{P}(s'|s, a, b)$$

$$= \sup_s \|\pi_{\psi_1}(\cdot|s) - \pi_{\psi_2}(\cdot|s)\|_1, \tag{106}$$

and

$$\sup_{s,a} |\widetilde{r}_{(x_1,\psi_1)}(s,a) - \widetilde{r}_{(x_2,\psi_2)}(s,a)|$$

$$\leq \sup_{s,a} \left| \sum_b \pi_\psi(b|s) \left( r_{x_1}(s,a,b) - r_{x_2}(s,a,b) \right) \right| + \sup_{s,a} \left| \sum_b \left( \pi_{\psi_1}(b|s) - \pi_{\psi_2}(b|s) \right) r_{x_2}(s,a,b) \right|$$

$$+ \sup_{s,a} |\tau_\psi \left( H(\pi_{\psi_1}(\cdot|s)) - H(\pi_{\psi_2}(\cdot|s)) \right)|$$

$$\leq C_r \|x_1 - x_2\| + B_r \sup_s \|\pi_{\psi_1}(\cdot|s) - \pi_{\psi_2}(\cdot|s)\|_1 + \tau_\psi \sup_s |H(\pi_{\psi_1}(\cdot|s)) - H(\pi_{\psi_2}(\cdot|s))|. \tag{107}$$

For any $s \in \mathcal{S}$, any $\pi_1, \pi_2 \in \Delta_n$ and $\pi_1, \pi_2 \geq \delta_\pi \mathbf{1}$, the entropy function $H(\cdot)$ is Lipschitz continuous with Lipschitz constant $H_{\delta_\pi} = \max\{1, |1 + \log \delta_\pi|\}$, which means

$$|H(\pi_{\psi_1}(\cdot|s)) - H(\pi_{\psi_2}(\cdot|s))| \leq H_{\delta_\pi} \|\pi_{\psi_1}(\cdot|s) - \pi_{\psi_2}(\cdot|s)\|_1. \tag{108}$$

And we also have for softmax parameterized policy $\pi_\psi$ and any state $s$,

$$\|\pi_{\psi_1}(\cdot|s) - \pi_{\psi_2}(\cdot|s)\|_1 \leq \sqrt{|\mathcal{B}|} \|\pi_{\psi_1}(\cdot|s) - \pi_{\psi_2}(\cdot|s)\| \leq \frac{\sqrt{|\mathcal{B}|}}{2} \|\psi_1(s) - \psi_2(s)\| \leq \frac{\sqrt{|\mathcal{B}|}}{2} \|\psi_1 - \psi_2\|. \tag{109}$$

Then we obtain

$$\sup_{s,a} |\widetilde{r}_{(x_1,\psi_1)}(s,a) - \widetilde{r}_{(x_2,\psi_2)}(s,a)|$$

$$\leq \frac{\sqrt{|\mathcal{B}|}}{2} \left( B_r + \tau_\psi H_{\delta_\pi} \right) \|\psi_1 - \psi_2\| + C_r \|x_1 - x_2\|$$

$$\leq \widetilde{C}_r \|(x_1, \psi_1) - (x_2, \psi_2)\|, \tag{110}$$

where $\widetilde{C}_r = \sqrt{2} \max\{C_r, \frac{\sqrt{|\mathcal{B}|}}{2}(B_r + \tau_\psi H_{\delta_\pi})\}$, and

$$\sup_{s,a} \|\widetilde{\mathcal{P}}_{\psi_1}(s'|s,a) - \widetilde{\mathcal{P}}_{\psi_2}(s'|s,a)\|_1 \leq \frac{\sqrt{|\mathcal{B}|}}{2} \|\psi_1 - \psi_2\| \leq \frac{\sqrt{|\mathcal{B}|}}{2} \|(x_1, \psi_1) - (x_2, \psi_2)\|. \tag{111}$$

Now we will show that $\pi_\phi^*(x, \psi)$ is Lipschitz continuous in $\psi$. Let $Q^*(x, \psi) \in \mathbb{R}^{|\mathcal{S}| \times |\mathcal{A}|}$ be the optimal soft Q-value for the MDP $\widetilde{\mathcal{M}}(x, \psi)$. Since the assumptions required for Lemma C.1 are satisfied, by Lemma C.1, there exists a constant $C_Q$ such that for any $x$ and $\psi_1, \psi_2$,

$$\|Q^*(x, \psi_1) - Q^*(x, \psi_2)\| \leq C_Q \|(x, \psi_1) - (x, \psi_2)\|.$$

And the optimal policy $\pi_\phi^*(x, \psi)$ can be represented as

$$\pi_\phi^*(a|s; x, \psi) = \frac{\exp(Q_{sa}^*(x, \psi)/\tau_\phi)}{\sum_{a'} \exp(Q_{sa'}^*(x, \psi)/\tau_\phi)}, \quad \forall s \in \mathcal{S}, a \in \mathcal{A}.$$

Therefore, we have for any $x$ and $\psi_1, \psi_2$,

$$\|\pi_\phi^*(x_1, \psi_1) - \pi_\phi^*(x_2, \psi_2)\|$$

$$\leq \frac{1}{2\tau_\phi} \|Q^*(x_1, \psi_1) - Q^*(x_2, \psi_2)\|$$

$$\leq \frac{C_Q}{2\tau_\phi} \|(x_1, \psi_1) - (x_2, \psi_2)\|$$

$$\leq \frac{C_Q}{2\tau_\phi} \left( \|x_1 - x_2\| + \|\psi_1 - \psi_2\| \right), \tag{112}$$

which completes the proof for the first part. The second part can be proved similarly for the opponent policy $\psi$. $\square$

**Lemma C.10.** *Under Assumptions 1 and 2, for any $\phi, \psi$, define $\phi^*(x, \psi) = \arg\min_\phi J(x, \phi, \psi)$ and $\psi^*(x, \phi) = \arg\max_\psi J(x, \phi, \psi)$, then we have for any $x_1, x_2, \psi_1, \psi_2, \phi_1, \phi_2$, there exists a constant $C_\pi > 0$ such that*

$$\text{dist}\left(\phi^*(x_1, \psi_1), \phi^*(x_2, \psi_2)\right) \leq C_\pi \left(\|x_1 - x_2\| + \|\psi_1 - \psi_2\|\right),$$
$$\text{dist}\left(\psi^*(x_1, \phi_1), \psi^*(x_2, \phi_2)\right) \leq C_\pi \left(\|x_1 - x_2\| + \|\phi_1 - \phi_2\|\right).$$

*Proof.* Since we know that for any given $x$ and $\psi$, $\pi_\phi^*(x, \psi) = \arg\min_{\pi_\phi} V_{\mathcal{M}(x)}^{\pi_\phi, \pi_\psi}(\rho)$ is unique, then we have

$$\phi^*(x, \psi) = \{\phi : \pi_\phi = \pi_\phi^*(x, \psi)\} = \{\log \pi_\phi^*(x, \psi) + \text{diag}(c)\mathbf{1}_{|\mathcal{S}| \times |\mathcal{A}|} : c \in \mathbb{R}^{|\mathcal{S}|}\}.$$

Here we consider $\log \pi$ and $\phi$ as a long vector in $\mathbb{R}^{|\mathcal{S}||\mathcal{A}|}$, and $\mathbf{1} \in \mathbb{R}^{|\mathcal{S}||\mathcal{A}|}$ is a vector with all entries being 1. Thus, for any $x_1, x_2, \psi_1$ and $\psi_2$, the distance between the two sets $\phi^*(x_1, \psi_1)$ and $\phi^*(x_2, \psi_2)$ can be calculated as

$$
\begin{aligned}
&\text{dist}\left(\phi^*(x_1, \psi_1), \phi^*(x_2, \psi_2)\right) \\
&= \inf_{\phi_1 \in \phi^*(x_1, \psi_1), \phi_2 \in \phi^*(x_2, \psi_2)} \|\phi_1 - \phi_2\| \\
&= \inf_{c_1 \in \mathbb{R}^{|\mathcal{S}|}, c_2 \in \mathbb{R}^{|\mathcal{S}|}} \|\log \pi_\phi^*(x_1, \psi_1) + \text{diag}(c_1)\mathbf{1}_{|\mathcal{S}| \times |\mathcal{A}|} - \log \pi_\phi^*(x_2, \psi_2) - \text{diag}(c_2)\mathbf{1}_{|\mathcal{S}| \times |\mathcal{A}|}\| \\
&= \inf_{c \in \mathbb{R}^{|\mathcal{S}|}} \|\log \pi_\phi^*(x_1, \psi_1) - \log \pi_\phi^*(x_2, \psi_2) + \text{diag}(c)\mathbf{1}_{|\mathcal{S}| \times |\mathcal{A}|}\| \\
&\leq \|\log \pi_\phi^*(x_1, \psi_1) - \log \pi_\phi^*(x_2, \psi_2)\| \\
&\leq \frac{1}{\delta_\pi} \|\pi_\phi^*(x_1, \psi_1) - \pi_\phi^*(x_2, \psi_2)\| \\
&\leq \frac{C_{\pi,1}}{\delta_\pi} \left(\|x_1 - x_2\| + \|\psi_1 - \psi_2\|\right).
\end{aligned}
\tag{113}
$$

Similarly, we can prove that for any $x_1, x_2, \phi_1$ and $\phi_2$,

$$\text{dist}\left(\psi^*(x_1, \phi_1), \psi^*(x_2, \phi_2)\right) \leq \frac{C_{\pi,2}}{\delta_\pi} \left(\|x_1 - x_2\| + \|\phi_1 - \phi_2\|\right). \tag{114}$$

By taking $C_\pi = \max\left\{\frac{C_{\pi,1}}{\delta_\pi}, \frac{C_{\pi,2}}{\delta_\pi}\right\}$, we complete the proof. $\qquad \square$

**Lemma C.11.** *Under Assumption 2, the approximated gap function $\tilde{g}(x, \phi, \psi, \tilde{\phi}, \tilde{\psi})$ is $C_{\tilde{g}}$-Lipschitz continuous, $L_{\tilde{g},1}$-smooth, and has an $L_{\tilde{g},2}$-Lipschitz continuous Hessian, for some constants $C_{\tilde{g}} > 0$, $L_{\tilde{g},1} > 0$, and $L_{\tilde{g},2} > 0$.*

*Proof.* Since the approximated gap function $\tilde{g}(x, \phi, \psi, \tilde{\phi}, \tilde{\psi})$ is defined as

$$\tilde{g}(x, \phi, \psi, \tilde{\phi}, \tilde{\psi}) = J(x, \phi, \tilde{\psi}) - J(x, \tilde{\phi}, \psi), \tag{115}$$

then by Assumption 2, $J(x, \phi, \psi)$ is $C_J$-Lipschitz continuous, $L_{J,1}$-Lipschitz smooth and $L_{J,2}$-Hessian Lipschitz continuous in $(x, \phi, \psi)$, thus the proof is straightforward by taking $C_{\tilde{g}} = 2C_J$, $L_{\tilde{g},1} = 2L_{J,1}$ and $L_{\tilde{g},2} = 2L_{J,2}$. $\qquad \square$

**Lemma C.12** (Lemma 2 in the main text). *Under Assumptions 1 and 2, the gap function $g(x, \phi, \psi)$ is $C_g$-Lipschitz continuous, $L_{g,1}$-Lipschitz smooth, and $L_{g,2}$-Hessian Lipschitz continuous in $(x, \phi, \psi)$, for some constants $C_g$, $L_{g,1}$ and $L_{g,2}$.*

*Proof.* By Assumption 2, we know that $J(x, \phi, \psi)$ is $C_J$-Lipschitz continuous, $L_{J,1}$-Lipschitz smooth and $L_{J,2}$-Hessian Lipschitz continuous in $(x, \phi, \psi)$. Then for any $x_1, x_2, \phi_1, \phi_2, \psi_1, \psi_2$,

$$
\begin{aligned}
|J(x_1, \phi_1, \psi_1) - J(x_2, \phi_2, \psi_2)| &\leq C_J \left(\|x_1 - x_2\| + \|\phi_1 - \phi_2\| + \|\psi_1 - \psi_2\|\right), \\
\|\nabla J(x_1, \phi_1, \psi_1) - \nabla J(x_2, \phi_2, \psi_2)\| &\leq L_{J,1} \left(\|x_1 - x_2\| + \|\phi_1 - \phi_2\| + \|\psi_1 - \psi_2\|\right), \\
\|\nabla^2 J(x_1, \phi_1, \psi_1) - \nabla^2 J(x_2, \phi_2, \psi_2)\| &\leq L_{J,2} \left(\|x_1 - x_2\| + \|\phi_1 - \phi_2\| + \|\psi_1 - \psi_2\|\right).
\end{aligned}
\tag{116}
$$

Consequently, we can obtain

$$
\begin{aligned}
& J_1(x_1, \phi_1) - J_1(x_2, \phi_2) \\
& = \max_{\psi_1} J(x_1, \phi_1, \psi_1) - \max_{\psi_2} J(x_2, \phi_2, \psi_2) \\
& = J(x_1, \phi_1, \psi') - \max_{\psi_2} J(x_2, \phi_2, \psi_2), \quad \psi' \in \psi^*(x_1, \phi_1) \\
& \le J(x_1, \phi_1, \psi') - J(x_2, \phi_2, \psi') \\
& \le C_J \left( \|x_1 - x_2\| + \|\phi_1 - \phi_2\| \right),
\end{aligned}
\tag{117}
$$

and similarly

$$
\begin{aligned}
& J_2(x_1, \psi_1) - J_2(x_2, \psi_2) \\
& = \min_{\phi_1} J(x_1, \phi_1, \psi_1) - \min_{\phi_2} J(x_2, \phi_2, \psi_2) \\
& \le J(x_1, \phi', \psi_1) - J(x_2, \phi', \psi_2), \quad \phi' \in \phi^*(x_2, \psi_2) \\
& \le C_J \left( \|x_1 - x_2\| + \|\psi_1 - \psi_2\| \right).
\end{aligned}
\tag{118}
$$

Therefore, we have

$$
\begin{aligned}
& |g(x_1, \phi_1, \psi_1) - g(x_2, \phi_2, \psi_2)| \\
& = |J_1(x_1, \phi_1) - J_2(x_1, \psi_1) - (J_1(x_2, \phi_2) - J_2(x_2, \psi_2))| \\
& \le |J_1(x_1, \phi_1) - J_1(x_2, \phi_2)| + |J_2(x_1, \psi_1) - J_2(x_2, \psi_2)| \\
& \le 2C_J \left( \|x_1 - x_2\| + \|\phi_1 - \phi_2\| + \|\psi_1 - \psi_2\| \right),
\end{aligned}
\tag{119}
$$

which proves the Lipschitz continuity of $g(x, \phi, \psi)$.

Next, we will prove the Lipschitz smoothness of $g(x, \phi, \psi)$. Since we know that for any $x$ and $\phi$, $\pi_\psi^*(x, \phi) = \arg\max_{\pi_\psi} V_{\mathcal{M}(x)}^{\pi_\phi, \pi_\psi}(\rho)$ exists and is unique. By Lemma A.2, we have

$$
\begin{aligned}
& \nabla_{(x,\phi)} J_1(x, \phi) \\
& = \nabla_{(x,\phi)} \max_\psi J(x, \phi, \psi) \\
& = \nabla_{(x,\phi)} \max_{\pi_\psi} V_{\mathcal{M}(x)}^{\pi_\phi, \pi_\psi}(\rho) \\
& = \nabla_{(x,\phi)} V_{\mathcal{M}(x)}^{\pi_\phi, \pi_\psi}(\rho) \Big|_{\pi_\psi = \pi_\psi^*(x,\phi)} \\
& = \nabla_{(x,\phi)} V_{\mathcal{M}(x)}^{\pi_\phi, \pi_\psi}(\rho) \Big|_{\psi \in \psi^*(x,\phi)}.
\end{aligned}
$$

Since any $\psi \in \psi^*(x, \phi)$ induces the same optimal policy $\pi_\psi^*(x, \phi)$, then we have for any $\psi', \psi'' \in \psi^*(x, \phi)$,

$$
\nabla_{(x,\phi)} V_{\mathcal{M}(x)}^{\pi_\phi, \pi_\psi}(\rho) \Big|_{\psi = \psi'} = \nabla_{(x,\phi)} V_{\mathcal{M}(x)}^{\pi_\phi, \pi_\psi}(\rho) \Big|_{\pi_\psi = \pi_\psi^*(x,\phi)} = \nabla_{(x,\phi)} V_{\mathcal{M}(x)}^{\pi_\phi, \pi_\psi}(\rho) \Big|_{\psi = \psi''}.
$$

Thus we can use any $\psi \in \psi^*(x, \phi)$ to calculate $\nabla_{(x,\phi)} J_1(x, \phi)$, and we choose $\psi = \log \pi_\psi^*(x, \phi)$, which is the canonical parameterization of the optimal policy $\pi_\psi^*(x, \phi)$, then we have

$$
\nabla_{(x,\phi)} J_1(x, \phi) = \nabla_{(x,\phi)} V_{\mathcal{M}(x)}^{\pi_\phi, \pi_\psi}(\rho) \Big|_{\psi = \log \pi_\psi^*(x,\phi)} = \nabla_{(x,\phi)} J(x, \phi, \log \pi_\psi^*(x, \phi)),
$$

which implies that for any $x_1, x_2, \phi_1, \phi_2$,

$$
\begin{aligned}
& \|\nabla_{(x,\phi)} J_1(x_1, \phi_1) - \nabla_{(x,\phi)} J_1(x_2, \phi_2)\| \\
& = \|\nabla_{(x,\phi)} J(x_1, \phi_1, \log \pi_\psi^*(x_1, \phi_1)) - \nabla_{(x,\phi)} J(x_2, \phi_2, \log \pi_\psi^*(x_2, \phi_2))\| \\
& \le L_{J,1} \left( \|x_1 - x_2\| + \|\phi_1 - \phi_2\| + \|\log \pi_\psi^*(x_1, \phi_1) - \log \pi_\psi^*(x_2, \phi_2)\| \right) \\
& \le L_{J,1}(1 + C_\pi) \left( \|x_1 - x_2\| + \|\phi_1 - \phi_2\| \right).
\end{aligned}
\tag{120}
$$

where the last inequality follows from Lemma C.10.

Similarly, we can prove that for any $x_1, x_2, \psi_1, \psi_2$,

$$\|\nabla_{(x,\psi)} J_2(x_1, \psi_1) - \nabla_{(x,\psi)} J_2(x_2, \psi_2)\| \leq L_{J,1}(1 + C_\pi)\left(\|x_1 - x_2\| + \|\psi_1 - \psi_2\|\right). \tag{121}$$

Therefore, we have

$$
\begin{aligned}
&\|\nabla g(x_1, \phi_1, \psi_1) - \nabla g(x_2, \phi_2, \psi_2)\| \\
&= \left\| \begin{bmatrix} \nabla_x J_1(x_1, \phi_1) - \nabla_x J_2(x_1, \psi_1) \\ \nabla_\phi J_1(x_1, \phi_1) \\ -\nabla_\psi J_2(x_1, \psi_1) \end{bmatrix} - \begin{bmatrix} \nabla_x J_1(x_2, \phi_2) - \nabla_x J_2(x_2, \psi_2) \\ \nabla_\phi J_1(x_2, \phi_2) \\ -\nabla_\psi J_2(x_2, \psi_2) \end{bmatrix} \right\| \\
&\leq \|\nabla_{(x,\phi)} J_1(x_1, \phi_1) - \nabla_{(x,\phi)} J_1(x_2, \phi_2)\| + \|\nabla_{(x,\psi)} J_2(x_1, \psi_1) - \nabla_{(x,\psi)} J_2(x_2, \psi_2)\| \\
&\leq 2L_{J,1}(1 + C_\pi)\left(\|x_1 - x_2\| + \|\phi_1 - \phi_2\| + \|\psi_1 - \psi_2\|\right),
\end{aligned} \tag{122}
$$

which proves the Lipschitz smoothness of $g(x, \phi, \psi)$.

Finally, we will prove the Hessian Lipschitz continuity of $g(x, \phi, \psi)$. By the chain rule, we have

$$\nabla^2 J_1(x, \phi) = \nabla_{11}^2 J(x, \phi, \log \pi_\psi^*(x, \phi)) + \nabla_{(x,\phi)} \log \pi_\psi^*(x, \phi) \nabla_{12}^2 J(x, \phi, \log \pi_\psi^*(x, \phi)), \tag{123}$$

where $\nabla_{11}^2 J(x, \phi, \log \pi_\psi^*(x, \phi)) \in \mathbb{R}^{(d_x + |\mathcal{S}||\mathcal{A}|) \times (d_x + |\mathcal{S}||\mathcal{A}|)}$ is the Hessian of $J$ with respect to the first two arguments $(x, \phi)$, and $\nabla_{12}^2 J(x, \phi, \log \pi_\psi^*(x, \phi)) \in \mathbb{R}^{|\mathcal{S}||\mathcal{B}| \times (d_x + |\mathcal{S}||\mathcal{A}|)}$ is the cross derivative of $J$ with respect to $(x, \phi)$ and $\psi$, and $\nabla_{(x,\phi)} \log \pi_\psi^*(x, \phi) \in \mathbb{R}^{(d_x + |\mathcal{S}||\mathcal{A}|) \times |\mathcal{S}||\mathcal{B}|}$ is the Jacobian of $\log \pi_\psi^*(x, \phi)$ with respect to $(x, \phi)$.

Since given any $x$ and $\phi$, $\pi_\psi^*(x, \phi)$ is the optimal policy for the MDP $\mathcal{M}^{\pi_\phi}(x)$ defined in the proof of Lemma C.5, where the reward function $\tilde{r}_{(x,\phi)}(s, b)$ is defined as

$$\tilde{r}_{(x,\phi)}(s, b) = \mathbb{E}_{a \sim \pi_\phi(\cdot|s)}\left[r_x(s, a, b)\right] - \tau_\phi H\left(\pi_\phi(\cdot|s)\right),$$

and the transition probability $\widetilde{\mathcal{P}}^{\pi_\phi}(s'|s, b)$ is defined as

$$\widetilde{\mathcal{P}}^{\pi_\phi}(s'|s, b) = \mathbb{E}_{a \sim \pi_\phi(\cdot|s)}\left[\mathcal{P}(s'|s, a, b)\right].$$

and in the proof of Lemma C.9, we also show that there exist constants $\tilde{B}_r, \tilde{C}_r$ and $\tilde{C}_P$ such that the following conditions are satisfied:

$$|\tilde{r}_{(x,\psi)}(s, b)| \leq B_r + \tau_\phi \log|\mathcal{A}| \triangleq \tilde{B}_r, \tag{124}$$

$$\sup_{s,b} \|\widetilde{\mathcal{P}}^{\pi_{\phi_1}}(s'|s, b) - \widetilde{\mathcal{P}}^{\pi_{\phi_2}}(s'|s, b)\|_1 \leq \tilde{C}_P \|(x_1, \phi_1) - (x_2, \phi_2)\|, \tag{125}$$

$$\sup_{s,b} |\tilde{r}_{(x_1,\phi_1)}(s, b) - \tilde{r}_{(x_2,\phi_2)}(s, b)| \leq \tilde{C}_r \|(x_1, \phi_1) - (x_2, \phi_2)\|. \tag{126}$$

Moreover, we also have

$$
\begin{aligned}
&\sup_{s,b} \left\|\nabla_x \tilde{r}_{(x_1,\phi_1)}(s, b) - \nabla_x \tilde{r}_{(x_2,\phi_2)}(s, b)\right\| \\
&= \sup_{s,b} \left\|\mathbb{E}_{a \sim \pi_{\phi_1}(\cdot|s)}\left[\nabla_x r_{x_1}(s, a, b)\right] - \mathbb{E}_{a \sim \pi_{\phi_2}(\cdot|s)}\left[\nabla_x r_{x_2}(s, a, b)\right]\right\| \\
&\leq \sup_{s,b} \left\|\mathbb{E}_{a \sim \pi_{\phi_1}(\cdot|s)}\left[\nabla_x r_{x_1}(s, a, b) - \nabla_x r_{x_2}(s, a, b)\right]\right\| \\
&\quad + \sup_{s,b} \left\|\mathbb{E}_{a \sim \pi_{\phi_1}(\cdot|s)}\left[\nabla_x r_{x_2}(s, a, b)\right] - \mathbb{E}_{a \sim \pi_{\phi_2}(\cdot|s)}\left[\nabla_x r_{x_2}(s, a, b)\right]\right\| \\
&\leq L_r \|x_1 - x_2\| + C_r \sup_s \|\pi_{\phi_1}(\cdot|s) - \pi_{\phi_2}(\cdot|s)\|_1 \\
&\leq L_r \|x_1 - x_2\| + C_r \frac{\sqrt{|\mathcal{A}|}}{2} \|\phi_1 - \phi_2\|,
\end{aligned} \tag{127}
$$

and

$$\sup_{s,b} \left\| \nabla_\phi \tilde{r}_{(x_1,\phi_1)}(s,b) - \nabla_\phi \tilde{r}_{(x_2,\phi_2)}(s,b) \right\|$$

$$\leq \sup_{s,b} \left\| \sum_a \nabla_\phi \pi_{\phi_1}(a|s) r_{x_1}(s,a,b) - \sum_a \nabla_\phi \pi_{\phi_2}(a|s) r_{x_2}(s,a,b) \right\|$$

$$+ \tau_\phi \sup_s \left\| \nabla_\phi H(\pi_{\phi_1}(\cdot|s)) - \nabla_\phi H(\pi_{\phi_2}(\cdot|s)) \right\|$$

$$\leq \sup_{s,b} \left\| \sum_a \nabla_\phi \pi_{\phi_1}(a|s) \left( r_{x_1}(s,a,b) - r_{x_2}(s,a,b) \right) \right\|$$

$$+ \sup_{s,b} \left\| \sum_a \left( \nabla_\phi \pi_{\phi_1}(a|s) - \nabla_\phi \pi_{\phi_2}(a|s) \right) r_{x_2}(s,a,b) \right\|$$

$$+ \tau_\phi \sup_s \left\| \nabla_\phi H(\pi_{\phi_1}(\cdot|s)) - \nabla_\phi H(\pi_{\phi_2}(\cdot|s)) \right\|$$

$$\leq |\mathcal{A}| \sup_{s,a,b} |r_{x_1}(s,a,b) - r_{x_2}(s,a,b)| + B_r |\mathcal{A}| \sup_{s,a} \left\| \nabla_\phi \pi_{\phi_1}(a|s) - \nabla_\phi \pi_{\phi_2}(a|s) \right\|$$

$$+ \tau_\phi \sup_s \left\| \nabla_\phi H(\pi_{\phi_1}(\cdot|s)) - \nabla_\phi H(\pi_{\phi_2}(\cdot|s)) \right\|$$

$$\leq \left( |\mathcal{A}| C_r + \frac{3}{2} B_r |\mathcal{A}| + \tau_\phi L_H \right) \| \phi_1 - \phi_2 \|, \tag{128}$$

where the last inequality follows from Lemma 6 in Zeng et al. (2022), and $L_H = \frac{4 + 8 \log |\mathcal{A}|}{(1-\gamma)^3}$.

Then there exists a constant $\widetilde{L}_r = \max\{L_r, |\mathcal{A}| C_r + \frac{3}{2} B_r |\mathcal{A}| + \tau_\phi L_H\}$ such that for any $x_1, x_2, \phi_1, \phi_2$,

$$\sup_{s,b} \left\| \nabla_{(x,\phi)} \tilde{r}_{(x_1,\phi_1)}(s,b) - \nabla_{(x,\phi)} \tilde{r}_{(x_2,\phi_2)}(s,b) \right\| \leq \widetilde{L}_r \| (x_1,\phi_1) - (x_2,\phi_2) \|. \tag{129}$$

For the transition probability, we have

$$\sup_{s,b} \left\| \nabla_{(x,\phi)} \tilde{\mathcal{P}}^{\pi_{\phi_1}} - \nabla_{(x,\phi)} \tilde{\mathcal{P}}^{\pi_{\phi_2}} \right\|$$

$$= \sup_{s,b} \left\| \sum_a \nabla_{(x,\phi)} \pi_{\phi_1}(a|s) \mathcal{P}(s'|s,a,b) - \sum_a \nabla_{(x,\phi)} \pi_{\phi_2}(a|s) \mathcal{P}(s'|s,a,b) \right\|$$

$$\leq \frac{3}{2} |\mathcal{A}| \| \phi_1 - \phi_2 \|$$

$$\leq \frac{3}{2} |\mathcal{A}| \| (x_1,\phi_1) - (x_2,\phi_2) \|$$

$$\triangleq \widetilde{C}_P \| (x_1,\phi_1) - (x_2,\phi_2) \|. \tag{130}$$

Then by Lemma C.3, we know that $\nabla_{(x,\phi)} \pi_\psi^*(x,\phi)$ is well-defined and there exists a constant $L_{\psi,1}$ such that for any $x_1, x_2, \phi_1, \phi_2$,

$$\| \nabla_{(x,\phi)} \pi_\psi^*(x_1,\phi_1) - \nabla_{(x,\phi)} \pi_\psi^*(x_2,\phi_2) \| \leq L_{\psi,1} \| (x_1,\phi_1) - (x_2,\phi_2) \|, \tag{131}$$

and by Lemma C.2, we know that there exists a constant $C_{\psi^*}$ such that for any $x, \phi$, $\| \nabla_{(x,\phi)} \pi_\psi^*(x,\phi) \| \leq C_{\psi^*}$.

Therefore, we have for any $x_1, x_2, \phi_1, \phi_2$,

$$
\begin{aligned}
&\|\nabla^2 J_1(x_1, \phi_1) - \nabla^2 J_1(x_2, \phi_2)\| \\
&\leq \|\nabla_{11}^2 J(x_1, \phi_1, \log \pi_\psi^*(x_1, \phi_1)) - \nabla_{11}^2 J(x_2, \phi_2, \log \pi_\psi^*(x_2, \phi_2))\| \\
&\quad + \|\nabla_{(x,\phi)} \log \pi_\psi^*(x_1, \phi_1) \nabla_{12}^2 J(x_1, \phi_1, \log \pi_\psi^*(x_1, \phi_1)) \\
&\quad - \nabla_{(x,\phi)} \log \pi_\psi^*(x_2, \phi_2) \nabla_{12}^2 J(x_2, \phi_2, \log \pi_\psi^*(x_2, \phi_2))\| \\
&\leq L_{J,2} \left( \|x_1 - x_2\| + \|\phi_1 - \phi_2\| + \|\log \psi^*(x_1, \phi_1) - \log \psi^*(x_2, \phi_2)\| \right) \\
&\quad + \|\nabla_{(x,\phi)} \log \pi_\psi^*(x_1, \phi_1) - \nabla_{(x,\phi)} \log \pi_\psi^*(x_2, \phi_2)\| \cdot \|\nabla_{12}^2 J(x_1, \phi_1, \log \pi_\psi^*(x_1, \phi_1))\| \\
&\quad + \|\nabla_{(x,\phi)} \log \pi_\psi^*(x_2, \phi_2)\| \cdot \|\nabla_{12}^2 J(x_1, \phi_1, \log \pi_\psi^*(x_1, \phi_1)) - \nabla_{12}^2 J(x_2, \phi_2, \log \pi_\psi^*(x_2, \phi_2))\| \\
&\leq L_{J,2}(1 + C_\pi) \left( \|x_1 - x_2\| + \|\phi_1 - \phi_2\| \right) \\
&\quad + L_{J,1} \sup_{s,b} \|\nabla_{(x,\phi)} \log \pi_\psi^*(b|s; x_1, \phi_1) - \nabla_{(x,\phi)} \log \pi_\psi^*(b|s; x_2, \phi_2)\| \\
&\quad + C_{\psi^*} L_{J,2}(1 + C_\pi) \left( \|x_1 - x_2\| + \|\phi_1 - \phi_2\| \right) \\
&\leq \left( L_{J,2}(1 + C_\pi) + C_{\psi^*} L_{J,2}(1 + C_\pi) \right) \left( \|x_1 - x_2\| + \|\phi_1 - \phi_2\| \right) \\
&\quad + L_{J,1} \frac{1}{\delta_\pi} \sup_{s,b} \|\nabla_{(x,\phi)} \pi_\psi^*(b|s; x_1, \phi_1) - \nabla_{(x,\phi)} \pi_\psi^*(b|s; x_2, \phi_2)\| \\
&\leq \left( L_{J,2}(1 + C_\pi) + C_{\psi^*} L_{J,2}(1 + C_\pi) + L_{J,1} \frac{L_{\psi,1}}{\delta_\pi} \right) \left( \|x_1 - x_2\| + \|\phi_1 - \phi_2\| \right). \quad (132)
\end{aligned}
$$

Thus, by taking $L_{J_1,2} = L_{J,2}(1 + C_\pi) + C_{\psi^*} L_{J,2}(1 + C_\pi) + L_{J,1} \frac{L_{\psi,1}}{\delta_\pi}$, we prove the Hessian Lipschitz continuity of $J_1(x, \phi)$.

Similarly, we can prove that for any $x_1, x_2, \psi_1, \psi_2$,

$$
\|\nabla^2 J_2(x_1, \psi_1) - \nabla^2 J_2(x_2, \psi_2)\| \leq L_{J_2,2} \left( \|x_1 - x_2\| + \|\psi_1 - \psi_2\| \right), \quad (133)
$$

for some constant $L_{J_2,2} = L_{J,2}(1 + C_\pi) + C_{\phi^*} L_{J,2}(1 + C_\pi) + L_{J,1} \frac{L_{\phi,1}}{\delta_\pi}$.

Therefore, we have for any $x_1, x_2, \phi_1, \phi_2, \psi_1, \psi_2$,

$$
\begin{aligned}
&\|\nabla^2 g(x_1, \phi_1, \psi_1) - \nabla^2 g(x_2, \phi_2, \psi_2)\| \\
&\leq \|\nabla^2 J_1(x_1, \phi_1) - \nabla^2 J_1(x_2, \phi_2)\| + \|\nabla^2 J_2(x_1, \psi_1) - \nabla^2 J_2(x_2, \psi_2)\| \\
&\leq (L_{J_1,2} + L_{J_2,2}) \left( \|x_1 - x_2\| + \|\phi_1 - \phi_2\| + \|\psi_1 - \psi_2\| \right), \quad (134)
\end{aligned}
$$

which proves the Hessian Lipschitz continuity of $g(x, \phi, \psi)$. $\qquad \square$

**Lemma C.13.** *Under Assumptions 2, let $\lambda \geq \lambda_0$, then the penalized objective function $h(x, \phi, \psi)$ is $L_{h,1}$-Lipschitz smooth for some constant $L_{h,1} > 0$.*

*Proof.* By definition of $h(x, \phi, \psi)$, we have

$$
h(x, \phi, \psi) = \frac{1}{\lambda} f(x, \phi, \psi) + g(x, \phi, \psi).
$$

From Assumption 2 and Lemma C.12, for any $\lambda > 0$, it is straightforward to verify that $h$ is $\left( \frac{1}{\lambda} L_{f,1} + L_{g,1} \right)$-smooth. Therefore, we just need to take $L_{h,1} = \frac{1}{\lambda_0} L_{f,1} + L_{g,1}$. $\qquad \square$

**Lemma C.14.** *Under Assumption 1 and 2, let $\lambda \geq \lambda_0$, then there exists a constant $L_F$ such that $F(x)$ is $L_F$-Lipschitz smooth.*

*Proof.* By Assumptions 1 and 2, we have $f(x, \phi, \psi)$ is $C_f$-Lipschitz, $L_{f,1}$-Lipschitz smooth and has $L_{f,2}$-Lipschitz Hessians in $(\phi, \psi)$.

By Lemma C.12, we also have $g(x, \phi, \psi)$ has $L_{g,1}$-Lipschitz gradients and $L_{g,2}$-Lipschitz Hessians.

Moreover, by Assumption 2 and Lemma C.8, $cf(x, \phi, \psi) + g(x, \phi, \psi)$ is $\mu_h$-PŁ in $(\phi, \psi)$ for any $0 \leq c \leq \frac{1}{\lambda_0}$.

Therefore, by Lemma 4.4 in Chen et al. (2024), we know that there exists a constant $L_F$ such that $F(x)$ is $L_F$-Lipschitz smooth.

$\square$

**Lemma C.15** (Estimation Error of Finite Horizon). *Under Assumption 1 and 2, there exists a constant $\sigma_H > 0$ such that for any $x$, $\phi$, $\psi$, compared with the true gradient, the estimation error of the estimated gradient using horizon $H$ satisfies*

$$\|\nabla_\phi J(x, \phi, \psi; H) - \nabla_\phi J(x, \phi, \psi)\|^2 \leq \gamma^{2H} H^2 \sigma_H^2,$$
$$\|\nabla_\psi J(x, \phi, \psi; H) - \nabla_\psi J(x, \phi, \psi)\|^2 \leq \gamma^{2H} H^2 \sigma_H^2,$$
$$\|\nabla_x J(x, \phi, \psi; H) - \nabla_x J(x, \phi, \psi)\|^2 \leq \gamma^{2H} \sigma_H^2.$$

*Proof.* By Lemma B.6, we have

$$\nabla_\phi J(x, \phi, \psi) = \mathbb{E}\left[\sum_{t=0}^{\infty} \gamma^t \nabla_\phi \log \pi_\phi(a_t|s_t) \left(Q_{\mathcal{M}(x)}^{\pi_\phi, \pi_\psi}(s_t, a_t, b_t) - \tau_\psi \log \pi_\psi(b_t|s_t) + \tau_\phi \log \pi_\phi(a_t|s_t)\right)\right]. \tag{135}$$

Then expanding the Q-function, we obtain

$$\nabla_\phi J(x, \phi, \psi) = \mathbb{E}\left[\sum_{t=0}^{\infty} \gamma^t \nabla_\phi \log \pi_\phi(a_t|s_t) G_t\right], \tag{136}$$

where

$$G_t \triangleq \mathbb{E}\left[\sum_{k=t}^{\infty} \gamma^{k-t}\left(r_x(s_k, a_k, b_k) - \tau_\psi \log \pi_\psi(b_k|s_k) + \tau_\phi \log \pi_\phi(a_k|s_k)\right)\bigg| s_0 = s_t, a_0 = a_t, b_0 = b_t\right]. \tag{137}$$

If we use a finite horizon $H$, then the estimated gradient is

$$\nabla_\phi J(x, \phi, \psi; H) = \mathbb{E}\left[\sum_{t=0}^{H-1} \gamma^t \nabla_\phi \log \pi_\phi(a_t|s_t) \widehat{G}_t^H\right], \tag{138}$$

where

$$\widehat{G}_t^H \triangleq \mathbb{E}\left[\sum_{k=t}^{H-1} \gamma^{k-t}\left(r_x(s_k, a_k, b_k) - \tau_\psi \log \pi_\psi(b_k|s_k) + \tau_\phi \log \pi_\phi(a_k|s_k)\right)\bigg| s_0 = s_t, a_0 = a_t, b_0 = b_t\right]. \tag{139}$$

Then the bias is bounded as

$$\|\nabla_\phi J(x, \phi, \psi; H) - \nabla_\phi J(x, \phi, \psi)\|$$
$$= \left\|\mathbb{E}\left[\sum_{t=0}^{H-1} \gamma^t \nabla_\phi \log \pi_\phi(a_t|s_t) \widehat{G}_t^H - \sum_{t=0}^{\infty} \gamma^t \nabla_\phi \log \pi_\phi(a_t|s_t) G_t\right]\right\|$$
$$\leq \left\|\mathbb{E}\left[\sum_{t=0}^{H-1} \gamma^t \nabla_\phi \log \pi_\phi(a_t|s_t) \widehat{G}_t^H - \sum_{t=0}^{H-1} \gamma^t \nabla_\phi \log \pi_\phi(a_t|s_t) G_t\right]\right\|$$
$$+ \left\|\mathbb{E}\left[\sum_{t=H}^{\infty} \gamma^t \nabla_\phi \log \pi_\phi(a_t|s_t) G_t\right]\right\|$$
$$= \left\|\mathbb{E}\left[\sum_{t=0}^{H-1} \gamma^t \nabla_\phi \log \pi_\phi(a_t|s_t) \left(\widehat{G}_t^H - G_t\right)\right]\right\| + \left\|\mathbb{E}\left[\sum_{t=H}^{\infty} \gamma^t \nabla_\phi \log \pi_\phi(a_t|s_t) G_t\right]\right\| \tag{140}$$

Moreover, we have the following bounds:

$$|G_t| \le \sum_{k=t}^{\infty} \gamma^{k-t} \max_{s,a,b} \left( |r_x(s,a,b)| + \tau_\psi |\log \pi_\psi(b|s)| + \tau_\phi |\log \pi_\phi(a|s)| \right)$$

$$\le \frac{B_r + (\tau_\psi + \tau_\phi) \log \frac{1}{\delta_\pi}}{1 - \gamma} \triangleq \frac{G_{\max}}{1 - \gamma}, \tag{141}$$

and

$$|\widehat{G}_t^H - G_t|$$

$$= \left| \mathbb{E}\left[ \sum_{k=H}^{\infty} \gamma^{k-t} \left( r_x(s_k, a_k, b_k) - \tau_\psi \log \pi_\psi(b_k|s_k) + \tau_\phi \log \pi_\phi(a_k|s_k) \right) \middle| s_0 = s_t, a_0 = a_t, b_0 = b_t \right] \right|$$

$$\le \gamma^{H-t} \frac{G_{\max}}{1 - \gamma}, \tag{142}$$

and

$$\|\nabla_\phi \log \pi_\phi(a|s)\| = \left\| \frac{\nabla_\phi \pi_\phi(a|s)}{\pi_\phi(a|s)} \right\| \le \frac{C_\pi}{\delta_\pi} \triangleq G_\pi. \tag{143}$$

Thus, we obtain

$$\|\nabla_\phi J(x, \phi, \psi; H) - \nabla_\phi J(x, \phi, \psi)\|^2$$

$$\le 2 \left\| \mathbb{E}\left[ \sum_{t=0}^{H-1} \gamma^t \nabla_\phi \log \pi_\phi(a_t|s_t) \left( \widehat{G}_t^H - G_t \right) \right] \right\|^2 + 2 \left\| \mathbb{E}\left[ \sum_{t=H}^{\infty} \gamma^t \nabla_\phi \log \pi_\phi(a_t|s_t) G_t \right] \right\|^2$$

$$\le \frac{2 G_\pi^2 G_{\max}^2}{(1 - \gamma)^2} \gamma^{2H} H^2 + \frac{2 G_\pi^2 G_{\max}^2}{(1 - \gamma)^4} \gamma^{2H}$$

$$\le \frac{4 G_\pi^2 G_{\max}^2}{(1 - \gamma)^4} \gamma^{2H} H^2. \tag{144}$$

Therefore, by taking $\sigma_H \ge \frac{2 G_\pi G_{\max}}{(1-\gamma)^2}$, we complete the proof of the estimation error for the gradient estimator of $\phi$. The proof for the estimation error of the gradient estimator of $\psi$ follows analogously.

The true gradient of $J(x, \phi, \psi)$ with respect to $x$ is

$$\nabla_x J(x, \phi, \psi) = \mathbb{E}\left[ \sum_{t=0}^{\infty} \gamma^t \nabla_x r_x(s_t, a_t, b_t) \right], \tag{145}$$

and the estimated gradient with finite horizon $H$ is

$$\nabla_x J(x, \phi, \psi; H) = \mathbb{E}\left[ \sum_{t=0}^{H-1} \gamma^t \nabla_x r_x(s_t, a_t, b_t) \right]. \tag{146}$$

Then the estimation error of using finite horizon $H$ is

$$\|\nabla_x J(x, \phi, \psi; H) - \nabla_x J(x, \phi, \psi)\|^2 = \left\| \mathbb{E}\left[ \sum_{t=H}^{\infty} \gamma^t \nabla_x r_x(s_t, a_t, b_t) \right] \right\|^2 \le C_r^2 \left( \sum_{t=H}^{\infty} \gamma^t \right)^2 = C_r^2 \frac{\gamma^{2H}}{(1 - \gamma)^2} \tag{147}$$

Therefore, by taking $\sigma_H = \max\{ \frac{C_r}{1-\gamma}, \frac{2 G_\pi G_{\max}}{(1-\gamma)^2} \} > 0$, we complete the proof. $\qquad \square$

**Lemma C.16.** *Under Assumptions 1 and 2, in Algorithm 1, for any $x$, $\phi$ and $\psi$, we have*

$$\mathbb{E}\left[\left\|\nabla_x \hat{J}(x,\phi,\psi;B_J,H) - \mathbb{E}\left[\nabla_x \hat{J}(x,\phi,\psi;H)\right]\right\|^2\right] \le \frac{\sigma_J^2}{B_J},$$

$$\mathbb{E}\left[\left\|\nabla_\phi \hat{J}(x,\phi,\psi;B_J,H) - \mathbb{E}\left[\nabla_\phi \hat{J}(x,\phi,\psi;H)\right]\right\|^2\right] \le \frac{\sigma_J^2}{B_J},$$

$$\mathbb{E}\left[\left\|\nabla_\psi \hat{J}(x,\phi,\psi;B_J,H) - \mathbb{E}\left[\nabla_\psi \hat{J}(x,\phi,\psi;H)\right]\right\|^2\right] \le \frac{\sigma_J^2}{B_J},$$

*for some constant $\sigma_J^2 > 0$.*

*Proof.* By Lemma B.5, we have

$$\nabla_x J(x,\phi,\psi) = \mathbb{E}\left[\sum_{t=0}^{\infty}\gamma^t \nabla_x r_x(s_t,a_t,b_t)\right],$$

thus the truncated gradient with horizon $H$ is given by

$$\mathbb{E}\left[\nabla_x \hat{J}(x,\phi,\psi;H)\right] = \nabla_x J(x,\phi,\psi;H) = \mathbb{E}\left[\sum_{t=0}^{H-1}\gamma^t \nabla_x r_x(s_t,a_t,b_t)\right],$$

where the first equality holds due to the unbiasedness of Monte Carlo sampling with horizon $H$.

The stochastic gradient estimator with batch size $B_J$ is given by

$$\nabla_x \hat{J}(x,\phi,\psi;B_J,H) = \frac{1}{B_J}\sum_{i=1}^{B_J}\left(\sum_{t=0}^{H-1}\gamma^t \nabla_x r_x(s_t^i,a_t^i,b_t^i)\right). \tag{148}$$

Then we can get

$$\mathbb{E}\left[\left\|\nabla_x \hat{J}(x,\phi,\psi;B_J,H) - \mathbb{E}\left[\nabla_x \hat{J}(x,\phi,\psi;H)\right]\right\|^2\right]$$

$$= \mathbb{E}\left[\left\|\frac{1}{B_J}\sum_{i=1}^{B_J}\left(\sum_{t=0}^{H-1}\gamma^t \nabla_x r_x(s_t^i,a_t^i,b_t^i) - \mathbb{E}\left[\sum_{t=0}^{H-1}\gamma^t \nabla_x r_x(s_t,a_t,b_t)\right]\right)\right\|^2\right] \tag{149}$$

$$= \frac{1}{B_J}\mathbb{E}\left[\left\|\sum_{t=0}^{H-1}\gamma^t \nabla_x r_x(s_t,a_t,b_t) - \mathbb{E}\left[\sum_{t=0}^{H-1}\gamma^t \nabla_x r_x(s_t,a_t,b_t)\right]\right\|^2\right] \tag{150}$$

$$= \frac{1}{B_J}\left(\mathbb{E}\left[\left\|\sum_{t=0}^{H-1}\gamma^t \nabla_x r_x(s_t,a_t,b_t)\right\|^2\right] - \left\|\mathbb{E}\left[\sum_{t=0}^{H-1}\gamma^t \nabla_x r_x(s_t,a_t,b_t)\right]\right\|^2\right) \tag{151}$$

$$\le \frac{1}{B_J}\mathbb{E}\left[\left\|\sum_{t=0}^{H-1}\gamma^t \nabla_x r_x(s_t,a_t,b_t)\right\|^2\right] \tag{152}$$

$$\le \frac{1}{B_J}\frac{C_r^2}{(1-\gamma)^2}, \tag{153}$$

where the second equality holds due to $\mathbb{E}\left[\left\|\frac{1}{n}\sum_{i=1}^{n}X_i - \mathbb{E}[X]\right\|^2\right] = \frac{1}{n}\mathbb{E}\left[\|X - \mathbb{E}[X]\|^2\right]$; the third equality holds due to $\mathbb{E}\left[\|X - \mathbb{E}[X]\|^2\right] = \mathbb{E}\|X\|^2 - \|\mathbb{E}[X]\|^2$; the last inequality follows from $C_r$-Lipschitz continuity of $r_x$.

By Lemma B.6, we have

$$\nabla_\phi J(x,\phi,\psi) = \mathbb{E}\left[\sum_{t=0}^{\infty}\gamma^t \nabla \log \pi_\phi(a_t|s_t)\left(Q_{\mathcal{M}(x)}^{\pi_\phi,\pi_\psi}(s_t,a_t,b_t) - \tau_\psi \log \pi_\psi(b_t|s_t) + \tau_\phi \log \pi_\phi(a_t|s_t)\right)\right],$$

then the truncated gradient with horizon $H$ is given by

$$
\begin{aligned}
&\mathbb{E}\left[\nabla_\phi \hat{J}(x, \phi, \psi; H)\right] \\
&= \nabla_\phi J(x, \phi, \psi; H) \\
&= \mathbb{E}\left[\sum_{t=0}^{H-1} \gamma^t \nabla \log \pi_\phi(a_t|s_t) G_{t,H}\right],
\end{aligned}
\tag{154}
$$

where

$$
G_{t,H} \triangleq \mathbb{E}\left[\sum_{k=t}^{H-1} \gamma^{k-t} \left(r_x(s_k, a_k, b_k) - \tau_\psi \log \pi_\psi(b_k|s_k) + \tau_\phi \log \pi_\phi(a_k|s_k)\right) \bigg| s_0 = s_t, a_0 = a_t, b_0 = b_t\right].
\tag{155}
$$

The stochastic gradient estimator with batch size $B_J$ is given by

$$
\begin{aligned}
&\nabla_\phi \hat{J}(x, \phi, \psi; B_J, H) \\
&= \frac{1}{B_J} \sum_{i=1}^{B_J} \left(\sum_{t=0}^{H-1} \gamma^t \nabla \log \pi_\phi(a_t^i|s_t^i) \widehat{G}_{t,H}^i\right),
\end{aligned}
\tag{156}
$$

where

$$
\widehat{G}_{t,H}^i \triangleq \sum_{k=t}^{H-1} \gamma^{k-t} \left(r_x(s_k^i, a_k^i, b_k^i) - \tau_\psi \log \pi_\psi(b_k^i|s_k^i) + \tau_\phi \log \pi_\phi(a_k^i|s_k^i)\right),
\tag{157}
$$

and $\{s_t^i, a_t^i, b_t^i\}_{t=0}^{H-1}$ is the $i$th trajectory i.i.d. sampled under policies $\pi_\phi$ and $\pi_\psi$ in MDP $\mathcal{M}(x)$.

The upper bound of the absolute value of $\widehat{G}_{t,H}$ is given by

$$
\begin{aligned}
|\widehat{G}_{t,H}^i| &\leq \sum_{k=t}^{H-1} \gamma^{k-t} \left(|r_x(s_k^i, a_k^i, b_k^i)| + \tau_\psi |\log \pi_\psi(b_k^i|s_k^i)| + \tau_\phi |\log \pi_\phi(a_k^i|s_k^i)|\right) \\
&\leq \sum_{k=t}^{H-1} \gamma^{k-t} G_{\max} \\
&\leq \frac{1}{1-\gamma} G_{\max},
\end{aligned}
\tag{158}
$$

where $G_{\max}$ is the upper bound of the absolute value of the reward and the entropy terms defined in Lemma C.15.

Then we have

$$
\mathbb{E}\left[\left\|\nabla_\phi \hat{J}(x, \phi, \psi; B_J, H) - \mathbb{E}\left[\nabla_\phi \hat{J}(x, \phi, \psi; H)\right]\right\|^2\right]
$$

$$
= \mathbb{E}\left[\left\|\frac{1}{B_J}\nabla_\phi \sum_{i=1}^{B_J}\hat{J}_i(x, \phi, \psi; H) - \mathbb{E}\left[\nabla_\phi \hat{J}(x, \phi, \psi; H)\right]\right\|^2\right] \tag{159}
$$

$$
= \frac{1}{B_J}\mathbb{E}\left[\left\|\nabla_\phi \hat{J}(x, \phi, \psi; H) - \mathbb{E}\left[\nabla_\phi \hat{J}(x, \phi, \psi; H)\right]\right\|^2\right] \tag{160}
$$

$$
\leq \frac{1}{B_J}\mathbb{E}\left[\left\|\nabla_\phi \hat{J}(x, \phi, \psi; H)\right\|^2\right] \tag{161}
$$

$$
= \frac{1}{B_J}\mathbb{E}\left[\left\|\sum_{t=0}^{H-1}\gamma^t \nabla \log \pi_\phi(a_t|s_t)\widehat{G}_{t,H}\right\|^2\right] \tag{162}
$$

$$
\leq \frac{1}{B_J}\frac{G_\pi^2 G_{\max}^2}{(1-\gamma)^2}\left(\sum_{t=0}^{H-1}\gamma^t\right)^2 \tag{163}
$$

$$
\leq \frac{1}{B_J}\frac{G_\pi^2 G_{\max}^2}{(1-\gamma)^4}, \tag{164}
$$

where (159)–(161) are obtained using the same argument as (149)–(152), and the second-to-last inequality uses the definitions of $G_\pi$ and $G_{\max}$ in Lemma C.15. The proof for $\nabla_\psi \hat{J}(x, \phi, \psi; B_J, H)$ follows analogously.

Therefore, by taking $\sigma_J = \max\{\frac{C_r}{1-\gamma}, \frac{G_\pi G_{\max}}{(1-\gamma)^2}\} > 0$, we complete the proof. $\qquad\square$

### C.4. Proofs of Theorem 1

**Lemma C.17.** *Suppose Assumptions 1 to 3 hold. Then, for each outer iteration $t$ of Algorithm 1, we have*

$$
\mathbb{E}\left[\|\ell_t - \mathbb{E}[\ell_t]\|^2\right] \leq \frac{2\sigma_f^2}{B} + \frac{8\lambda^2 \sigma_J^2}{B_J}.
$$

*Proof.* In Algorithm 1, we have

$$
\begin{aligned}
\ell_t &= \nabla_x \widehat{\mathcal{L}}_\lambda(x_t, y_{t+1}, z_{t+1}, \tilde{y}_{t+1}, \tilde{z}_{t+1}; B, B_J, H)\\
&= \nabla_x \hat{f}(x_t, \phi_t^K, \psi_t^K; B) + \lambda \nabla_x \widehat{g}(x_t, y_{t+1}, z_{t+1}, \tilde{y}_{t+1}, \tilde{z}_{t+1}; B_J, H)\\
&= \nabla_x \hat{f}(x_t, \phi_t^K, \psi_t^K; B) + \lambda \nabla_x \hat{J}(x_t, \phi_t^K, \tilde{\psi}_t^K; B_J, H) - \lambda \nabla_x \hat{J}(x_t, \tilde{\phi}_t^K, \psi_t^K; B_J, H). \tag{165}
\end{aligned}
$$

Therefore, it follows from Assumption 3 and Lemma C.16 that

$$
\begin{aligned}
\mathbb{E}\left[\|\ell_t - \mathbb{E}[\ell_t]\|^2\right] \leq{}& 2\mathbb{E}\left[\left\|\frac{1}{B}\sum_{i=1}^{B}\nabla_x \hat{f}_i(x_t, \phi_t^K, \psi_t^K) - \mathbb{E}\left[\nabla_x \hat{f}(x_t, \phi_t^K, \psi_t^K)\right]\right\|^2\right]\\
&+ 4\lambda^2 \mathbb{E}\left[\left\|\nabla_x \hat{J}(x_t, \phi_t^K, \tilde{\psi}_t^K; B_J, H) - \mathbb{E}\left[\nabla_x \hat{J}(x_t, \phi_t^K, \tilde{\psi}_t^K; H)\right]\right\|^2\right]\\
&+ 4\lambda^2 \mathbb{E}\left[\left\|\nabla_x \hat{J}(x_t, \tilde{\phi}_t^K, \psi_t^K; B_J, H) - \mathbb{E}\left[\nabla_x \hat{J}(x_t, \tilde{\phi}_t^K, \psi_t^K; H)\right]\right\|^2\right]\\
\leq{}& \frac{2\sigma_f^2}{B} + \frac{8\lambda^2 \sigma_J^2}{B_J}. \tag{166}
\end{aligned}
$$

$\qquad\square$

**Lemma C.18.** *Suppose Assumptions 1 to 3 hold. In Algorithm 1, if $\lambda \geq \lambda_0$, then*

$$\frac{1}{T}\sum_{t=0}^{T-1}\mathbb{E}\left[\|\mathbb{E}[\ell_t] - \nabla\mathcal{L}_\lambda^*(x_t)\|^2\right]$$

$$\leq \frac{C_1}{T}\frac{8L_\theta}{\mu_\theta}\exp(-\mu_h\eta_\theta K)\Delta_\theta + C_1\frac{8L_\theta}{\mu_\theta}\exp(-\mu_h\eta_\theta K)\frac{1}{T}\sum_{t=0}^{T-1}\mathbb{E}\left[\|x_{t+1} - x_t\|^2\right]$$

$$+ \mathcal{O}(\frac{\lambda^2}{B_J}) + \mathcal{O}(\frac{\sigma_f^2}{B}) + \mathcal{O}(\lambda^2\gamma^{2H}H^2),$$

*where $C_1 = \frac{1}{c}\left(2L_{f,1}^2 + 32\lambda^2 L_{J,1}^2 + 32\lambda^2 L_{J,1}^2 C_\pi^2\right)$, $c$ is a constant defined in (189), $L_\theta = \max\{L_{h,1}, L_{J,1}\}$, $\mu_\theta = \min\{\mu_h, \mu_J\}$, and $\Delta_\theta$ is the initial optimality gap of the parameters defined in (206).*

*Proof.* In this proof, we use the notation for $\theta$, $\theta^*(x)$, and $\theta^*(x, \theta)$ introduced in Table A.1.

In Algorithm 1, the estimation error is bounded as

$$\|\mathbb{E}[\ell_t] - \nabla\mathcal{L}_\lambda^*(x_t)\|^2$$

$$\leq 2\|\nabla f(x_t, \theta_\lambda^*(x_t)) - \nabla f(x_t, \theta_t^K)\|^2 + 2\lambda^2\|\nabla\tilde{g}(x_t, \theta_\lambda^*(x_t), \theta^*(x_t, \theta_\lambda^*(x_t))) - \nabla\tilde{g}(x_t, \theta_t^K, \tilde{\theta}_t^K; H)\|^2$$

$$\leq 2L_{f,1}^2\,\text{dist}^2(\theta_t^K, \theta_\lambda^*(x_t)) + 4\lambda^2\|\nabla\tilde{g}(x_t, \theta_\lambda^*(x_t), \theta^*(x_t, \theta_\lambda^*(x_t))) - \nabla\tilde{g}(x_t, \theta_t^K, \tilde{\theta}_t^K)\|^2$$

$$\quad + 4\lambda^2\|\nabla\tilde{g}(x_t, \theta_t^K, \tilde{\theta}_t^K) - \nabla\tilde{g}(x_t, \theta_t^K, \tilde{\theta}_t^K; H)\|^2$$

$$\leq 2L_{f,1}^2\,\text{dist}^2(\theta_t^K, \theta_\lambda^*(x_t)) + 2\lambda^2 L_{\tilde{g},1}^2\left(2\,\text{dist}^2(\theta_\lambda^*(x_t), \theta_t^K) + 2\,\text{dist}^2(\theta^*(x_t, \theta_\lambda^*(x_t)), \tilde{\theta}_t^K)\right)$$

$$\quad + 8\lambda^2\|\nabla_x J(x_t, \phi_t^K, \tilde{\psi}_t^K) - \nabla_x J(x_t, \phi_t^K, \tilde{\psi}_t^K; H)\|^2 + 8\lambda^2\|\nabla_x J(x_t, \tilde{\phi}_t^K, \psi_t^K) - \nabla_x J(x_t, \tilde{\phi}_t^K, \psi_t^K; H)\|^2$$

$$\leq \left(2L_{f,1}^2 + 4\lambda^2 L_{\tilde{g},1}^2\right)\text{dist}^2(\theta_\lambda^*(x_t), \theta_t^K) + 4\lambda^2 L_{\tilde{g},1}^2\,\text{dist}^2(\theta^*(x_t, \theta_\lambda^*(x_t)), (\tilde{\phi}_t^K, \tilde{\psi}_t^K)) + 16\lambda^2\gamma^{2H}\sigma_H^2$$

$$\leq \left(2L_{f,1}^2 + 4\lambda^2 L_{\tilde{g},1}^2 + 8\lambda^2 L_{\tilde{g},1}^2 C_\pi^2\right)\text{dist}^2(\theta_\lambda^*(x_t), \theta_t^K) + 8\lambda^2 L_{\tilde{g},1}^2\,\text{dist}^2(\theta^*(x_t, \theta_t^K), \tilde{\theta}_t^K) + 16\lambda^2\gamma^{2H}\sigma_H^2, \quad (167)$$

where the second inequality decomposes the estimation error of $\tilde{g}$ into an optimization error and a stochastic error arising from the finite horizon $H$; the second-to-last inequality is implied by Lemma C.15; and the last inequality follows from Lemma C.10.

We next bound the distance terms arising from the inner loop. By the $L_{h,1}$-smoothness of $h(x, \theta)$, we have for $\eta_\theta \leq \frac{1}{2L_{h,1}}$,

$$h(x_t, \theta_t^{k+1}) \leq h(x_t, \theta_t^k) + \langle\nabla_\theta h(x_t, \theta_t^k), \theta_t^{k+1} - \theta_t^k\rangle + \frac{L_{h,1}}{2}\|\theta_t^{k+1} - \theta_t^k\|^2$$

$$= h(x_t, \theta_t^k) - \frac{\eta_\theta}{2}\|\nabla_\theta h(x_t, \theta_t^k)\|^2 + \left(\frac{L_{h,1}\eta_\theta^2}{2} - \frac{\eta_\theta}{2}\right)\|g_t^k\|^2 + \frac{\eta_\theta}{2}\|b_t^k\|^2$$

$$\leq h(x_t, \theta_t^k) - \frac{\eta_\theta}{2}\|\nabla_\theta h(x_t, \theta_t^k)\|^2 - \frac{1}{4\eta_\theta}\|\theta_t^{k+1} - \theta_t^k\|^2 + \frac{\eta_\theta}{2}\|b_t^k\|^2, \quad (168)$$

where $b_t^k \triangleq \nabla_\theta h(x_t, \theta_t^k) - g_t^k$ is the gradient-estimation error.

Since $h(x, \theta)$ satisfies the $\mu_h$-PŁ condition with respect to $\theta$, it follows that

$$h(x_t, \theta_t^{k+1}) - h^*(x_t) \leq h(x_t, \theta_t^k) - h^*(x_t) - \frac{\eta_\theta}{2}\|\nabla_\theta h(x_t, \theta_t^k)\|^2 - \frac{1}{4\eta_\theta}\|\theta_t^{k+1} - \theta_t^k\|^2 + \frac{\eta_\theta}{2}\|b_t^k\|^2$$

$$\leq (1 - \mu_h\eta_\theta)\left(h(x_t, \theta_t^k) - h^*(x_t)\right) - \frac{1}{4\eta_\theta}\|\theta_t^{k+1} - \theta_t^k\|^2 + \frac{\eta_\theta}{2}\|b_t^k\|^2, \quad (169)$$

where $h^*(x_t) = \min_\theta h(x_t, \theta)$.

For the error term $b_t^k$, we obtain

$$
\begin{aligned}
\|b_t^k\|^2 &= \left\|\nabla_\theta h(x_t, \theta_t^k) - g_t^k\right\|^2 \\
&= \left\|\frac{1}{\lambda}\nabla_\theta f(x_t, \theta_t^k) - \frac{1}{\lambda}\nabla_\theta \hat{f}(x_t, \theta_t^k; B) + \nabla_\theta \tilde{g}(x_t, \theta_t^k, \theta^*(x_t, \theta_t^k)) - \nabla_\theta \widehat{\tilde{g}}(x_t, \theta_t^k, \tilde{\theta}_t^{k+1}; B_J, H)\right\|^2 \\
&\leq \frac{2}{\lambda^2}\frac{\sigma_f^2}{B} + 4\left\|\nabla_\theta \tilde{g}(x_t, \theta_t^k, \theta^*(x_t, \theta_t^k)) - \nabla_\theta \tilde{g}(x_t, \theta_t^k, \tilde{\theta}_t^{k+1})\right\|^2 \\
&\quad + 4\left\|\nabla_\theta \tilde{g}(x_t, \theta_t^k, \tilde{\theta}_t^{k+1}) - \nabla_\theta \widehat{\tilde{g}}(x_t, \theta_t^k, \tilde{\theta}_t^{k+1}; B_J, H)\right\|^2,
\end{aligned}
\tag{170}
$$

where the second equality follows from $g(x_t, \theta_t^k) = \tilde{g}(x_t, \theta_t^k, \theta^*(x_t, \theta_t^k))$ and the corresponding gradient identity with respect to $\theta$.

We first consider the second error term between $\nabla_\theta \tilde{g}$ and $\nabla_\theta \widehat{\tilde{g}}(\cdot; B_J, H)$, which arises from the stochastic gradient estimation with batch size $B_J$ and the finite-horizon truncation with horizon length $H$. We have

$$
\begin{aligned}
&\mathbb{E}\left[\left\|\nabla_\theta \tilde{g}(x_t, \theta_t^k, \tilde{\theta}_t^{k+1}) - \nabla_\theta \widehat{\tilde{g}}(x_t, \theta_t^k, \tilde{\theta}_t^{k+1}; B_J, H)\right\|^2\right] \\
&\leq 2\mathbb{E}\left[\left\|\nabla_\theta \tilde{g}(x_t, \theta_t^k, \tilde{\theta}_t^{k+1}) - \nabla_\theta \widehat{\tilde{g}}(x_t, \theta_t^k, \tilde{\theta}_t^{k+1}; H)\right\|^2\right] \\
&\quad + 2\mathbb{E}\left[\left\|\nabla_\theta \widehat{\tilde{g}}(x_t, \theta_t^k, \tilde{\theta}_t^{k+1}; H) - \frac{1}{B_J}\sum_{i=1}^{B_J}\nabla_\theta \widehat{\tilde{g}}_i(x_t, \theta_t^k, \tilde{\theta}_t^{k+1}; H)\right\|^2\right]
\end{aligned}
\tag{171}
$$

$$
\leq 4\gamma^{2H}H^2\sigma_H^2 + \frac{4\sigma_J^2}{B_J},
\tag{172}
$$

where the last inequality follows from Lemma C.15 and Lemma C.16.

The first error term can be bounded as

$$
\begin{aligned}
&\|\nabla_\theta \tilde{g}(x_t, \theta_t^k, \theta^*(x_t, \theta_t^k)) - \nabla_\theta \tilde{g}(x_t, \theta_t^k, \tilde{\theta}_t^{k+1})\|^2 \\
&\leq L_{\tilde{g},1}^2 \operatorname{dist}(\theta^*(x_t, \theta_t^k), \tilde{\theta}_t^{k+1})^2 \\
&= L_{\tilde{g},1}^2 \operatorname{dist}^2(\phi^*(x_t, \psi_t^k), \tilde{\phi}_t^{k+1}) + L_{\tilde{g},1}^2 \operatorname{dist}^2(\psi^*(x_t, \phi_t^k), \tilde{\psi}_t^{k+1}) \\
&\leq \frac{2L_{\tilde{g},1}^2}{\mu_J}\left(J(x_t, \tilde{\phi}_t^{k+1}, \psi_t^k) - J_2(x_t, \psi_t^k)\right) + \frac{2L_{\tilde{g},1}^2}{\mu_J}\left(-J(x_t, \phi_t^k, \tilde{\psi}_t^{k+1}) + J_1(x_t, \phi_t^k)\right),
\end{aligned}
\tag{173}
$$

$$\tag{174}$$

where the first inequality follows from the $L_{\tilde{g},1}$-smoothness of $\tilde{g}$ in Lemma C.11; the second inequality follows from the QG condition implied by the $\mu_J$-PŁ condition of $J(x, \phi, \psi)$ in Lemma C.7.

We next bound the optimality gap of $J$. By the $L_{J,1}$-smoothness of $J(x, \phi, \psi)$ and $\mu_J$-PŁ of $J(x, \phi, \psi)$ in $\phi$, we have for $\eta_\phi \leq \frac{1}{L_{J,1}}$,

$$
\begin{aligned}
&J(x_t, \tilde{\phi}_t^{k+1}, \psi_t^k) - J_2(x_t, \psi_t^k) \\
&\leq J(x_t, \tilde{\phi}_t^k, \psi_t^k) - J_2(x_t, \psi_t^k) + \langle \nabla_\phi J(x_t, \tilde{\phi}_t^k, \psi_t^k), \tilde{\phi}_t^{k+1} - \tilde{\phi}_t^k\rangle + \frac{L_{J,1}}{2}\|\tilde{\phi}_t^{k+1} - \tilde{\phi}_t^k\|^2 \\
&= J(x_t, \tilde{\phi}_t^k, \psi_t^k) - J_2(x_t, \psi_t^k) - \frac{\eta_\phi}{2}\|\nabla_\phi J(x_t, \tilde{\phi}_t^k, \psi_t^k)\|^2 \\
&\quad + \left(\frac{L_{J,1}\eta_\phi^2}{2} - \frac{\eta_\phi}{2}\right)\|u_t^k\|^2 + \frac{\eta_\phi}{2}\|\nabla_\phi J(x_t, \tilde{\phi}_t^k, \psi_t^k) - u_t^k\|^2 \\
&\leq J(x_t, \tilde{\phi}_t^k, \psi_t^k) - J_2(x_t, \psi_t^k) - \frac{\eta_\phi}{2}\|\nabla_\phi J(x_t, \tilde{\phi}_t^k, \psi_t^k)\|^2 + \frac{\eta_\phi}{2}\|\nabla_\phi J(x_t, \tilde{\phi}_t^k, \psi_t^k) - u_t^k\|^2 \\
&\leq (1 - \mu_J\eta_\phi)\left(J(x_t, \tilde{\phi}_t^k, \psi_t^k) - J_2(x_t, \psi_t^k)\right) + \frac{\eta_\phi}{2}\|\nabla_\phi J(x_t, \tilde{\phi}_t^k, \psi_t^k) - u_t^k\|^2,
\end{aligned}
\tag{175}
$$

where the first inequality follows from the $L_{J,1}$-smoothness of $J(x, \phi, \psi)$; the first equality follows from the update rule of $\tilde{\phi}_t^k$ and the identity $\langle a, b \rangle = \frac{1}{2}\|a\|^2 + \frac{1}{2}\|b\|^2 - \frac{1}{2}\|a - b\|^2$; the second inequality follows from $\eta_\phi \leq \frac{1}{L_{J,1}}$; and the last inequality follows from the $\mu_J$-PŁ condition of $J(x, \phi, \psi)$.

The error term can be bounded as

$$
\mathbb{E}\left[\left\|\nabla_\phi J(x_t, \tilde{\phi}_t^k, \psi_t^k) - u_t^k\right\|^2\right]
$$

$$
= \mathbb{E}\left[\left\|\nabla_\phi J(x_t, \tilde{\phi}_t^k, \psi_t^k) - \nabla_\phi \hat{J}(x_t, \tilde{\phi}_t^k, \psi_t^k; B_J, H)\right\|^2\right]
$$

$$
\leq 2\mathbb{E}\left[\left\|\nabla_\phi J(x_t, \tilde{\phi}_t^k, \psi_t^k) - \nabla_\phi J(x_t, \tilde{\phi}_t^k, \psi_t^k; H)\right\|^2\right]
$$

$$
+ 2\mathbb{E}\left[\left\|\nabla_\phi J(x_t, \tilde{\phi}_t^k, \psi_t^k; H) - \frac{1}{B_J}\sum_{i=1}^{B_J}\nabla_\phi \hat{J}_i(x_t, \tilde{\phi}_t^k, \psi_t^k; H)\right\|^2\right]
$$

$$
\leq 2\gamma^{2H}H^2\sigma_H^2 + \frac{2\sigma_J^2}{B_J}, \tag{176}
$$

where the last inequality follows from Lemma C.15 and Lemma C.16.

Therefore, the optimality gap of $J$ w.r.t. $\phi$ can be bounded by

$$
J(x_t, \tilde{\phi}_t^{k+1}, \psi_t^k) - J_2(x_t, \psi_t^k) \leq (1 - \mu_J\eta_\phi)\left(J(x_t, \tilde{\phi}_t^k, \psi_t^k) - J_2(x_t, \psi_t^k)\right) + \eta_\phi\left(\gamma^{2H}H^2\sigma_H^2 + \frac{\sigma_J^2}{B_J}\right). \tag{177}
$$

Similarly, for $\eta_\psi \leq \frac{1}{L_{J,1}}$, we can bound the optimality gap of $J$ w.r.t. $\psi$ by

$$
-J(x_t, \phi_t^k, \tilde{\psi}_t^{k+1}) + J_1(x_t, \phi_t^k) \leq (1 - \mu_J\eta_\psi)\left(-J(x_t, \phi_t^k, \tilde{\psi}_t^k) + J_1(x_t, \phi_t^k)\right) + \eta_\psi\left(\gamma^{2H}H^2\sigma_H^2 + \frac{\sigma_J^2}{B_J}\right). \tag{178}
$$

Thus, substituting the above two inequalities back into (174), we have

$$
\|\nabla_\theta \tilde{g}(x_t, \theta_t^k, \theta^*(x_t, \theta_t^k)) - \nabla_\theta \tilde{g}(x_t, \theta_t^k, \tilde{\theta}_t^{k+1})\|^2
$$

$$
\leq \frac{2L_{\tilde{g},1}^2}{\mu_J}\left(J(x_t, \tilde{\phi}_t^{k+1}, \psi_t^k) - J_2(x_t, \psi_t^k)\right) + \frac{2L_{\tilde{g},1}^2}{\mu_J}\left(-J(x_t, \phi_t^k, \tilde{\psi}_t^{k+1}) + J_1(x_t, \phi_t^k)\right)
$$

$$
\leq \frac{2L_{\tilde{g},1}^2}{\mu_J}(1 - \mu_J\eta_\phi)\left(J(x_t, \tilde{\phi}_t^k, \psi_t^k) - J_2(x_t, \psi_t^k)\right) + \frac{2L_{\tilde{g},1}^2}{\mu_J}(1 - \mu_J\eta_\psi)\left(-J(x_t, \phi_t^k, \tilde{\psi}_t^k) + J_1(x_t, \phi_t^k)\right)
$$

$$
+ \frac{2L_{\tilde{g},1}^2}{\mu_J}(\eta_\phi + \eta_\psi)\left(\gamma^{2H}H^2\sigma_H^2 + \frac{\sigma_J^2}{B_J}\right)
$$

$$
\leq \frac{2L_{\tilde{g},1}^2}{\mu_J}(1 - \mu_J\eta_\phi)\left(J(x_t, \tilde{\phi}_t^k, \psi_t^k) - J_2(x_t, \psi_t^k)\right) + \frac{2L_{\tilde{g},1}^2}{\mu_J}(1 - \mu_J\eta_\psi)\left(-J(x_t, \phi_t^k, \tilde{\psi}_t^k) + J_1(x_t, \phi_t^k)\right)
$$

$$
+ \frac{16L_{J,1}}{\mu_J}\left(\gamma^{2H}H^2\sigma_H^2 + \frac{\sigma_J^2}{B_J}\right)
$$

$$
\leq \kappa_b\left(J(x_t, \tilde{\phi}_t^k, \psi_t^k) - J_2(x_t, \psi_t^k) - J(x_t, \phi_t^k, \tilde{\psi}_t^k) + J_1(x_t, \phi_t^k)\right) + \frac{16L_{J,1}}{\mu_J}\left(\gamma^{2H}H^2\sigma_H^2 + \frac{\sigma_J^2}{B_J}\right), \tag{179}
$$

where $\kappa_b \triangleq \max\left\{\frac{2L_{\tilde{g},1}^2}{\mu_J}(1 - \mu_J\eta_\phi), \frac{2L_{\tilde{g},1}^2}{\mu_J}(1 - \mu_J\eta_\psi)\right\}$, and the second-to-last inequality follows from $L_{\tilde{g},1} = 2L_{J,1}$ in Lemma C.11 and $\eta_\phi \leq \frac{1}{L_{J,1}}, \eta_\psi \leq \frac{1}{L_{J,1}}$.

Combining the preceding estimates yields the following bound on the error term $b_t^k$:

$$\mathbb{E}\left[\|b_t^k\|^2\right] \leq \kappa_b \mathbb{E}\left(J(x_t, \tilde{\phi}_t^k, \psi_t^k) - J_2(x_t, \psi_t^k) - J(x_t, \phi_t^k, \tilde{\psi}_t^k) + J_1(x_t, \phi_t^k)\right)$$
$$+ 16\left(1 + \frac{L_{J,1}}{\mu_J}\right)\left(\gamma^{2H} H^2 \sigma_H^2 + \frac{\sigma_J^2}{B_J}\right) + \frac{2\sigma_f^2}{B\lambda^2}. \tag{180}$$

Moreover, by the $L_{J,1}$-smoothness of $J(x, \phi, \psi)$, and $L_{J_2,1}$-smoothness of $J_2(x, \psi)$ established in Lemma C.12, where $L_{J_2,1}$ is absorbed into the NI smoothness constant $L_{g,1}$, we have, for $\eta_\theta \leq 1/(L_{J,1} + L_{J_2,1})$,

$$J(x_t, \tilde{\phi}_t^{k+1}, \psi_t^{k+1}) - J_2(x_t, \psi_t^{k+1})$$
$$\leq J(x_t, \tilde{\phi}_t^{k+1}, \psi_t^k) - J_2(x_t, \psi_t^k) + \langle \nabla_\psi J(x_t, \tilde{\phi}_t^{k+1}, \psi_t^k) - \nabla_\psi J_2(x_t, \psi_t^k), \psi_t^{k+1} - \psi_t^k \rangle$$
$$+ \frac{L_{J,1} + L_{J_2,1}}{2}\|\psi_t^{k+1} - \psi_t^k\|^2$$
$$\leq J(x_t, \tilde{\phi}_t^{k+1}, \psi_t^k) - J_2(x_t, \psi_t^k) + \eta_\theta L_{J,1} \operatorname{dist}(\tilde{\phi}_t^{k+1}, \phi^*(x_t, \psi_t^k))\|\frac{\psi_t^{k+1} - \psi_t^k}{\eta_\theta}\|$$
$$+ \frac{(L_{J,1} + L_{J_2,1})\eta_\theta^2}{2}\|\frac{\psi_t^{k+1} - \psi_t^k}{\eta_\theta}\|^2$$
$$\leq J(x_t, \tilde{\phi}_t^{k+1}, \psi_t^k) - J_2(x_t, \psi_t^k) + \frac{\alpha\eta_\theta L_{J,1}}{2}\operatorname{dist}^2(\tilde{\phi}_t^{k+1}, \phi^*(x_t, \psi_t^k))$$
$$+ \left(\frac{\eta_\theta L_{J,1}}{2\alpha} + \frac{(L_{J,1} + L_{J_2,1})\eta_\theta^2}{2}\right)\|\frac{\psi_t^{k+1} - \psi_t^k}{\eta_\theta}\|^2, \quad \forall \alpha > 0$$
$$\leq J(x_t, \tilde{\phi}_t^{k+1}, \psi_t^k) - J_2(x_t, \psi_t^k) + \frac{\alpha\eta_\theta L_{J,1}}{\mu_J}\left(J(x_t, \tilde{\phi}_t^{k+1}, \psi_t^k) - J_2(x_t, \psi_t^k)\right)$$
$$+ \left(\frac{\eta_\theta L_{J,1}}{2\alpha} + \frac{(L_{J,1} + L_{J_2,1})\eta_\theta^2}{2}\right)\|\frac{\psi_t^{k+1} - \psi_t^k}{\eta_\theta}\|^2, \quad \forall \alpha > 0$$
$$= \left(1 + \frac{\alpha\eta_\theta L_{J,1}}{\mu_J}\right)\left(J(x_t, \tilde{\phi}_t^{k+1}, \psi_t^k) - J_2(x_t, \psi_t^k)\right)$$
$$+ \left(\frac{\eta_\theta L_{J,1}}{2\alpha} + \frac{(L_{J,1} + L_{J_2,1})\eta_\theta^2}{2}\right)\|\frac{\psi_t^{k+1} - \psi_t^k}{\eta_\theta}\|^2, \quad \forall \alpha > 0, \tag{181}$$

where the second inequality is due to the $L_{J,1}$-smoothness of $J$; the third inequality uses the fact that $ab \leq \frac{\alpha}{2}a^2 + \frac{1}{2\alpha}b^2$ for any $\alpha > 0$; the last inequality follows from the $\mu_J$-PŁ condition of $J$.

Now substituting (177) into the above inequality, we have

$$J(x_t, \tilde{\phi}_t^{k+1}, \psi_t^{k+1}) - J_2(x_t, \psi_t^{k+1})$$
$$\leq \left(1 + \frac{\alpha\eta_\theta L_{J,1}}{\mu_J}\right)(1 - \eta_\phi \mu_J)\left(J(x_t, \tilde{\phi}_t^k, \psi_t^k) - J_2(x_t, \psi_t^k)\right) + \left(\frac{\eta_\theta L_{J,1}}{2\alpha} + \frac{(L_{J,1} + L_{J_2,1})\eta_\theta^2}{2}\right)\|\frac{\psi_t^{k+1} - \psi_t^k}{\eta_\theta}\|^2$$
$$+ \left(1 + \frac{\alpha\eta_\theta L_{J,1}}{\mu_J}\right)\eta_\phi\left(\gamma^{2H} H^2 \sigma_H^2 + \frac{\sigma_J^2}{B_J}\right), \quad \forall \alpha > 0. \tag{182}$$

Let

$$\kappa_{1,1} \triangleq \left(1 + \frac{\alpha\eta_\theta L_{J,1}}{\mu_J}\right)(1 - \eta_\phi \mu_J),$$
$$\kappa_{2,1} \triangleq \frac{L_{J,1}}{2\eta_\theta \alpha} + \frac{L_{J,1} + L_{J_2,1}}{2},$$

where the definition of $\kappa_{2,1}$ absorbs the factor $\eta_\theta^{-2}$ from $\|(\psi_t^{k+1} - \psi_t^k)/\eta_\theta\|^2$. Then we have

$$
\begin{aligned}
&J(x_t, \tilde{\phi}_t^{k+1}, \psi_t^{k+1}) - J_2(x_t, \psi_t^{k+1}) \\
&\leq \kappa_{1,1}\left(J(x_t, \tilde{\phi}_t^k, \psi_t^k) - J_2(x_t, \psi_t^k)\right) + \left(\frac{\eta_\theta L_{J,1}}{2\alpha} + \frac{(L_{J,1} + L_{J_2,1})\eta_\theta^2}{2}\right)\|\frac{\psi_t^{k+1} - \psi_t^k}{\eta_\theta}\|^2 \\
&\quad + \left(1 + \frac{\alpha\eta_\theta L_{J,1}}{\mu_J}\right)\eta_\phi\left(\gamma^{2H}H^2\sigma_H^2 + \frac{\sigma_J^2}{B_J}\right) \\
&= \kappa_{1,1}\left(J(x_t, \tilde{\phi}_t^k, \psi_t^k) - J_2(x_t, \psi_t^k)\right) + \kappa_{2,1}\|\psi_t^{k+1} - \psi_t^k\|^2 + \left(1 + \frac{\alpha\eta_\theta L_{J,1}}{\mu_J}\right)\eta_\phi\left(\gamma^{2H}H^2\sigma_H^2 + \frac{\sigma_J^2}{B_J}\right). \quad (183)
\end{aligned}
$$

Similarly, by the $L_{J,1}$-smoothness of $J(x, \phi, \psi)$, and $L_{J_1,1}$-smoothness of $J_1(x, \phi)$, we have for $\eta_\theta \leq 1/(L_{J,1} + L_{J_1,1})$ and any $\alpha' > 0$,

$$
\begin{aligned}
&- J(x_t, \phi_t^{k+1}, \tilde{\psi}_t^{k+1}) + J_1(x_t, \phi_t^{k+1}) \\
&\leq \kappa_{1,2}\left(-J(x_t, \phi_t^k, \tilde{\psi}_t^k) + J_1(x_t, \phi_t^k)\right) + \left(\frac{\eta_\theta L_{J,1}}{2\alpha'} + \frac{(L_{J,1} + L_{J_1,1})\eta_\theta^2}{2}\right)\|\frac{\phi_t^{k+1} - \phi_t^k}{\eta_\theta}\|^2 \\
&\quad + \left(1 + \frac{\alpha'\eta_\theta L_{J,1}}{\mu_J}\right)\eta_\psi\left(\gamma^{2H}H^2\sigma_H^2 + \frac{\sigma_J^2}{B_J}\right) \\
&= \kappa_{1,2}\left(-J(x_t, \phi_t^k, \tilde{\psi}_t^k) + J_1(x_t, \phi_t^k)\right) + \kappa_{2,2}\|\phi_t^{k+1} - \phi_t^k\|^2 + \left(1 + \frac{\alpha'\eta_\theta L_{J,1}}{\mu_J}\right)\eta_\psi\left(\gamma^{2H}H^2\sigma_H^2 + \frac{\sigma_J^2}{B_J}\right), \quad (184)
\end{aligned}
$$

where

$$
\begin{aligned}
\kappa_{1,2} &\triangleq \left(1 + \frac{\alpha'\eta_\theta L_{J,1}}{\mu_J}\right)(1 - \eta_\psi\mu_J), \\
\kappa_{2,2} &\triangleq \frac{L_{J,1}}{2\eta_\theta\alpha'} + \frac{L_{J,1} + L_{J_1,1}}{2}.
\end{aligned}
$$

Therefore, we have

$$
\begin{aligned}
&J(x_t, \tilde{\phi}_t^{k+1}, \psi_t^{k+1}) - J_2(x_t, \psi_t^{k+1}) - J(x_t, \phi_t^{k+1}, \tilde{\psi}_t^{k+1}) + J_1(x_t, \phi_t^{k+1}) \\
&\leq \kappa_1\left(J(x_t, \tilde{\phi}_t^k, \psi_t^k) - J_2(x_t, \psi_t^k) - J(x_t, \phi_t^k, \tilde{\psi}_t^k) + J_1(x_t, \phi_t^k)\right) + \kappa_2\|\theta_t^{k+1} - \theta_t^k\|^2 \\
&\quad + \left(1 + \frac{\alpha\eta_\theta L_{J,1}}{\mu_J}\right)\eta_\phi\left(\gamma^{2H}H^2\sigma_H^2 + \frac{\sigma_J^2}{B_J}\right) + \left(1 + \frac{\alpha'\eta_\theta L_{J,1}}{\mu_J}\right)\eta_\psi\left(\gamma^{2H}H^2\sigma_H^2 + \frac{\sigma_J^2}{B_J}\right), \quad (185)
\end{aligned}
$$

where $\kappa_1 = \max\{\kappa_{1,1}, \kappa_{1,2}\}$ and $\kappa_2 = \max\{\kappa_{2,1}, \kappa_{2,2}\}$.

Since our goal is to derive a joint recursion involving both $h$ and $J$, we multiply the preceding inequality by a positive constant $c > 0$ and add it to (169). Define

$$
J_t^k = J(x_t, \tilde{\phi}_t^k, \psi_t^k) - J_2(x_t, \psi_t^k) - J(x_t, \phi_t^k, \tilde{\psi}_t^k) + J_1(x_t, \phi_t^k).
$$

Then we have

$$
\begin{aligned}
&h(x_t, \theta_t^{k+1}) - h^*(x_t) + cJ_t^{k+1} \\
&\leq (1 - \mu_h\eta_\theta)\left(h(x_t, \theta_t^k) - h^*(x_t)\right) + \left(c\kappa_2 - \frac{1}{4\eta_\theta}\right)\|\theta_t^{k+1} - \theta_t^k\|^2 + \frac{\eta_\theta}{2}\|b_t^k\|^2 + c\kappa_1 J_t^k \\
&\quad + c\left(1 + \frac{\alpha\eta_\theta L_{J,1}}{\mu_J}\right)\eta_\phi\left(\gamma^{2H}H^2\sigma_H^2 + \frac{\sigma_J^2}{B_J}\right) + c\left(1 + \frac{\alpha'\eta_\theta L_{J,1}}{\mu_J}\right)\eta_\psi\left(\gamma^{2H}H^2\sigma_H^2 + \frac{\sigma_J^2}{B_J}\right). \quad (186)
\end{aligned}
$$

Taking expectation with respect to all sources of randomness and substituting the bound on $b_t^k$ from (180) into the preceding inequality, we obtain

$$
\mathbb{E}\left[h(x_t, \theta_t^{k+1}) - h^*(x_t) + cJ_t^{k+1}\right]
$$

$$
\leq (1 - \mu_h \eta_\theta)\mathbb{E}\left[h(x_t, \theta_t^k) - h^*(x_t)\right] + \left(c\kappa_1 + \frac{\eta_\theta}{2}\kappa_b\right)\mathbb{E}\left[J_t^k\right] + \left(c\kappa_2 - \frac{1}{4\eta_\theta}\right)\mathbb{E}\left[\|\theta_t^{k+1} - \theta_t^k\|^2\right]
$$

$$
+ 8\eta_\theta \left(1 + \frac{L_{J,1}}{\mu_J}\right)\left(\gamma^{2H}H^2\sigma_H^2 + \frac{\sigma_J^2}{B_J} + \frac{\sigma_f^2}{B\lambda^2}\right) + c\left(1 + \frac{\alpha\eta_\theta L_{J,1}}{\mu_J}\right)\eta_\phi\left(\gamma^{2H}H^2\sigma_H^2 + \frac{\sigma_J^2}{B_J}\right)
$$

$$
+ c\left(1 + \frac{\alpha'\eta_\theta L_{J,1}}{\mu_J}\right)\eta_\psi\left(\gamma^{2H}H^2\sigma_H^2 + \frac{\sigma_J^2}{B_J}\right). \tag{187}
$$

To obtain a contraction, it suffices to impose the following two conditions:

$$
c\kappa_2 - \frac{1}{4\eta_\theta} = \frac{cL_{J,1}}{2\eta_\theta \min\{\alpha, \alpha'\}} + \frac{cL_{J,1} + cL_{J,\star}}{2} - \frac{1}{4\eta_\theta} \tag{188}
$$

$$
= \frac{1}{2\eta_\theta}\left(c\left(\frac{L_{J,1}}{\min\{\alpha, \alpha'\}} + L_{J,1}\eta_\theta + L_{J,\star}\eta_\theta\right) - \frac{1}{2}\right) \leq 0,
$$

where $L_{J,\star} = \max\{L_{J_1,1}, L_{J_2,1}\}$, and

$$
c\kappa_1 + \frac{\eta_\theta}{2}\kappa_b < c(1 - \mu_h \eta_\theta).
$$

Since

$$
\kappa_1 = \max\left\{\left(1 + \frac{\alpha\eta_\theta L_{J,1}}{\mu_J}\right)(1 - \eta_\phi \mu_J), \left(1 + \frac{\alpha'\eta_\theta L_{J,1}}{\mu_J}\right)(1 - \eta_\psi \mu_J)\right\},
$$

and $\kappa_b = \max\left\{\frac{2L_{\tilde{g},1}^2}{\mu_J}(1 - \mu_J\eta_\phi), \frac{2L_{\tilde{g},1}^2}{\mu_J}(1 - \mu_J\eta_\psi)\right\} > 0$ is independent of $\eta_\theta$, it suffices to ensure that the following three inequalities hold:

$$
c\left(1 + \frac{\alpha\eta_\theta L_{J,1}}{\mu_J}\right)(1 - \eta_\phi \mu_J) + \frac{\eta_\theta}{2}\kappa_b - c(1 - \mu_h \eta_\theta) < 0,
$$

$$
c\left(1 + \frac{\alpha'\eta_\theta L_{J,1}}{\mu_J}\right)(1 - \eta_\psi \mu_J) + \frac{\eta_\theta}{2}\kappa_b - c(1 - \mu_h \eta_\theta) < 0,
$$

$$
c\left(\frac{L_{J,1}}{\min\{\alpha, \alpha'\}} + L_{J,1}\eta_\theta + L_{J,\star}\eta_\theta\right) \leq \frac{1}{2}.
$$

Choose

$$
\begin{cases}
\alpha = \frac{\mu_J}{L_{J,1}(1 - \eta_\phi \mu_J)} > 0 \\
\alpha' = \frac{\mu_J}{L_{J,1}(1 - \eta_\psi \mu_J)} > 0 \\
c = \min\{\frac{1}{2\left(\frac{L_{J,1}}{\min\{\alpha, \alpha'\}} + L_{J,1} + L_{J,\star}\right)}, 1\} > 0 \\
\eta_\theta \leq \min\{(\eta_\phi \mu_J)(2 + 2\mu_h + \frac{\kappa_b}{c})^{-1}, (\eta_\psi \mu_J)(2 + 2\mu_h + \frac{\kappa_b}{c})^{-1}, \frac{1}{2L_{h,1}}, \frac{1}{\mu_h + 1}, 1\} \\
\eta_\phi \leq \min\{\frac{1}{L_{J,1}}, \frac{1}{\mu_J + 1}, 1\} \\
\eta_\psi \leq \min\{\frac{1}{L_{J,1}}, \frac{1}{\mu_J + 1}, 1\}
\end{cases} \tag{189}
$$

With these choices, $c = \mathcal{O}(1)$, $\eta_\theta = \mathcal{O}(1)$, $\eta_\phi = \mathcal{O}(1)$, and $\eta_\psi = \mathcal{O}(1)$. Moreover, we obtain

$$c\left(1 + \frac{\alpha\eta_\theta L_{J,1}}{\mu_J}\right)(1 - \eta_\phi\mu_J) + \frac{\eta_\theta}{2}\kappa_b - c(1 - \mu_h\eta_\theta) = c\left(-\eta_\phi\mu_J + \eta_\theta\left(1 + \frac{\kappa_b}{2c} + \mu_h\right)\right) \le -c\frac{\eta_\phi\mu_J}{2} < 0, \quad (190)$$

$$c\left(1 + \frac{\alpha'\eta_\theta L_{J,1}}{\mu_J}\right)(1 - \eta_\psi\mu_J) + \frac{\eta_\theta}{2}\kappa_b - c(1 - \mu_h\eta_\theta) = c\left(-\eta_\psi\mu_J + \eta_\theta\left(1 + \frac{\kappa_b}{2c} + \mu_h\right)\right) \le -c\frac{\eta_\psi\mu_J}{2} < 0, \quad (191)$$

$$c\left(\frac{L_{J,1}}{\min\{\alpha, \alpha'\}} + L_{J,1}\eta_\theta + L_{J,\star}\eta_\theta\right) \le c\left(\frac{L_{J,1}}{\min\{\alpha, \alpha'\}} + L_{J,1} + L_{J,\star}\right) \le \frac{1}{2}, \quad (192)$$

$$\max\left\{\left(1 + \frac{\alpha\eta_\theta L_{J,1}}{\mu_J}\right), \left(1 + \frac{\alpha'\eta_\theta L_{J,1}}{\mu_J}\right)\right\} \le \max\left\{1 + \frac{\eta_\theta}{1 - \eta_\phi\mu_J}, 1 + \frac{\eta_\theta}{1 - \eta_\psi\mu_J}\right\} \le 2. \quad (193)$$

Therefore, we have

$$\mathbb{E}\left[h(x_t, \theta_t^{k+1}) - h^*(x_t) + cJ_t^{k+1}\right]$$

$$\le (1 - \mu_h\eta_\theta)\mathbb{E}\left[h(x_t, \theta_t^k) - h^*(x_t) + cJ_t^k\right] + 8\eta_\theta\left(1 + \frac{L_{J,1}}{\mu_J}\right)\left(\gamma^{2H}H^2\sigma_H^2 + \frac{\sigma_J^2}{B_J} + \frac{\sigma_f^2}{B\lambda^2}\right)$$

$$+ c\left(1 + \frac{\alpha\eta_\theta L_{J,1}}{\mu_J}\right)\eta_\phi\left(\gamma^{2H}H^2\sigma_H^2 + \frac{\sigma_J^2}{B_J}\right) + c\left(1 + \frac{\alpha'\eta_\theta L_{J,1}}{\mu_J}\right)\eta_\psi\left(\gamma^{2H}H^2\sigma_H^2 + \frac{\sigma_J^2}{B_J}\right)$$

$$\le (1 - \mu_h\eta_\theta)\mathbb{E}\left[h(x_t, \theta_t^k) - h^*(x_t) + cJ_t^k\right] + 12\left(1 + \frac{L_{J,1}}{\mu_J}\right)\left(\gamma^{2H}H^2\sigma_H^2 + \frac{\sigma_J^2}{B_J} + \frac{\sigma_f^2}{B\lambda^2}\right), \quad (194)$$

where the last inequality follows from the parameter choices in (189) and the upper bound in (193).

Denote

$$E_\theta \triangleq \left(1 + \frac{L_{J,1}}{\mu_J}\right)\left(\gamma^{2H}H^2\sigma_H^2 + \frac{\sigma_J^2}{B_J} + \frac{\sigma_f^2}{B\lambda^2}\right) = \mathcal{O}(\gamma^{2H}H^2) + \mathcal{O}\left(\frac{1}{B_J}\right) + \mathcal{O}\left(\frac{\sigma_f^2}{B\lambda^2}\right). \quad (195)$$

Here, $\sigma_H$ and $\sigma_J$ are constants from Lemma C.15 and Lemma C.16, respectively, while $\sigma_f$ is the variance bound in Assumption 3. Thus, only the dependence on $\sigma_f$ is retained explicitly in the final bound.

By unrolling the preceding inequality over $k = 0, 1, \ldots, K - 1$, we obtain

$$\mathbb{E}\left[h(x_t, \theta_t^K) - h^*(x_t) + cJ_t^K\right]$$

$$\le (1 - \mu_h\eta_\theta)^K\mathbb{E}\left[h(x_t, \theta_t^0) - h^*(x_t) + cJ_t^0\right] + 12\sum_{k=0}^{K-1}(1 - \mu_h\eta_\theta)^{K-k-1}E_\theta$$

$$\le (1 - \mu_h\eta_\theta)^K\mathbb{E}\left[h(x_t, \theta_t^0) - h^*(x_t) + cJ_t^0\right] + \frac{12}{\mu_h\eta_\theta}E_\theta. \quad (196)$$

For notational simplicity, we suppress the expectation notation whenever it is clear from context. Then we have

$$\text{dist}^2(\theta_t^K, \theta_\lambda^*(x_t)) + c\,\text{dist}^2(\tilde{\theta}_t^K, \theta^*(x_t, \theta_t^K))$$

$$\overset{(a)}{\leq} \frac{2}{\mu_h}\left(h(x_t, \theta_t^K) - h^*(x_t)\right) + c\frac{2}{\mu_J}J_t^K \tag{197}$$

$$\overset{(b)}{\leq} \frac{2}{\mu_\theta}\left(h(x_t, \theta_t^K) - h^*(x_t) + cJ_t^K\right) \tag{198}$$

$$\overset{(c)}{\leq} \frac{2}{\mu_\theta}(1 - \mu_h\eta_\theta)^K\left(h(x_t, \theta_t^0) - h^*(x_t) + cJ_t^0\right) + \frac{24}{\mu_\theta^2\eta_\theta}E_\theta \tag{199}$$

$$\overset{(d)}{\leq} \frac{2}{\mu_\theta}(1 - \mu_h\eta_\theta)^K L_\theta\left(\text{dist}^2\left(\theta_t^0, \theta_\lambda^*(x_t)\right) + c\,\text{dist}^2\left(\tilde{\theta}_t^0, \theta^*(x_t, \theta_t^0)\right)\right) + \frac{24}{\mu_\theta^2\eta_\theta}E_\theta \tag{200}$$

$$= \frac{2}{\mu_\theta}(1 - \mu_h\eta_\theta)^K L_\theta\left(\text{dist}^2\left(\theta_{t-1}^K, \theta_\lambda^*(x_t)\right) + c\,\text{dist}^2\left(\tilde{\theta}_{t-1}^K, \theta^*(x_t, \theta_{t-1}^K)\right)\right) + \frac{24}{\mu_\theta^2\eta_\theta}E_\theta \tag{201}$$

$$\overset{(e)}{\leq} \frac{4}{\mu_\theta}(1 - \mu_h\eta_\theta)^K L_\theta\left(\text{dist}^2\left(\theta_{t-1}^K, \theta_\lambda^*(x_{t-1})\right) + c\,\text{dist}^2\left(\tilde{\theta}_{t-1}^K, \theta^*(x_{t-1}, \theta_{t-1}^K)\right)\right) + \frac{24}{\mu_\theta^2\eta_\theta}E_\theta$$

$$+ \frac{4}{\mu_\theta}(1 - \mu_h\eta_\theta)^K L_\theta\left(\text{dist}^2\left(\theta_\lambda^*(x_t), \theta_\lambda^*(x_{t-1})\right) + c\,\text{dist}^2\left(\theta^*(x_t, \theta_{t-1}^K), \theta^*(x_{t-1}, \theta_{t-1}^K)\right)\right) \tag{202}$$

$$\overset{(f)}{\leq} \frac{4}{\mu_\theta}\exp(-\mu_h\eta_\theta K)L_\theta\left(\text{dist}^2\left(\theta_{t-1}^K, \theta_\lambda^*(x_{t-1})\right) + c\,\text{dist}^2\left(\tilde{\theta}_{t-1}^K, \theta^*(x_{t-1}, \theta_{t-1}^K)\right)\right) + \frac{24}{\mu_\theta^2\eta_\theta}E_\theta$$

$$+ \frac{4}{\mu_\theta}\exp(-\mu_h\eta_\theta K)L_\theta\left(\frac{L_{h,1}^2}{\mu_h^2} + cC_\pi^2\right)\|x_t - x_{t-1}\|^2 \tag{203}$$

$$\leq \frac{4L_\theta}{\mu_\theta}\exp(-\mu_h\eta_\theta K)\left(\text{dist}^2\left(\theta_{t-1}^K, \theta_\lambda^*(x_{t-1})\right) + c\,\text{dist}^2\left(\tilde{\theta}_{t-1}^K, \theta^*(x_{t-1}, \theta_{t-1}^K)\right)\right) + \frac{24}{\mu_\theta^2\eta_\theta}E_\theta$$

$$+ \frac{4L_\theta}{\mu_\theta}\exp(-\mu_h\eta_\theta K)\left(\frac{L_\theta^2}{\mu_\theta^2} + C_\pi^2\right)\|x_t - x_{t-1}\|^2, \tag{204}$$

where $(a)$ follows from the QG condition implied by the $\mu_h$-PŁ condition of $h$ and the $\mu_J$-PŁ condition of $J$; $(b)$ follows by setting $\mu_\theta \triangleq \min\{\mu_h, \mu_J\} > 0$; $(c)$ applies the recursion in (196); $(d)$ relies on the $L_{h,1}$-smoothness of $h$, the $L_{J,1}$-smoothness of $J$, and the definition $L_\theta \triangleq \max\{L_{h,1}, L_{J,1}\} > 0$; $(e)$ is obtained by applying $\text{dist}^2(A, B) \leq 2\,\text{dist}^2(A, C) + 2\,\text{dist}^2(B, C)$ to separate the terms involving $x_t$ and $x_{t-1}$; and $(f)$ is a consequence of Lemma 4.1 of Chen et al. (2024) and Lemma C.10, which give the Lipschitz continuity of $\theta_\lambda^*(x)$ and $\theta^*(x, \theta)$ with respect to $x$. Note that the distance in Definition 1 is no greater than the Hausdorff distance used in Chen et al. (2024).

By choosing $K \geq \frac{1}{\mu_h\eta_\theta}\log\frac{8L_\theta}{\mu_\theta}$, we ensure that $\frac{4L_\theta}{\mu_\theta}\exp(-\mu_h\eta_\theta K) \leq \frac{1}{2}$. Therefore, we obtain

$$\text{dist}^2(\theta_t^K, \theta_\lambda^*(x_t)) + c\,\text{dist}^2(\tilde{\theta}_t^K, \theta^*(x_t, \theta_t^K))$$

$$\leq \frac{1}{2^t}\left(\text{dist}^2\left(\theta_0^K, \theta_\lambda^*(x_0)\right) + c\,\text{dist}^2\left(\tilde{\theta}_0^K, \theta^*(x_0, \theta_0^K)\right)\right) + \frac{4L_\theta}{\mu_\theta}\exp(-\mu_h\eta_\theta K)\left(\frac{L_\theta^2}{\mu_\theta^2} + C_\pi^2\right)\sum_{i=0}^{t-1}\frac{\|x_{i+1} - x_i\|^2}{2^{t-i-1}}$$

$$+ \frac{24}{\mu_\theta^2\eta_\theta}E_\theta\sum_{i=0}^{t-1}\frac{1}{2^{t-i-1}}$$

$$\leq \frac{1}{2^t}\frac{2L_\theta}{\mu_\theta}(1 - \mu_h\eta_\theta)^K\left(\text{dist}^2\left(\theta_0^0, \theta_\lambda^*(x_0)\right) + c\,\text{dist}^2\left(\tilde{\theta}_0^0, \theta^*(x_0, \theta_0^0)\right)\right) + \frac{4L_\theta}{\mu_\theta}\exp(-\mu_h\eta_\theta K)\sum_{i=0}^{t-1}\frac{\|x_{i+1} - x_i\|^2}{2^{t-i-1}}$$

$$+ \frac{48}{\mu_\theta^2\eta_\theta}E_\theta + \frac{1}{2^t}\frac{24}{\mu_\theta^2\eta_\theta}E_\theta$$

$$\leq \frac{1}{2^t}\frac{4L_\theta}{\mu_\theta}\exp(-\mu_h\eta_\theta K)\Delta_\theta + \frac{4L_\theta}{\mu_\theta}\exp(-\mu_h\eta_\theta K)\sum_{i=0}^{t-1}\frac{\|x_{i+1} - x_i\|^2}{2^{t-i-1}} + \frac{72}{\mu_\theta^2\eta_\theta}E_\theta, \tag{205}$$

where

$$\Delta_\theta \triangleq \operatorname{dist}^2\left((\phi_0, \psi_0), (\phi^*_\lambda(x_0), \psi^*_\lambda(x_0))\right) + c\operatorname{dist}^2\left((\phi_0, \psi_0), (\phi^*(x_0, \psi_0), \psi^*(x_0, \phi_0))\right) \tag{206}$$

is the initial optimality gap of the parameters, and the second inequality is obtained by applying (200) at $t = 0$..

Substituting this bound back into (167), we obtain

$$
\begin{aligned}
&\mathbb{E}\left[\|\mathbb{E}[\ell_t] - \nabla\mathcal{L}^*_\lambda(x_t)\|^2\right] \\
&\leq \frac{2L^2_{f,1} + 32\lambda^2 L^2_{J,1} + 32\lambda^2 L^2_{J,1}C^2_\pi}{c}\mathbb{E}\left[\operatorname{dist}^2(\theta^*_\lambda(x_t), \theta^K_t) + c\operatorname{dist}^2(\theta^*(x_t, \theta^K_t), \tilde{\theta}^K_t)\right] + 16\lambda^2\gamma^{2H}H^2\sigma^2_H \\
&\leq C_1\frac{1}{2^t}\frac{4L_\theta}{\mu_\theta}\exp(-\mu_h\eta_\theta K)\Delta_\theta + C_1\frac{4L_\theta}{\mu_\theta}\exp(-\mu_h\eta_\theta K)\sum^{t-1}_{i=0}\mathbb{E}\left[\frac{\|x_{i+1} - x_i\|^2}{2^{t-i-1}}\right] \\
&\quad + C_1\frac{72}{\mu^2_\theta\eta_\theta}E_\theta + 16\lambda^2\gamma^{2H}H^2\sigma^2_H,
\end{aligned}
\tag{207}
$$

where $L_{\tilde{g},1} = 2L_{J,1}$ by Lemma C.11 and $C_1 \triangleq \frac{2L^2_{f,1} + 32\lambda^2 L^2_{J,1} + 32\lambda^2 L^2_{J,1}C^2_\pi}{c} = \mathcal{O}(\lambda^2)$.

Summing the above inequality over $t = 0, 1, \ldots, T - 1$, we have

$$
\begin{aligned}
&\frac{1}{T}\sum^{T-1}_{t=0}\mathbb{E}\left[\|\mathbb{E}[\ell_t] - \nabla\mathcal{L}^*_\lambda(x_t)\|^2\right] \\
&\leq C_1\frac{4L_\theta}{\mu_\theta}\exp(-\mu_h\eta_\theta K)\Delta_\theta\frac{1}{T}\sum^{T-1}_{t=0}\frac{1}{2^t} + C_1\frac{4L_\theta}{\mu_\theta}\exp(-\mu_h\eta_\theta K)\frac{1}{T}\sum^{T-1}_{t=0}\sum^{t-1}_{i=0}\mathbb{E}\left[\frac{\|x_{i+1} - x_i\|^2}{2^{t-i-1}}\right] \\
&\quad + C_1\frac{72}{\mu^2_\theta\eta_\theta}E_\theta + 16\lambda^2\gamma^{2H}H^2\sigma^2_H \\
&\leq \frac{C_1}{T}\frac{8L_\theta}{\mu_\theta}\exp(-\mu_h\eta_\theta K)\Delta_\theta + C_1\frac{8L_\theta}{\mu_\theta}\exp(-\mu_h\eta_\theta K)\frac{1}{T}\sum^{T-1}_{t=0}\mathbb{E}\left[\|x_{t+1} - x_t\|^2\right] \\
&\quad + \mathcal{O}\left(\frac{\lambda^2}{B_J}\right) + \mathcal{O}(\gamma^{2H}H^2\lambda^2) + \mathcal{O}\left(\frac{\sigma^2_f}{B}\right)
\end{aligned}
\tag{208}
\tag{209}
$$

where the second inequality follows from the definition of $E_\theta$ in (195) and the geometric-series bounds $\sum^{T-1}_{t=0}2^{-t} \leq 2$ and $\sum^{T-1}_{t=i+1}2^{-(t-i-1)} \leq 2$. $\qquad\square$

**Lemma C.19.** *Under Assumptions 1, 2, and 3, suppose that Algorithm 1 is run with $\lambda \geq \lambda_0$. Then we have*

$$
\begin{aligned}
&\frac{1}{T}\sum^{T-1}_{t=0}\mathbb{E}\left[\|\ell_t - \nabla F(x_t)\|^2\right] \\
&\leq \frac{3}{T}\sum^{T-1}_{t=0}\mathbb{E}\left[\|\mathbb{E}[\ell_t] - \nabla\mathcal{L}^*_\lambda(x_t)\|^2\right] + \mathcal{O}\left(\frac{\sigma^2_f}{B}\right) + \mathcal{O}\left(\frac{\lambda^2}{B_J}\right) + \mathcal{O}(\lambda^{-2}).
\end{aligned}
$$

*Proof.* By adding and subtracting $\mathbb{E}[\ell_t]$ and $\nabla\mathcal{L}^*_\lambda(x_t)$, we have

$$\|\ell_t - \nabla F(x_t)\| \leq \|\ell_t - \mathbb{E}[\ell_t]\| + \|\mathbb{E}[\ell_t] - \nabla\mathcal{L}^*_\lambda(x_t)\| + \|\nabla\mathcal{L}^*_\lambda(x_t) - \nabla F(x_t)\|, \tag{210}$$

which implies that

$$
\begin{aligned}
&\mathbb{E}\left[\|\ell_t - \nabla F(x_t)\|^2\right] \\
&\leq 3\mathbb{E}\left[\|\ell_t - \mathbb{E}[\ell_t]\|^2\right] + 3\mathbb{E}\left[\|\mathbb{E}[\ell_t] - \nabla\mathcal{L}^*_\lambda(x_t)\|^2\right] + 3\|\nabla\mathcal{L}^*_\lambda(x_t) - \nabla F(x_t)\|^2 \\
&\leq 3\mathbb{E}\left[\|\ell_t - \mathbb{E}[\ell_t]\|^2\right] + 3\mathbb{E}\left[\|\mathbb{E}[\ell_t] - \nabla\mathcal{L}^*_\lambda(x_t)\|^2\right] + \mathcal{O}(\lambda^{-2}),
\end{aligned}
\tag{211}
$$

where the last inequality follows from Lemma 4.3 in Chen et al. (2024).

Moreover, by Lemma C.17, the first error term can be bounded by

$$\mathbb{E}\left[\|\ell_t - \mathbb{E}[\ell_t]\|^2\right] \leq \frac{2\sigma_f^2}{B} + \frac{8\lambda^2\sigma_J^2}{B_J}. \tag{212}$$

Therefore, telescoping from $t = 0$ to $T - 1$, we have

$$\frac{1}{T}\sum_{t=0}^{T-1}\mathbb{E}\left[\|\ell_t - \nabla F(x_t)\|^2\right]$$

$$\leq \frac{3}{T}\sum_{t=0}^{T-1}\mathbb{E}\left[\|\ell_t - \mathbb{E}[\ell_t]\|^2\right] + \frac{3}{T}\sum_{t=0}^{T-1}\mathbb{E}\left[\|\mathbb{E}[\ell_t] - \nabla\mathcal{L}_\lambda^*(x_t)\|^2\right] + \mathcal{O}(\lambda^{-2})$$

$$\leq \frac{6\sigma_f^2}{B} + \frac{24\lambda^2\sigma_J^2}{B_J} + \frac{3}{T}\sum_{t=0}^{T-1}\mathbb{E}\left[\|\mathbb{E}[\ell_t] - \nabla\mathcal{L}_\lambda^*(x_t)\|^2\right] + \mathcal{O}(\lambda^{-2})$$

$$\leq \frac{3}{T}\sum_{t=0}^{T-1}\mathbb{E}\left[\|\mathbb{E}[\ell_t] - \nabla\mathcal{L}_\lambda^*(x_t)\|^2\right] + \mathcal{O}\left(\frac{\sigma_f^2}{B}\right) + \mathcal{O}\left(\frac{\lambda^2}{B_J}\right) + \mathcal{O}(\lambda^{-2}). \tag{213}$$

$\square$

**Theorem C.1** (Theorem 1). *Suppose Assumptions 1, 2, and 3 hold. Let $\lambda \geq \lambda_0$, and let the step sizes satisfy*

$$\eta_x \leq \frac{1}{2L_F}, \quad \eta_\theta \asymp \kappa^{-5}, \quad \eta_\phi, \eta_\psi \leq \min\left\{\frac{1}{L_{J,1}}, \frac{1}{\mu_g + 1}, 1\right\}.$$

*Then, for the iterates generated by Algorithm 1, choosing $K = \mathcal{O}(\log\lambda)$ yields*

$$\frac{1}{T}\sum_{t=0}^{T-1}\mathbb{E}\left[\|\nabla F(x_t)\|^2\right] \leq \mathcal{O}\left(\frac{\Delta_F}{T}\right) + \mathcal{O}\left(\frac{\sigma_f^2}{B}\right) + \mathcal{O}(\lambda^2\gamma^{2H}H^2) + \mathcal{O}(\lambda^{-2}) + \mathcal{O}\left(\frac{\lambda^2}{B_J}\right),$$

*where $\Delta_F = \mathbb{E}\left[F(x_0) - F(x_T)\right]$.*

*Choosing $T = \mathcal{O}(\epsilon^{-1})$, $\lambda = \mathcal{O}(\epsilon^{-1/2})$, $K = \mathcal{O}(\log\epsilon^{-1})$, $H = \mathcal{O}(\log\epsilon^{-1})$, $B_J = \mathcal{O}(\epsilon^{-2})$ and $B = \mathcal{O}(\epsilon^{-1})$, we can obtain*

$$\frac{1}{T}\sum_{t=0}^{T-1}\mathbb{E}\left[\|\nabla F(x_t)\|^2\right] \leq \mathcal{O}(\epsilon).$$

*Proof.* By the $L_F$-smoothness of $F(x)$ in Lemma C.14, we have for $\eta_x \leq \frac{1}{2L_F}$,

$$F(x_{t+1}) \leq F(x_t) + \langle\nabla F(x_t), x_{t+1} - x_t\rangle + \frac{L_F}{2}\|x_{t+1} - x_t\|^2$$

$$= F(x_t) - \frac{\eta_x}{2}\|\nabla F(x_t)\|^2 - (\frac{\eta_x}{2} - \frac{\eta_x^2 L_F}{2})\|\ell_t\|^2 + \frac{\eta_x}{2}\|\ell_t - \nabla F(x_t)\|^2$$

$$\leq F(x_t) - \frac{\eta_x}{2}\|\nabla F(x_t)\|^2 - \frac{1}{4\eta_x}\|x_{t+1} - x_t\|^2 + \frac{\eta_x}{2}\|\ell_t - \nabla F(x_t)\|^2. \tag{214}$$

Taking expectation over all the randomness and telescoping the preceding inequality from $t = 0$ to $T - 1$, we obtain

$$\frac{1}{T}\sum_{t=0}^{T-1}\mathbb{E}\left[\|\nabla F(x_t)\|^2\right] \leq \frac{2\mathbb{E}\left[F(x_0) - F(x_T)\right]}{\eta_x T} + \frac{2}{T}\sum_{t=0}^{T-1}\mathbb{E}\left[\|\ell_t - \nabla F(x_t)\|^2\right] - \frac{1}{2\eta_x^2}\frac{1}{T}\sum_{t=0}^{T-1}\mathbb{E}\left[\|x_{t+1} - x_t\|^2\right]. \tag{215}$$

By Lemma C.18 and Lemma C.19, we have

$$\frac{1}{T}\sum_{t=0}^{T-1}\mathbb{E}\left[\|\nabla F(x_t)\|^2\right]$$

$$\leq \frac{2\mathbb{E}\left[F(x_0)-F(x_T)\right]}{\eta_x T} + \frac{2}{T}\sum_{t=0}^{T-1}\mathbb{E}\left[\|\ell_t-\nabla F(x_t)\|^2\right] - \frac{1}{2\eta_x^2}\frac{1}{T}\sum_{t=0}^{T-1}\mathbb{E}\left[\|x_{t+1}-x_t\|^2\right] \tag{216}$$

$$\leq \frac{2\mathbb{E}\left[F(x_0)-F(x_T)\right]}{\eta_x T} + \frac{6}{T}\sum_{t=0}^{T-1}\mathbb{E}\left[\|\mathbb{E}[\ell_t]-\nabla\mathcal{L}_\lambda^*(x_t)\|^2\right] + \mathcal{O}\left(\frac{\sigma_f^2}{B}\right) + \mathcal{O}\left(\frac{\lambda^2}{B_J}\right) + \mathcal{O}\left(\lambda^2\gamma^{2H}H^2\right) + \mathcal{O}(\lambda^{-2})$$

$$- \frac{1}{2\eta_x^2}\frac{1}{T}\sum_{t=0}^{T-1}\mathbb{E}\left[\|x_{t+1}-x_t\|^2\right] \tag{217}$$

$$\leq \frac{2\mathbb{E}\left[F(x_0)-F(x_T)\right]}{\eta_x T} + \mathcal{O}\left(\frac{\sigma_f^2}{B}\right) + \mathcal{O}\left(\lambda^{-2}\right) + \mathcal{O}\left(\lambda^2\gamma^{2H}H^2\right) + \mathcal{O}\left(\frac{\lambda^2}{B_J}\right)$$

$$+ \mathcal{O}\left(\frac{\lambda^2 e^{-K}\Delta_\theta}{T}\right) + \left(48C_1\frac{L_\theta}{\mu_\theta}\exp(-\mu_h\eta_\theta K) - \frac{1}{2\eta_x^2}\right)\frac{1}{T}\sum_{t=0}^{T-1}\mathbb{E}\left[\|x_{t+1}-x_t\|^2\right], \tag{218}$$

where $C_1 = \frac{2L_{f,1}^2 + 32\lambda^2 L_{J,1}^2 + 32\lambda^2 L_{J,1}^2 C_\pi^2}{c} = \mathcal{O}(\lambda^2)$ and $\Delta_\theta$ is the initial optimality gap of the parameters defined in (206). By taking $K = \mathcal{O}(\log\lambda)$ such that

$$48C_1\frac{L_\theta}{\mu_\theta}\exp(-\mu_h\eta_\theta K) - \frac{1}{2\eta_x^2} \leq 0, \quad \mathcal{O}(\frac{\lambda^2 e^{-K}\Delta_\theta}{T}) \leq \mathcal{O}(\lambda^{-2}),$$

we have

$$\frac{1}{T}\sum_{t=0}^{T-1}\mathbb{E}\left[\|\nabla F(x_t)\|^2\right] \leq \mathcal{O}\left(\frac{\Delta_F}{T}\right) + \mathcal{O}\left(\frac{\sigma_f^2}{B}\right) + \mathcal{O}(\lambda^2\gamma^{2H}H^2) + \mathcal{O}\left(\lambda^{-2}\right) + \mathcal{O}\left(\frac{\lambda^2}{B_J}\right), \tag{219}$$

where $\Delta_F = \mathbb{E}\left[F(x_0)-F(x_T)\right]$.

Now we consider the requirements for the step sizes to ensure the above bound. For the step size $\eta_x$ in the outer loop, we just need $\eta_x \leq \frac{1}{2L_F}$, which is required by the descent lemma of $F(x)$. For the step sizes $\eta_\theta, \eta_\phi, \eta_\psi$ in the inner loop, we need the conditions in (189) to hold. Thus, $\eta_\phi$ and $\eta_\psi$ should satisfy $\eta_\phi, \eta_\psi \leq \min\{\frac{1}{L_{J,1}}, \frac{1}{\mu_g+1}, 1\}$. Note that $\mu_J = \mu_g$ from Lemma C.7 and C.8. In (189), it can also be verified that $c \asymp \kappa^{-2}$ and $\kappa_b \asymp \kappa^2$. Thus, $\frac{\min\{\eta_\phi,\eta_\psi\}\mu_J}{2+2\mu_h+\frac{\kappa_b}{c}} \asymp \kappa^{-5}$, so we can choose $\eta_\theta \asymp \kappa^{-5}$.

Choosing $T = \mathcal{O}(\epsilon^{-1})$, $\lambda = \mathcal{O}(\epsilon^{-1/2})$, $K = \mathcal{O}(\log\epsilon^{-1})$, $H = \mathcal{O}(\log\epsilon^{-1})$, $B_J = \mathcal{O}(\epsilon^{-2})$, and $B = \mathcal{O}(\epsilon^{-1})$, we obtain

$$\frac{1}{T}\sum_{t=0}^{T-1}\mathbb{E}\left[\|\nabla F(x_t)\|^2\right] \leq \mathcal{O}(\epsilon). \tag{220}$$

The resulting total sample complexity is

$$T \cdot K \cdot (B + B_J) = \tilde{\mathcal{O}}(\epsilon^{-3}).$$

$\square$

# D. Additional Experiment Details

## D.1. Synthetic Problem

In this experiment, all algorithms are evaluated using the same environment and initial parameters. The discount factor $\gamma$ is set to 0.99, and the maximum trajectory length is 3. All algorithms use Monte Carlo sampling to estimate policy gradients, with a batch size of 16. The regularization coefficients $\tau_\phi$ and $\tau_\psi$ are set to 0.1 for all methods. Each algorithm adopts a double-loop structure, with 10 iterations in the inner loop. For the PANDA algorithm, the learning rate for the UL parameter $x$ is set to 0.05, while the learning rates for the policy parameters $\phi$, $\psi$, $\tilde{\phi}$, and $\tilde{\psi}$ are all set to 0.1. The penalty parameter $\lambda$ is set to 4.0. For the PBRL algorithm, the learning rate for $x$ is 0.05, the learning rates for $\phi$ and $\psi$ are 0.1, and we also use a learning rate of 0.1 for updating the policy parameters in the inner loop to estimate $\phi^*$ and $\psi^*$. The penalty parameter $\lambda$ is set to 4.0. For the DA algorithm, the learning rate for $x$ is 0.05, and the learning rates for $\phi$ and $\psi$ are both 0.1.

## D.2. Sentinel-Intruder

### D.2.1. $5 \times 5$ GRID

In this experiment, the environment discount factor $\gamma$ is set to 0.99, the maximum trajectory length is 20, and the regularization coefficient is set to 0.1 for all algorithms. The state in the environment is represented as a $4 \times 5 \times 5$ tensor. The first, second, and third channels are one-hot encodings of the sentinel's position, the intruder's position, and the target location, respectively, while the fourth channel represents the positions of restricted areas. The action space of each policy consists of five actions: moving up, down, left, or right, and staying in place.

Each policy is composed of a convolutional layer followed by a fully connected layer. The convolutional layer uses a kernel size of 3, stride 1, padding 1, and outputs 32 channels. After a ReLU activation and a global average pooling operation, the features are passed through a $32 \times 5$ fully connected layer and a softmax layer to produce a probability distribution over the actions. The parameterized reward function $r_x$ has a similar architecture, consisting of a convolutional layer and a fully connected layer. The convolutional layer shares the same structure as that of the policy network and outputs 32 channels. After ReLU activation and global average pooling, a hidden representation of the state is obtained. Meanwhile, the actions of the two policies are mapped to two 8-dimensional vectors through a fixed embedding layer. The state representation and the two action embeddings are then concatenated and passed through a $48 \times 1$ fully connected layer and a sigmoid layer to produce the final reward value.

All algorithms use Monte Carlo sampling to estimate policy gradients, with a batch size of 64. Each algorithm adopts a double-loop structure, with 10 iterations in the inner loop. The learning rates for the UL parameter $x$ and the policy parameters $\phi$ and $\psi$ are selected by grid search over in $\{1, 2, 5\} \times \{10^{-3}, 10^{-4}, 10^{-5}\}$. The penalty parameter $\lambda$ is set to 4.0 for both the PANDA and PBRL algorithms.

### D.2.2. $20 \times 20$ GRID

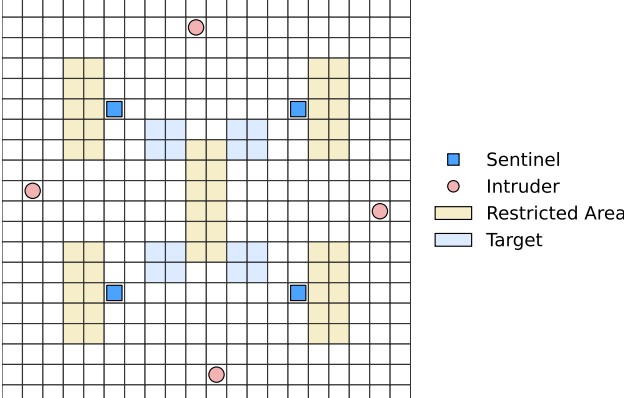

*Figure 5.* The $20 \times 20$ grid environment for the Sentinel-Intruder game. Dark blue cells represent possible sentinel spawn locations, red cells represent possible intruder spawn locations, yellow cells denote restricted areas, and light blue cells indicate target locations.

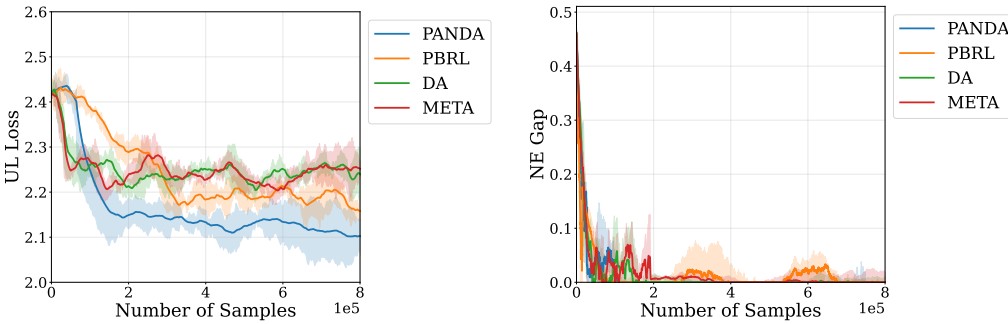

*Figure 6.* Sentinel-Intruder results on the $5 \times 5$ grid, averaged over three random seeds. Left: UL loss vs. number of sampled trajectories. Right: LL NE gap vs. number of sampled trajectories.

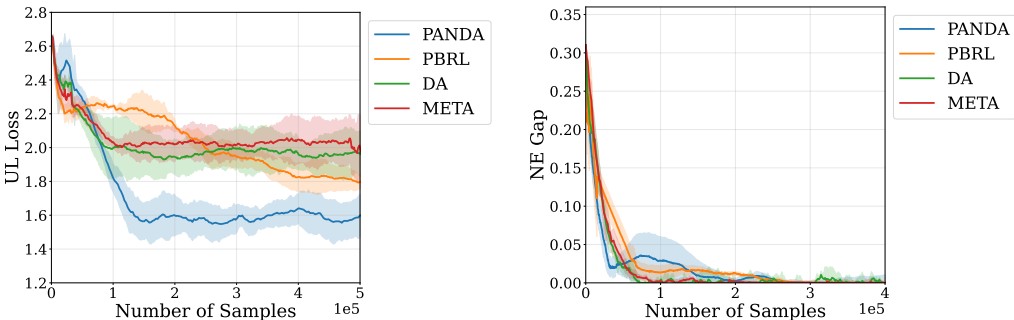

*Figure 7.* Sentinel-Intruder results on $20 \times 20$ grid, averaged over three random seeds. Left: UL loss vs. number of sampled trajectories. Right: LL NE gap vs. number of sampled trajectories.

The basic experimental configuration is the same as in the $5 \times 5$ grid environment, except for the state dimensionality and network architecture. In the environment, each state is represented as a $4 \times 20 \times 20$ tensor, where each channel has the same interpretation as in the $5 \times 5$ grid environment.

We use CNN-based policy networks. The network contains two convolutional layers with $32$ and $64$ output channels, respectively. Both layers use kernel size $3$, stride $1$, and padding $1$, and each is followed by a ReLU activation. The resulting feature map is flattened and fed into a fully connected layer with $256$ hidden units, followed by another ReLU activation. The final linear layer outputs logits over the $5$ discrete actions, and a softmax layer converts these logits into an action probability distribution. The parameterized reward function $r_x$ is also implemented by a CNN-based neural reward model. The state tensor is first encoded by three convolutional layers with $32$, $64$, and $64$ output channels, respectively, each using kernel size $3$, stride $1$, and padding $1$, followed by a ReLU activation. The resulting feature map is processed by adaptive average pooling to obtain a $64$-dimensional state representation. The actions of the two agents are embedded into two $16$-dimensional vectors, which are concatenated with the state representation. The concatenated $96$-dimensional feature vector is then passed through an MLP with one hidden layer of dimension $128$, followed by a sigmoid output layer to produce the final reward value.

All algorithms use Monte Carlo sampling to estimate policy gradients, with a batch size of $64$. Each algorithm adopts a double-loop structure, with $10$ iterations in the inner loop. The learning rates for the UL parameter $x$ and the policy parameters $\phi$ and $\psi$ are selected from the best combination in $\{1, 2, 3, 5, 8\} \times \{10^{-3}, 10^{-4}, 10^{-5}\}$. The penalty parameter $\lambda$ is set to $4.0$ for both the PANDA and PBRL algorithms.

### D.2.3. ADDITIONAL EXPERIMENTAL RESULTS

The results in Figures 6 and 7 show that, in both the $5 \times 5$ and $20 \times 20$ grid environments, all algorithms reduce the LL NE gap to nearly zero, indicating that they can effectively solve the LL problem. However, PANDA achieves lower UL loss than the baseline algorithms.

