# OpenReview forum: "Bilevel Optimization over Saddle Points of Zero-Sum Markov Games"
_ICML.cc/2026/Conference — ICML 2026 regular_

### Official Review · Reviewer_TTNZ · 2026-03-11

**Soundness:** 3
**Presentation:** 4
**Significance:** 3
**Originality:** 3
**Overall Recommendation:** 5
**Confidence:** 2

**Summary:**

This paper studies the bilevel optimization problem where the lower level problem is a zero-sum Markov game via only first-order oracle. This topic remains large unexplored. The complexities they proved match the optimal complexity of the lower level problem. They also provide numerical experiments to compare with existing methods and further confirm the effectiveness of their algorithm. Generally, this paper is well-written and technically sound. The presentation is clear.

**Compliance With Llm Reviewing Policy:**

Affirmed.

**Final Justification:**

I will keep the rating. The rebuttal has resolved my main concerns.

**Key Questions For Authors:**

1. Is it possible to extend the main results where lower level problem assumes unknown transition probabilities and stochastic rewards rather than the full-information setting?
2. How sensitive is the algorithm's performance to its hyperparameters?

**Limitations:**

Yes

**Strengths And Weaknesses:**

Strength:
1. The complexities proved match the optimal complexity of the lower level problem which is nontrivial considering that many additional terms will be involved due to the bilevel structure.
2. The assumptions used in this paper are generally standard, and only first-order oracle is used without the need of second-order oracles. This makes the alogrithm also practically meaningful.
3. Presentations are clear and proofs are easy to follow.

Weakness:
1. Almost all equations in appendix are numbered. It would be better to number equations which are referenced.
2. It would be more convincing to test the algorithms on larger scales (for example, Sentinel-Intruder experiments are only on a grid of 5 $\times$ 5)

---

> ### Author Rebuttal · Authors · 2026-03-31
>
> # Reviewer 4
> Dear Reviewer TTNZ,
>
> We sincerely thank you for your valuable time and feedback.
>
> ---
> W1: Almost all equations in appendix are numbered. It would be better to number equations which are referenced.
>
> >Thank you for this helpful suggestion and we will address it in the revision.
>
> ---
> W2: It would be more convincing to test the algorithms on larger scales (for example, Sentinel-Intruder experiments are only on a grid of $5\times 5$)
>
> > We extend the Sentinel-Intruder experiment to a $20 \times 20$ grid setting, with the same experimental setup as in the paper. It exceeds the scale of many prior works. For example, PBRL (Shen et al. 2025) considers an incentive design problem with $|S|=10$, $|A|=5$, and DA (Wang et al. 2023) evaluates on $7 \times 7$ and $5 \times 5$ grid environments. We adopt neural network policies in this setting. **Please see** our results in **[Fig](https://anonymous.4open.science/r/icml_rebuttal-2072/larger-scale.pdf)**, where PANDA continues to outperform existing methods.
>
> ---
> Q1: Is it possible to extend the main results where lower level problem assumes unknown transition probabilities and stochastic rewards rather than the full-information setting?
>
> > Thank you for the insightful comment.
> >
> > + Our analysis already operates in the unknown-transition, stochastic setting, where transition dynamics are unknown to the policy and all gradients are estimated via Monte Carlo sampling, leading to sample-based convergence guarantees with $\tilde{O}(\epsilon^{-3})$ sample complexity, rather than relying on exact expectations or a full-information oracle.
> >
> > + However, regarding the stochastic reward setting you mentioned, our current theoretical results do not yet cover this case. Addressing it would require reanalyzing the properties of the LL problem, which differ from those under the existing deterministic reward model. This is an important direction for future work toward handling more general RL settings.
>
> ---
> Q2: How sensitive is the algorithm's performance to its hyperparameters?
> > We provide an ablation study on $\lambda$ in Experiment 1 (**Please see [Fig](http://anonymous.4open.science/r/icml_rebuttal-2072/sensitivity.pdf)**):
> >* When $\lambda = 1$: The UL objective achieves a higher value, but the NE gap remains large, indicating that the LL solution is not an equilibrium.
> >* When $\lambda = 4$ or $10$: The NE gap is close to zero, ensuring approximate LL equilibrium.
> >* When $\lambda = 10$: Stronger penalization slightly degrades the UL objective.
> >
> > Overall, $\lambda$ controls a trade-off between LL equilibrium accuracy and UL performance in practice.
>
>
> ---
>
> We sincerely thank you for your suggestions and would greatly appreciate it if our responses help strengthen your confidence in our contributions and improve your evaluation.

---

> > ### Author Rebuttal · Reviewer_TTNZ · 2026-03-31
> >
> > My main concerns have been largely resolved. Especially, the proposed algorithm has a clear advantage on games of larger scales. The results of different $\lambda$ align with intuition and help clarify tradeoff between upper-level and lower-upper problems. It is fine to leave stochastic rewards as future work because technically stochastic rewards will not change the essence of the proofs, only adding more tedious terms.

---

> > > ### Author Response · Authors · 2026-04-02
> > >
> > > We sincerely thank the reviewer for the positive feedback and for recognizing the value of our work. We will consider incorporating the stochastic reward setting into our analysis in future work.

---

### Official Review · Reviewer_jhLf · 2026-03-11

**Soundness:** 3
**Presentation:** 3
**Significance:** 3
**Originality:** 2
**Overall Recommendation:** 4
**Confidence:** 2

**Summary:**

Bilevel reinforcement learning models the interaction between two coupled decision-makers: an active agent optimizing its policy and a higher-level agent whose decisions influence this optimization. Here, both agents are antagonistic, and their objective is to reach a Nash equilibrium as they compete with each other. To this end, a metric measuring the distance to this equilibrium is introduced. This metric is dynamically approximated using policy-gradient updates and optimized with respect to the active agent’s policy. Given this updated policy, the environment subsequently computes its response. The resulting bilevel procedure alternates between these steps, and the algorithm is proven to converge.

**Compliance With Llm Reviewing Policy:**

Affirmed.

**Key Questions For Authors:**

No question

**Limitations:**

yes

**Strengths And Weaknesses:**

Strengths:
- The paper provides a good presentation of recent approaches; although some sections are somewhat dense, it is overall well written.
- I did not have sufficient time to thoroughly check the final proofs, but the ones I examined appeared complete.
- The algorithm is compared with its recent competitors.

Weaknesses:
- The experimental section is somewhat limited given the density and scope of the work, one more benchmark would have been welcome.

---

> ### Author Rebuttal · Authors · 2026-03-31
>
> Dear Reviewer jhLf,
>
> We sincerely thank you for your valuable time and feedback.
>
> In response to your concerns regarding the experimental evaluation, we have further conducted additional experiments at a larger scale.
>
> We extend the Sentinel–Intruder experiment to a $20 \times 20$ grid setting, with the same experimental setup as in the paper. It exceeds the scale of many prior works. For example, PBRL (Shen et al. 2025) considers an incentive design problem with $|S|=10$, $|A|=5$, and DA (Wang et al. 2023) evaluates on $7 \times 7$ and $5 \times 5$ grid environments. We adopt neural network policies in this setting. **Please see** our results in **[Fig](https://anonymous.4open.science/r/icml_rebuttal-2072/larger-scale.pdf)**, where PANDA continues to outperform existing methods.
>
> We sincerely thank you for your suggestions and would greatly appreciate it if our responses help strengthen your confidence in our contributions and improve your evaluation.

---

> > ### Author Rebuttal · Reviewer_jhLf · 2026-04-02
> >
> > The authors clarified all the ambiguous points. I keep my current evaluation.

---

> > > ### Author Response · Authors · 2026-04-02
> > >
> > > We sincerely thank the reviewer for your valuable feedback and for recognizing the value of our work.

---

### Official Review · Reviewer_HoJi · 2026-03-13

**Soundness:** 2
**Presentation:** 2
**Significance:** 3
**Originality:** 2
**Overall Recommendation:** 4
**Confidence:** 3

**Summary:**

This paper proposes a penalty-based first-order policy-gradient method for solving bilevel optimization over saddle points of zero-sum Markov games, supported by experimental results demonstrating improved performance over competing methods. The proposed method adopts the Nikaido–Isoda function to handle the min–max optimization structure in the lower level. The authors also establish the convergence rate and sample complexity of the proposed algorithm.

**Compliance With Llm Reviewing Policy:**

Affirmed.

**Final Justification:**

My main concerns have been addressed. I have adjusted the score accordingly and suggest the authors clarify the main contributions of this work, the formulations of problems (1) and (5), and comparisons with related work in the revision, as stated in the Reply Rebuttal Comment by Authors.

**Key Questions For Authors:**

Q1: For Eq. (5) in Section 2.1, it is stated that $\phi^* $ and $\psi^* $ denote an optimal solution of the lower-level min–max game for a given $x$. Are these solutions unique? Otherwise, how is $F(x)$ well-defined, i.e., how can we ensure that its value does not depend on the particular choice of $\phi^* $ and $\psi^* $? If uniqueness can be guaranteed, what theoretical result establishes this property? Note that this question concerns $\phi^* $ and $\psi^* $, not $\pi_{\phi}^* (x)$ and $\pi_{\psi}^* (x)$.

Q2: Does the recent work [1] listed below apply to the problems considered in this paper? In [1], the authors study bilevel optimization problems with a minimax lower-level structure and propose a Hessian-free algorithm.

[1] Yao et al., Overcoming Lower-Level Constraints in Bilevel Optimization: A Novel Approach with Regularized Gap Functions, ICLR 2025.

Q3: There are two different types of assumptions in the paper. One concerns the Markov games at the lower level (see Assumption 1),  while the other is imposed directly on the lower-level objective function (see Assumption 2). In the setting of this work, these assumptions are closely related. Therefore, it would be helpful to provide a clearer clarification of the theoretical assumptions, for example, explaining why $J$ is Lipschitz continuous under Assumption 1 and the assumptions used in Proposition 1.

Q4: The algorithm involves several parameters, such as the batch size, horizon length, and the penalty parameter. In theory, these parameters all depend on $\epsilon$. How sensitive is the algorithm to these parameters in practice? Are there any guidelines for selecting them?

Minor comment: Please define the PL condition before it is used in Assumption 2.

**Limitations:**

The authors did not adequately discuss the limitations of their work. As mentioned earlier, the paper could be improved by including additional experiments and by clarifying the scope and limitations of the underlying assumptions.

**Strengths And Weaknesses:**

**Strengths:**

S1: This work proposes a penalty-based first-order policy-gradient method to tackle the challenging problem of bilevel optimization over saddle points of zero-sum Markov games, supported by experimental results demonstrating improved performance over competing methods.

S2: The theoretical analysis appears sound, although some assumptions should be clarified.

**Weaknesses:**

W1: In Section 1.2 (Main Contributions of This Work), the authors claim that ``we prove that PANDA converges to approximate stationary points of the original bilevel problem without imposing restrictive conditions, such as strong convexity, on either the UL or LL objectives.” However, in Proposition 1 it is assumed that the regularization functions satisfy the strong convexity–concavity property. Therefore, the proposed method does in fact rely on strong convexity through the regularization terms. It would be helpful to clarify this point either in the abstract or in the statement of contributions, and to provide a clearer discussion of the theoretical assumptions in Section 3.

W2: In the convergence analysis of PANDA, the batch size, horizon length, and penalty parameter are all required to depend on $\epsilon$.

W3: The experiments are limited to small environments and do not include evaluations on large-scale RL benchmarks.

---

> ### Author Rebuttal · Authors · 2026-03-31
>
> We thank reviewer Hoji for your insightful comments and questions.
> ## Convexity-Concavity
> **W1**
> >+ **Convex–concave regularization.** Please note that our analysis does not rely on convexity–concavity of the LL objective. The regularization strength $\tau_\phi$ and $\tau_\psi$ can be chosen as **arbitrarily** small constants for strongly convex–concave regularization, which does not change the nonconvex nature of the LL objective.
> >+ **Regularization in RL.** Strongly convex regularization is standard in RL, which promotes exploration and stabilizes training. Regularized formulations are common in RL/BRL, e.g., KL [1], strongly convex regularizers [2], entropy [3], and our LL is explicitly a **regularized game**. Convex–concave regularization is part of the problem formulation, not an additional assumption.
> >
> >As suggested, we will clarify this point in the revision.
> ## $\epsilon$-dependence
> **W2**
> These quantities are required to attain the prescribed error tolerance $\epsilon$.
>
> > * **Penalty parameter $\lambda$.** Its role is inherent. The distance between the hypergradient of the penalty-based objective and that of the original objective can be controlled as $||\nabla L_\lambda^\*(x)-\nabla F(x)||=O(\lambda^{-1})$ [4]. Therefore, $\lambda$ must be sufficiently large to guarantee convergence to stationary points of the original problem.
>
>
> ## Optimal parameters
> **Q1**
> >The optimal parameters $\phi^\*$ and $\psi^\*$ are not necessarily unique. Note that the UL objective is evaluated at the LL equilibrium policies $\pi_\phi^\*, \pi_\psi^\*$ rather than  $\phi^\*$ and $\psi^\*$. Although multiple $\phi^\*, \psi^\*$ may exist, they all yield the same policies and thus the same value $f(x,\phi^\*,\psi^\*)$.
> ## Related work
> **Q2**
> > The method suggested in reference is not directly applicable to our problem class. Specifically, it (1) assumes a strongly convex–concave LL objective, which is not satisfied in our setting; (2) does not establish sample complexity guarantees; and (3) is not designed for Markov games or RL settings. In this sense, it is complementary rather than directly comparable to our work.
> >
> > That said, we appreciate the reviewer’s suggestion and will include a discussion of this reference in the revised version.
>
>
> ## Assumptions
> **Q3**
> >* Assumption 1 specifies properties of the **Markov game** (policy, reward model, transition dynamics), while Assumption 2 concerns the **optimization problem**, imposing PL and smoothness conditions on the UL and LL objectives. These two sets of assumptions address different aspects of the problem and are standard in BRL (e.g., Assumptions 1 and 3 in [1]).
> >* **Lipschitz continuity of $J$.** We do not state $J$ is Lipschitz under Assumption 1; this is explicitly assumed in Assumption 2.
> >* **Assumptions for Proposition 1.** Proposition 1 does not rely on assumptions in Section 3. It holds for all regularized Markov games under our formulation and is a general result.
> ## Sensitivity
> **Q4**
> >The $\epsilon$-dependence has been clarified in the previous section. In practice, larger batch sizes and horizons reduce sampling variance in the LL Markov game, with bounds $O(\gamma^HH\sigma_H^2)$ and $O(\sigma_J^2/B_J)$, where $\sigma_H$ and $\sigma_J$ are problem-dependent constants (Lemma C.15-C.16).
> >
> >In contrast, $\lambda$ does not monotonically improve performance in practice. We conduct a sensitivity analysis in Experiment 1 (**Please see** [Fig](https://anonymous.4open.science/r/icml_rebuttal-2072/sensitivity.pdf)):
> >* $\lambda=1$: higher UL objective but large NE gap, indicating the LL is not at equilibrium.
> >* $\lambda=4,10$: NE gap approaches zero, ensuring approximate LL equilibrium.
> >* $\lambda=10$: stronger penalization on LL slightly degrades the UL objective.
> >
> >Overall, $\lambda$ controls a trade-off between LL equilibrium accuracy and UL performance.
> ## Larger-scale
> **W3**
> >We extend Experiment 2 to a $20\times 20$ grid. This scale exceeds many prior works, e.g., PBRL with $|S|=10, |A|=5$, and DA on $7\times 7$ grids. Results show that PANDA still outperforms. **Please see** [Fig](https://anonymous.4open.science/r/icml_rebuttal-2072/larger-scale.pdf).
>
> ## Limitations
> We have clarified the assumptions and provided additional experiments. Our work is currently limited to the two-player setting at the LL; extending it to more general multi-agent scenarios is left for future work.
>
> ---
> [1]Gaur et al. 2025, On the sample complexity bounds in bilevel reinforcement learning
>
> [2]Shen et al. 2025, Principled Penalty-based Methods for Bilevel Reinforcement Learning and RLHF
>
> [3]Cen et al. 2021, Fast policy extragradient methods for competitive games with entropy regularization
>
> [4]Chen et al. 2024, On finding small hyper-gradients in bilevel optimization: Hardness results and improved analysis

---

> > ### Author Rebuttal · Reviewer_HoJi · 2026-04-01
> >
> > Thank you for the rebuttal, which addresses some of my concerns. However, several issues remain only partially resolved. Below I provide further details.
> >
> > **W1: Convexity-Concavity**
> >
> > My question concerns whether this work leverages any special structure of the problem being addressed. To illustrate this, consider the following: what is the definition of a stationary point for the original bilevel problem when, as stated in Section 1.2. Main Contributions of This Work, no restrictive conditions, such as strong convexity, on either the UL or LL objectives, are imposed.
> >
> > In particular, what notion of stationary point is adopted in Problem (1)? If the lower-level problem is in fact a regularized game, then the corresponding claims should be revised or clarified accordingly. Note that, the results of Chen et al. (2024), cited in Remark 7, also rely on certain restrictive assumptions on the lower-level problem.
> >
> > **W2: $\epsilon$-dependence**
> >
> > My concern is that the choices of batch size, horizon length, and penalty parameter may be impractical, as they are all required to depend on $\epsilon$ or the total number of iterations. There are approaches to address this issue in bilevel optimization; see, e.g., A Fully First-Order Method for Stochastic Bilevel Optimization (ICML 2023), as well as subsequent works that adopt gradually decreasing penalty parameters.
> >
> > **Q1: Clarification of the formulations of Problems (1) and (5)**
> >
> > In Problem (1), it is not explicitly stated whether the upper-level objective is evaluated at the lower-level equilibrium policies. If this is indeed the case, then the formulation of Problem (1) differs from a standard bilevel optimization problem. As a result, it may not be appropriate to directly compare the contributions of this work with those for general bilevel optimization.
> >
> > **Q2: Related Work**
> >
> > First, the related work does not assume a strongly convex–concave lower-level objective; rather, it assumes a convex–concave structure (cf. Lemma 5.1 therein). I suggested this paper because it studies general bilevel optimization problems with a minimax lower-level structure, which is closely related to the setting considered in this work.
> >
> > **Q3: Assumptions**
> >
> > My original concern is that the focus of this work is an integrated framework (PANDA), and the functions in Assumptions 1 and 2 are closely related. Hence, it would be helpful to provide a clearer and more coherent presentation of the theoretical assumptions.
> >
> > In particular, the lower-level objective $J$ is determined by the min–max zero-sum Markov game. While this is stated in Assumption 2, it remains unclear why the properties assumed for $J$ follow from Assumption 1 together with the conditions used in Proposition 1. A more explicit justification of this connection would be appreciated.
> >
> > Overall, since the work studies an integrated framework, it may be confusing to formulate assumptions using two closely related but distinct sets of notation. It would be helpful to ensure consistency in notation and clarify how these assumptions align.
> >
> > **Limitations**
> >
> > I could not find a location where the authors adequately discuss the limitations of this work. If I have overlooked it, please indicate the specific location.

---

> > > ### Author Response · Authors · 2026-04-02
> > >
> > > Thanks for your insightful questions.
> > > # W1
> > > Our notion of stationarity is measured via $\nabla F(x)$, whose definition follows Lemma 4.3 in Chen et al. (2024). Specifically:
> > > + Our Prop. 1 establishes that the regularized zero-sum min–max Markov game (RZMMG) at LL admits unique equilibrium policies, allowing the LL problem to be equivalently reformulated via the minimization of the NI function $g$, whose minimum is 0 and attained at the Nash equilibrium of the LL problem. Thus, our problem is equivalent to $\min f, s.t. g\leq 0$.
> > > + The assumptions required by their Lemma 4.3 are satisfied by our Assumption 2 and Lemmas 1-2.
> > > + Thus, following their Lemma 4.3, $\nabla F(x)$ is defined as $\lim_{\lambda\to+\infty}\nabla L_\lambda^\*(x)$, where our $L_\lambda^\*(x)$ uses $\lambda$ as penalty parameter, while they use $1/\sigma$ with $\sigma\to 0^+$.
> > > + A point $x$ is $\epsilon$-stationary if $\mathbb{E}||\nabla F(x)||^2\le\epsilon$.
> > >
> > > This notion relies on the NI reformulation and the uniqueness of LL equilibrium, and does not extend to general nonconvex–nonconcave min–max games.
> > > # W2
> > > + **Batch size**
> > >     + We agree that recent works [1,2] propose momentum-based methods to address it in general bilevel optimization (BO). Whether such momentum methods can be used in PANDA to reduce the $\epsilon$-dependence of batch size is a promising direction, but is beyond the scope of this work.
> > >     + Our main contribution is to provide the **first sample complexity analysis** for BO over RZMMG, matching the best-known rates in bilevel RL (BRL).
> > >     + $\epsilon$-dependent batch sizes commonly appear in BRL works [3,4].
> > > + **Horizon H**
> > > This is inherent in RL due to trajectory-based sampling. Even with the discount factor $0<\gamma<1$, the truncation error decays geometrically in $H$ as $\gamma^H$. Hence, to guarantee an error of at most $\epsilon$, it suffices to take $H=O(\log(1/\epsilon))$ (or $O(\log T)$). This dependence on the horizon length is standard in RL and BRL works [3,4].
> > > + **Penalty parameter**
> > > To guarantee convergence to a stationary point of the original problem, $\epsilon$-dependent penalty parameters are generally unavoidable.
> > >     + In [1] mentioned by the reviewer, Thm. 4.3 and Cor. 4.4 require $\delta_k/\lambda_k\le\mu_g\beta_k/8, \delta_k=\gamma_k/\alpha_k-\lambda_k$, where $\lambda_k$ is the penalty parameter. Cor. 4.4 sets $\alpha_k\asymp k^{-3/5},\gamma_k\asymp k^{-2/5}$, implying $\lambda_k\asymp k^{1/5}$. Achieving $\epsilon$-stationarity requires $K=\tilde O(\epsilon^{-5/2})$, thus $\lambda_K=\tilde O(\epsilon^{-1/2})$.
> > >     + In [2], the penalty formulation is $\sigma f+g$, while ours is $f+\lambda g$. These are equivalent up to rescaling, with $\sigma=1/\lambda$. Cor. 5.5 gives $\sigma_k\asymp k^{-1/5}$, which yields $\sigma_K=\tilde O(\epsilon^{1/2})$ for squared gradient norm. Their decreasing $\sigma$ is equivalent to our growing $\lambda$.
> > >     + Both share the same $\epsilon$-dependent scaling with ours.
> > > # Q1
> > > The UL objective is evaluated at the LL equilibrium policies. We will clarify it in the revision. The structure of BO over RZMMG is different from general BO. Accordingly, all comparisons (e.g., Table 1) are restricted to BO works over RL or Markov games.
> > > # Q2
> > > Thanks for your careful reading. It is indeed convex-concave LL. However, as noted before, it does not affect our contributions.
> > > # Q3
> > > + Asm. 1 specifies properties of RZMMG's structural components (reward, policy, etc.). Asm. 2 imposes conditions on the objectives arising in the optimization problem, including $f$, $cf+g$ and the original LL objective $J$.
> > > + Regarding the reviewer’s concern about $J$ in Asm. 2.2, we clarify that $J$ is the value function induced by RZMMG, and its properties are not purely structural properties of the game itself. Specifically:
> > >     + Under Asm. 1, the continuity and smoothness of $J$ can be established using Lemmas 5–6 in [5] together with straightforward derivations. We assume them explicitly to simplify notations. It does not introduce extra restrictions beyond Asm. 1.
> > >     + However, the Hessian Lipschitz of $J$ cannot follow from Asm. 1, and is required to ensure that stationary points are well-defined.
> > > + Such assumptions are common in BRL works [3].
> > > + We will clarify the connections in the revision.
> > >
> > >
> > > # limitations
> > > We acknowledge that the limitations were not adequately discussed. Our LL is limited to RZMMG and does not extend to general min–max or multi-agent settings. The limitations raised in rebuttal were intended as clarification. We will include this discussion in revision.
> > >
> > > ---
> > > [1]A Fully First-Order Method for Stochastic Bilevel Optimization
> > >
> > > [2]On Penalty Methods for Nonconvex Bilevel Optimization and First-Order Stochastic Approximation
> > >
> > > [3]On the sample complexity bounds in bilevel reinforcement learning
> > >
> > > [4]Bilevel reinforcement learning via the development of hyper-gradient without lower-level convexity
> > >
> > > [5]Regularized Gradient Descent Ascent for Two-Player Zero-Sum Markov Games

---

### Official Review · Reviewer_GyNx · 2026-03-18

**Soundness:** 3
**Presentation:** 3
**Significance:** 2
**Originality:** 2
**Overall Recommendation:** 4
**Confidence:** 2

**Summary:**

The paper proposes PANDA, a first-order stochastic policy-gradient method for bilevel RL where the lower level is a regularized min-max zero-sum Markov game. The authors use the Nikaido-Isoda function as a penalty to encode the lower-level equilibrium constraint, avoiding hypergradient computation and Hessian inversion. The main result is that PANDA matches the best-known iteration and sample complexity for single-policy bilevel RL, despite the harder two-player lower-level structure.

**Compliance With Llm Reviewing Policy:**

Affirmed.

**Key Questions For Authors:**

1. How does the complexity scale with regularization strength $\tau_\phi, \tau_\psi$? What happens as $\tau \to 0$?
2. Can you run at least one experiment with neural network policies on a larger game?
3. How do the stationary points found by PANDA compare to the global optimum in your experiments?

**Limitations:**

Partially discussed. Regularization dependence and stationary-point limitation should be more prominent.

**Strengths And Weaknesses:**

**Strengths:**

- Extension from single-policy LL to min-max game LL is technically non-trivial — the PL property of the NI function (Lemma 1) requires careful handling of two-player coupling
- First-order only, no Hessians — practical advantage over DA and PARL. Convergence to the *original* bilevel problem (not the penalized surrogate) is meaningful
- Table 1 clearly positions the contribution; PANDA fills a genuine gap (stochastic + first-order + min-max LL)

**Weaknesses:**

- The approach follows the existing penalty-based BRL template (Gaur et al. 2025, Chen et al. 2024, Shen et al. 2025/PBRL) closely. The NI function itself is classical. Once the structural properties (PL, smoothness) are verified for the game setting, the convergence analysis is routine
- Heavy reliance on entropy regularization ($\tau_\phi, \tau_\psi > 0$). The PL constant degrades with $\delta_\pi^2$ (minimum policy probability), and the paper never discusses what happens as $\tau \to 0$
- Experiments are tiny: $|S|=5, |A|=|B|=3$ tabular + a 5x5 grid. No function approximation despite the introduction promising large-scale applications. Theory assumes tabular softmax but experiments use CNNs — gap not addressed
- Missing comparison with Meta-Gradient (Yang et al. 2022) and ablation on penalty parameter $\lambda$

---

> ### Author Rebuttal · Authors · 2026-03-31
>
> We thank reviewer GyNx for your helpful comments and questions.
> ## Clarification on Contributions
> **W1:** The approach follows existing penalty-based BRL templates.
> > Our work is inspired by penalty-based BRL frameworks, but differs in both setting and design.
> >+ **Comparison with [1,2].** Our method targets a more challenging min–max LL problem. Unlike single-policy BRL with LL objective $J(x,\theta)$, our NI function at LL is $g(x,\phi,\psi)=\max_{\psi'}J(x,\phi,\psi')-\min_{\phi'}J(x,\phi',\psi)$, which introduces an additional level of optimization. A naïve extension of their templates [1,2] yields a triple-loop algorithm, whereas our algorithm preserves the overall double-loop structure by solving the LL game within a single inner loop, without any loss in complexity guarantees.
> > + **Comparison with [3].** We adopt a hierarchical scheme: first solve $\min_{\phi,\psi}L_\lambda(x,\phi,\psi)$, and then optimize $\min_x L_\lambda^\*(x)$. In contrast, [3] directly optimizes $L_\lambda$ jointly over $x,\phi,\psi$. This hierarchical separation allows us to control the gradient gap $||\nabla L_\lambda^\*(x)-\nabla F(x)||=\mathcal{O}(\lambda^{-1})$, ensuring convergence to stationary points of the original problem, which cannot be established in [3]. Also, [3] lacks stochastic analysis.
>
> **W1**: Convergence analysis is routine.
> > As mentioned above, the reduced loop structure of the LL game solver requires a new contraction-property analysis beyond existing work, as the coupled variables $\phi,\psi,\phi',\psi'$ are updated in opposite directions and must be jointly controlled.
> ## Regularization
> **W2:** Heavy reliance on entropy regularization.
> > Regularization has long been a standard and practical component in RL, promoting exploration and stabilizing training in practice through variants such as KL [1] and entropy [5]. In this work, we consider regularized Markov games, which have been widely studied [4].
>
> **W2:** The PL constant degrades with the minimum policy probability.
> > The derived constant characterizes the intrinsic geometric structure of the problem itself, which is consistent with the single-policy scenario [6].
>
> **W2 & Q1:** What happens as $\tau\to 0$? How does complexity scale with $\tau$?
> > Our complexity analysis focuses on the regularized regime with constant $\tau>0$, where $\tau$ can be chosen sufficiently small before runing the algorithm. This is consistent with prior BRL works [1,3,5]. The dependence of complexity on $\tau$, as well as the regime $\tau\to 0$, is beyond the scope of this work.
> ## Parameterization
> **W3**: No function approximation despite the introduction promising large-scale applications.
> > Extending the analysis to general function approximation requires additional structural assumptions, which we leave as future work.
>
> **W3**: Theory assumes tabular softmax while experiments use CNNs.
> > Experiments with both tabular softmax policies (Experiment 1) and neural network policies (Experiment 2) demonstrate the superiority of PANDA over competing methods.
> ## Experiments
> **W4 & Q2 & Q3**
>
> **Please see** all experimental results in **[Fig](https://anonymous.4open.science/r/icml_rebuttal-2072/main.pdf)**.
> > **Meta-Gradient**. Included. Inferior to PANDA.
>
> > **Comparison with oracle.** Due to the nonconvexity of the problem, computing exact global optima is intractable. We construct a strong oracle based on dynamic programming to get the exact value function, and use 2nd-order information to compute exact hypergradients. Due to high computational cost, this oracle is only used in Experiment 1.
>
> >**Ablation on $\lambda$.** Results in Experiment 1:
> >* $\lambda=1$: higher UL objective but large NE gap, indicating the LL is not at equilibrium.
> >* $\lambda=4,10$: NE gap approaches zero, ensuring approximate LL equilibrium.
> >* $\lambda=10$: stronger penalization on LL slightly degrades the UL objective.
> >
> >$\lambda$ controls a trade-off between LL equilibrium accuracy and UL performance.
>
> >**Larger scale**. We extend Experiment 2 to a $20\times 20$ grid. This size exceeds many prior works, e.g., PBRL with $|S|=10,|A|=5$, and DA on $7\times 7$ grids. We use CNN policies. Results show that PANDA still outperforms.
> ## Limitations
> Our work currently focuses on the two-player setting; extending it to multi-agent scenarios is left for future work.
>
> ---
> [1]Gaur et al. 2025, On the sample complexity bounds in bilevel reinforcement learning
>
> [2]Chen et al. 2024, On finding small hyper-gradients in bilevel optimization: Hardness results and improved analysis
>
> [3]Shen et al. 2025, Principled Penalty-based Methods for Bilevel Reinforcement Learning and RLHF
>
> [4]Cen et al. 2021, Fast policy extragradient methods for competitive games with entropy regularization
>
> [5]Yang et al. 2025, Bilevel reinforcement learning via the development of hyper-gradient without lower-level convexity
>
> [6]Mei et al. 2020, On the Global Convergence Rates of Softmax Policy Gradient Methods

---

### Decision · Program_Chairs · 2026-04-30

**Decision:**

Accept (regular)

**Comment:**

This paper studies bilevel optimization problems where the lower-level problem involves finding saddle points of zero-sum Markov games, a challenging setting that combines hierarchical optimization with game-theoretic equilibrium computation in RL. The reviewers acknowledged the novelty and technical depth of the problem formulation, which connects two important areas—bilevel optimization and multi-agent RL—in a principled manner. The proposed algorithms come with convergence guarantees, and the theoretical analysis carefully addresses the unique challenges arising from the interplay between the bilevel structure and the saddle-point nature of the lower-level problem. While some reviewers initially raised concerns about the clarity of presentation and the practical scope of the setting, the authors provided thorough rebuttals that clarified the key technical contributions and better situated the work relative to existing approaches.

After reviewing the full discussion, I find that the paper makes a meaningful theoretical contribution to an important and relatively unexplored problem at the intersection of bilevel optimization and game-theoretic RL. The technical results are sound, the problem is well-motivated, and the authors have been responsive to reviewer feedback. I recommend acceptance as a poster.